# $\beta$-STOCHASTIC SIGN SGD: A BYZANTINE RESILIENT AND DIFFERENTIALLY PRIVATE GRADIENT COMPRESSOR FOR FEDERATED LEARNING

## ABSTRACT

Federated Learning (FL) is a nascent privacy-preserving learning framework under which the local data of participating clients is kept locally throughout model training. Scarce communication resources and data heterogeneity are two defining characteristics of FL. Besides, a FL system is often implemented in a harsh environment, leaving the clients vulnerable to Byzantine attacks. To the best of our knowledge, no gradient compressors simultaneously achieve quantitative Byzantine resilience and privacy preservation. In this paper, we fill this gap via revisiting the stochastic sign SGD Jin et al. (2020). We propose $\beta$-stochastic sign SGD, which contains a gradient compressor that encodes a client's gradient information in sign bits subject to the privacy budget $\beta > 0$. We show that $\beta$-stochastic sign SGD converges in the presence of partial client participation, mobile static and adaptive Byzantine faults, and that it achieves quantifiable Byzantine-resilience and differential privacy simultaneously even with non-IID local data. We show that our compressor works for both bounded and unbounded stochastic gradients, i.e., both light-tailed and heavy-tailed distributions. As a byproduct, we show that when the clients report sign messages, the popular information aggregation rules simple mean, trimmed mean, median and majority vote are identical in terms of the output signs. Our theories are corroborated by experiments on MNIST and CIFAR-10 datasets.

## 1 INTRODUCTION

Federated Learning (FL) is a nascent learning framework that enables privacy sensitive clients to collectively train a model without disclosing their raw data McMahan et al. (2017); Kairouz et al. (2021). Expensive communication overhead and non-IID local data are two defining characteristics of FL. A variety of communication-saving techniques have been introduced, including periodic averaging McMahan et al. (2017), large mini-batch sizes Lin et al. (2020), and gradient compressors Xu et al. (2020); Alistarh et al. (2017); Bernstein et al. (2018; 2019); Jin et al. (2020); Safaryan et al. (2021); Wang et al. (2021). However, challenges remain.

A FL system is often massive in scale and is implemented in harsh environment – leaving the clients vulnerable to unstructured faults such as Byzantine faults Lynch (1996). Moreover, FL clients are privacy-sensitive. Despite clients' privacy is partially preserved via denying raw data access, quantitative privacy preservation is still desirable. Observing this, Bernstein et al. (2019) proposed signSGD with majority vote which is provably resilient to Byzantine faults. However, even in the absence of Byzantine faults, SignSGD fails to converge in the presence of non-IID data Safaryan & Richtárik (2021); Chen et al. (2020), and is not differentially private. To handle non-IID data, Jin et al. (2020) proposed stochastic sign SGD and its differentially-private (DP) variant, whose gradient compressors are simple yet elegant. Unfortunately, their DP variant does not converge [1], and their standard stochastic sign SGD is not differentially-private (shown in our Theorem 1). We will discuss the relations between Jin et al. (2020) and our work in the related work.

---

[1] Their Theorem 6 analysis contains major flaws.

**Contributions.**    In this paper, we revisit the elegant compressor in Jin et al. (2020). We propose $\beta$-stochastic sign SGD, which contains a gradient compressor that encodes a client's gradient information in sign bits subject to the privacy budget $\beta > 0$, and works for unbounded and mini-batch stochastic gradients. A parameter $B > 0$ is chosen carefully to clip the unbounded gradients.

- We first show (in Theorem 1) that when $\beta = 0$, the compressor is not differentially private. In sharp contrast, when $\beta > 0$, the compressor is $d \cdot \log\big((2B + \beta)/\beta\big)$-differentially private, where $d$ is the gradient dimension. We provide a finer characterization of the differential privacy preservation (in Theorem 2 and Corollary 1). In addition, to help the readers interpret our DP, we show (in Proposition 2) that our compressor with $\beta > 0$ can be viewed as a composition of a randomized sign flipping and stochastic sign SGD compressor. To the best of our knowledge, this is the first result to establish DP with signed compressors in FL.

- We show (in Theorem 4) $\beta$-stochastic sign SGD works for both bounded and unbounded stochastic gradient. Specifically, convergence bounds are derived for both light-tailed and heavy-tailed stochastic gradients. In addition, we show (in Theorem 4) the convergence of $\beta$-stochastic sign SGD in the presence of partial client participation and mobile Byzantine faults, showing that it achieves Byzantine-resilience and DP simultaneously. Both static and adaptive adversaries are considered.

- As a byproduct, we show (in Proposition 1) that when the clients report sign messages, the popular information aggregation rules simple mean, trimmed mean, median, and majority vote are identical in terms of the output signs. This implies majority vote is a counterpart of "middle-seeking" Byzantine resilient algorithms in the realm of sign aggregations.

- Our theoretical findings are validated with experiments on the MNIST and CIFAR-10 datasets

## 2    RELATED WORK

**Communication Efficiency.**    Communication is a scare resource in FL McMahan et al. (2017); Kairouz et al. (2021). Numerous efforts have been made to improve the provable communication efficiency of FL. FedAvg – the most widely-adopted FL algorithm – and its Algorithm 1s save communication via performing multiple local updates at the client side McMahan et al. (2017); Wang & Joshi (2019); Stich (2019); Li et al. (2020a). Large mini-batch size is another communication-saving technique yet its performance turns out be often inferior to FedAvg Lin et al. (2020). Gradient compressors Xu et al. (2020) take the physical layer of communication into account and are used to reduce the number of bits used in encoding local gradient information. Quantized SGD (QSGD) Alistarh et al. (2017) is a lossy compressor with provable trade-off between the number of bits communicated per iteration with the variance added to the process. However, its performance is shown to be inferior to simple compressor such as SignSGD Bernstein et al. (2019), which, based on the sign, compresses a local gradient into a single bit. Nevertheless, SignSGD fails to converge in the presence of non-IID data Safaryan & Richtárik (2021); Chen et al. (2020), and is not differentially private. This is because SignSGD neglects the information contained in the gradient magnitude.

**Byzantine Resilience.** Despite its popularity, FedAvg is vulnerable to Byzantine attacks on the participating clients Kairouz et al. (2021); Blanchard et al. (2017); Chen et al. (2017). This is because that under FedAvg the PS aggregates the local gradients via simple averaging. Alternative aggregation rules such as Krum Blanchard et al. (2017), geometric medianChen et al. (2017), coordinate-wise median and trimmed mean Yin et al. (2018) are shown to be resilient to Byzantine attacks though different in levels of resilience protection with respect to the number of Byzantine faults, the model complexity, and underlying data statistics in the presence of IID local data. Assuming the PS can get access to sufficiently many freshly drawn data samples in each iteration, Xie et al. Xie et al. (2019) proposed an algorithm *Zeno* that can tolerate more than 1/2 fraction of clients to be Byzantine. Unfortunately, their analysis is restricted to homogeneous and balanced local data using techniques from robust statistics. However, it is not straightforward to extend the results to non-IID data, which stems from the difficulty of distinguishing the statistical heterogeneity from Byzantine attacks Li et al. (2019). Many efforts have been devoted to mitigate the negative impacts stemming from heterogeneous data Ghosh et al. (2019); Karimireddy et al. (2022). Ghosh et al.Ghosh et al. (2019) used robust clustering techniques whose correctness crucially relies on large local dataset and local cost functions to be strongly convex. Karimireddy et al. Karimireddy et al. (2022) derived

convergence under the strong assumption of bounded dissimilarity of local gradients, which often does not hold when the data efficiency is taken into account Su et al. (2022).

**Privacy Preservation.** FL is renowned for its capability to decouple model training from the raw data collections by communicating only model/gradient parameters McMahan et al. (2017) between the clients and the PS. However, recent results show that both weight and gradient sharing schemes may leak sensitive information Zhu et al. (2019); Phong et al. (2018). Two notions of differential privacy exist in FL Truex et al. (2020): (A) central privacy, where a trusted server masks the data and shares the perturbed updates with the distributed clients, and (B) local privacy, where each client protects their data from any external parties, including the server. Most of the current works focus on the perturbed updates by adding Laplace Huang et al. (2015), GaussianWang et al. (2021), or Binomial perturbation Agarwal et al. (2018). The former two are applied to traditional gradients/updates, and the latter is applied to compressed gradients/updates whose values are represented in terms of bits. Focusing on distributed mean estimation problem, Agarwal et al. (2018) proposed a communication-efficient algorithm that achieves the same privacy and error trade-off as that of the Gaussian mechanism provided that the model dimension is at most comparable to the client population size. In this work, we focus on protecting local privacy.

**Comparison with Jin et al. (2020).** While we admit that $\beta$-stochastic sign SGD and sto-sign compressor in Jin et al. (2020) are structurally alike, our results depart in significant ways: their theoretical guarantees are flawed since no explicit forms are given for the residual terms and Byzantine resilience. They consider only full-batch gradients, which may not hold as edge clients in FL are often with limited computing power and storage McMahan et al. (2017). On top of all, no partial client participation is considered. In a sharp contrast, our work contains a quantitative characterization of the interactions between FL system hyperparameters such as client number $M$, mini-batch size $n$, sampling rate $p$, etc. We show our work converges in the presence of both static and adaptive mobile Byzantine adversaries. We build upon our $\beta$ compressor on mini-batch stochastic gradient and derive the convergence bounds under light-tailed and heavy-tailed noise using a variety of concentration bounds, which is technically non-trivial. We also show our compressor works for partial client participation. We reserve a point-by-point comparison in Appendix A.

## 3 PROBLEM SETUP

We consider a cross-device FL system, where a large number of clients are involved Kairouz et al. (2021). The system consists of one parameter server (PS) and $M$ clients that collaborate to minimize

$$\min_{w \in \mathbb{R}^d} F(w) := \frac{1}{M} \sum_{m=1}^{M} f_m(w), \tag{1}$$

where $f_m(w) := \mathbb{E}_{\mathcal{D}_m}[f_m(w, x, y)]$ is the local cost function at client $m \in [M] := \{1, \cdots, M\}$ with the expectation taken over heterogeneous local data $(x, y) \sim \mathcal{D}_m$.

*Client unavailability.* Clients are also heterogeneous in their computation speeds and communication channel conditions, which result in intermittent clients unavailability. To capture this, following the literature Kairouz et al. (2021); Li et al. (2020b); Philippenko & Dieuleveut (2020), instead of full participation, we assume that, in each iteration, a client successfully uploads its local update with probability $p$ independently across rounds, and independently from the PS and other clients.

*Mobile Byzantine attacks.* In each iteration $t$, up to $\tau$ clients suffer Byzantine faults. Denote by $\mathcal{B}(t) \subseteq [M]$ the set of clients are Byzantine in iteration $t$, which is unknown to the PS. Let $\tau(t) = |\mathcal{B}(t)|$. We refer to the clients in $\mathcal{B}(t)$ as Byzantine clients at iteration $t$. We consider both static and adaptive system adversaries. In the former, the system adversary does not know client unavailability in each iteration; in the latter, the system adversary adaptively chooses $\mathcal{B}(t)$ according to the client unavailability in each iteration $t$.

*Differential privacy.* In addition to Byzantine resilience, we also aim to provide quantitative privacy protection. Towards this, we use differential privacy framework.

**Definition 1** (Definition 2.4 Dwork et al. (2014))**.** *For any $\epsilon > 0$, a randomized algorithm $\mathcal{M}$ with domain $\mathbb{N}^{|\mathcal{X}|}$ is $\epsilon$-differentially private if $\mathbb{P}\{\mathcal{M}(x) \in \mathcal{S}\} \leq \exp(\epsilon)\mathbb{P}\{\mathcal{M}(y) \in \mathcal{S}\}$ holds for all $\mathcal{S} \subseteq Range(\mathcal{M})$ and for all $x, y \in \mathbb{N}^{|\mathcal{X}|}$ such that $\|x - y\|_1 \leq 1$.*

## 4 Algorithm

Due to the randomness in data $(x, y)$, $\nabla_i F(w, x, y)$ varies significantly across different realizations, and could even be unbounded. To cope with this, in Algorithm 1, we clip the gradient coordinate-wise, which are then used to stochastically generate signs. Formal definitions are given next.

**Definition 2.** *The clipping function with parameter $B$, denoted by $\mathsf{clip}\{\cdot, B\}$, projects $g \in \mathbb{R}$ onto $[-B, B]$ as $\mathsf{clip}\{g, B\} = \max\{-B, \min\{B, g\}\}$.*

**Definition 3.** *For any privacy budget $\beta \geq 0$ and any clipping parameter $B > 0$, we define a gradient compressor $\mathcal{M}_{B,\beta}$ as*

$$[\mathcal{M}_{B,\beta}]_i(\boldsymbol{g}) := \begin{cases} 1, & \text{with probability } \frac{B+\beta+\mathsf{clip}\{g_i, B\}}{2B+2\beta}; \\ -1, & \text{otherwise,} \end{cases} \tag{2}$$

*where $g_i$ and $[\mathcal{M}_{B,\beta}]_i(\boldsymbol{g})$ are the $i$-th coordinates of $\boldsymbol{g}$ and $\mathcal{M}_{B,\beta}(\boldsymbol{g})$, respectively.*

---

**Algorithm 1:** Federated Learning with $\beta$-Stochastic Sign SGD

**Input:** $T, \eta, \beta, n, B$, and $\nu$
**Output:** $w(T)$

1 **Initialization:** $w(0) \leftarrow \nu$ for each $m \in [M]$, and the PS samples each client $m \in [M]$ with probability $p$ to form $\mathcal{S}(0)$;
2 **for** $t = 0, \cdots, T - 1$ **do**
    /* On each $m \in \mathcal{S}(t) \setminus \mathcal{B}(t)$     */
3     Get $n$ stochastic gradients $\boldsymbol{g}_m^1(t), \ldots, \boldsymbol{g}_m^n(t)$;
4     **for** $i = 1, \cdots, d$ **do**
5         $\widehat{g}_{mi}(t) \leftarrow 1$ with probability $\frac{B+\beta+\mathsf{clip}\{\frac{1}{n}\sum_{j=1}^n \boldsymbol{g}_{mi}^j(t), B\}}{2B+2\beta}$; $\widehat{g}_{mi}(t) \leftarrow -1$ otherwise.
6     **end**
7     Report $\widehat{g}_m(t)$ to the PS;
    /* On the PS     */
8     Wait to receive messages $\widehat{u}_m(t) \in \mathbb{R}^d$ from the sampled clients $\mathcal{S}(t)$
9     $\widetilde{\boldsymbol{g}}(t) \leftarrow \mathsf{sign}(\mathsf{agg}\{\widehat{u}_m(t) : m \in \mathcal{S}(t)\})$
10     Sample each client $m \in [M]$ with probability $p$ to obtain $\mathcal{S}(t+1)$;
11     Broadcast $\widetilde{\boldsymbol{g}}(t)$ to all clients;
    /* On each client $m \in \mathcal{S}(t+1) \setminus \mathcal{B}(t)$     */
12     Upon receiving $\widetilde{\boldsymbol{g}}(t)$: $w(t+1) \leftarrow w(t) - \eta \widetilde{\boldsymbol{g}}(t)$;
13 **end**

---

Our algorithm is formally described in Algorithm 1, which takes $T, \eta, \beta, n, B$, and $\nu \in \mathbb{R}^d$ as inputs, where $T$ is the iteration horizon $T$, $\eta > 0$ is the stepsize, $\beta \geq 0$ is the privacy budget, $n$ is the mini-batch size of local stochastic gradients, $B > 0$ is a parameter used to clip stochastic gradients, and $\nu$ serves as the initial values of $w(0)$.

In each iteration $t$ of Algorithm 1, a client $m$ is selected by the PS with probability $p$. Let $\mathcal{S}(t)$ be the set of selected clients at time $t$. Since Byzantine clients can deviate from Algorithm 1 arbitrarily, lines 2-8 are executed at clients in $\mathcal{S}(t) \setminus \mathcal{B}(t)$ only. In each iteration $t$, client $m \in \mathcal{S}(t) \setminus \mathcal{B}(t)$ first obtains $n$ stochastic gradients $\boldsymbol{g}_m^1(t), \ldots, \boldsymbol{g}_m^n(t)$. Then it passes $\frac{1}{n}\sum_{j=1}^n \boldsymbol{g}_m^j(t)$ to $\mathsf{clip}\{\cdot, B\}$ coordinate-wise, and compresses the clipped gradient via the compressor $\mathcal{M}_{B,\beta}$ in lines 5 and 6. For ease of exposition, let $\widehat{g}_{mi}(t) = [\mathcal{M}_{B,\beta}]_i(\frac{1}{n}\sum_{j=1}^n \boldsymbol{g}_m^j(t))$. Finally, client $m$ reports $\widehat{g}_m(t)$ to the PS. On the PS side (lines 8 - 12), it first waits to receive messages $\widehat{u}_m(t)$ from the sampled clients, for which $\widehat{u}_m(t) = \widehat{g}_m(t)$ if $m \notin \mathcal{B}(t)$. Then the PS passes $\{\widehat{u}_m : m \in \mathcal{S}(t)\}$ to an aggregation function $\mathsf{agg}$, and takes the coordinate-wise sign of the function output to obtained $\widetilde{\boldsymbol{g}}(t)$. Next, it broadcasts $\widetilde{\boldsymbol{g}}(t)$ to $\mathcal{S}(t+1)$.

For convenience of exposition, if no message is received from a selected client $m$ (which only occurs when $m \in \mathcal{B}(t)$), then the PS treats $\widehat{u}_m$ as $\mathbf{0}$. Notably, if $m \in \mathcal{B}(t)$, the received $\widehat{u}_m(t)$ could take arbitrary value. Since $\widehat{g}_m \in \{\pm 1\}^d$ for $m \notin \mathcal{B}(t)$, if $\widehat{u}_{mi}(t) \notin \{-1, 1\}$, then it must be true that client $m \in \mathcal{B}(t)$. Thus, $\widehat{u}_{mi}(t)$ will be removed from aggregation by the PS. In other words, it is always a better strategy for a Byzantine client to restrict $\widehat{u} \in \{\pm 1\}^d$. Henceforth, without loss of generality, we assume that $\widehat{u}_m(t) \in \{\pm 1\}^d$ for all received compressed gradients.

## 4.1 Aggregation Functions

Simple mean (i.e. naive averaging) is one of the widely-adopted aggregation rule of FL algorithms Kairouz et al. (2021); Blanchard et al. (2017), yet it is vulnerable to Byzantine attacks. Alternative aggregation rules such as Krum Blanchard et al. (2017), geometric median Chen et al. (2017), coordinate-wise median and trimmed mean Yin et al. (2018) are shown to be resilient to Byzantine attacks yet with different levels of resilience protection, we show that when the inputs of the aggregation functions are a collection of binary vectors in $\{\pm 1\}^d$, the signs of the outputs of the simple mean, trimmed mean, and median aggregation rules are identical. Moreover, they are all equivalent to the coordinate-wise majority vote aggregation rule. We denote by $\text{agg}_{avg}$, $\text{agg}_{trimmed,k}$, $\text{agg}_{median}$, and $\text{agg}_{maj}$, the coordinate-wise mean, $k$-trimmed-mean, median, and majority vote aggregation rules, respectively, whose definitions are deferred to Appendix B.

**Proposition 1.** *For any given $\mathcal{S} \subseteq [M]$ and any given $\widehat{u}_m \in \{\pm 1\}^d$ for $m \in \mathcal{S}$, the aggregation rules $\text{agg}_{avg}$, $\text{agg}_{trimmed,k}$ with $k < |\mathcal{S}|/2$, $\text{agg}_{median}$, and $\text{agg}_{maj}$ are equivalent in terms of their signs.*

## 5 Privacy Preservation

In this section, we characterize the DP of our gradient compressor $\mathcal{M}_{B,\beta}$. Over the entire training time horizon, the quantification of the differential privacy preserved for any given client can be obtained by applying the composition theorem of $\epsilon$-differentially private algorithms (Dwork et al., 2014, Corollary 3.15). We first show that $\beta$ is an enablor of DP for our compressor.

**Theorem 1.** *$\mathcal{M}_{B,0}$ is not differentially private. That is, there does not exist a finite $\epsilon > 0$ for which Definition 1 holds. When $\beta > 0$, $\mathcal{M}_{B,\beta}$ is $d \cdot \log\left(\frac{2B+\beta}{\beta}\right)$-DP for all gradients.*

Theorem 1 also implies that as long as $\beta > 0$, $\mathcal{M}_{B,\beta}$ ensures $\epsilon$-differential privacy for $\epsilon = O(d)$ at any iteration $t$. This might be pessimistic in the presence of a deep neural network. However, it is possible to reduce the order $\mathcal{O}(d)$, and we pave the way as outlined in Theorem 2 and Corollary 1.

**Definition 4.** *For any given $B > 0$, let $\mathcal{C}_B := (-\infty, -B) \cup (B, \infty)$. For each $g \in \mathbb{R}$, define $\text{dist}(g, \mathcal{C}_B) := \inf_{g' \in \mathcal{C}_B} |g - g'|$.*

**Theorem 2.** *Let $g, g' \in \mathcal{G} \subseteq \mathbb{R}^d$ be an arbitrary pair of gradient inputs such that $g' \neq g$, and define $l_1$ sensitivity as $\Delta_1 := \max_{g,g' \in \mathcal{G}} \|g - g'\|_1$. $\mathcal{M}_{B,\beta}$ is $\max_{g \in \mathcal{G}} \sum_{i=1}^d \log\left(1 + \frac{\Delta_1}{\beta + \text{dist}(g_i, \mathcal{C}_B)}\right)$-DP on $\mathcal{G}$ for $\beta > 0$.*

**Corollary 1.** *Given the same definitions as in Theorem 2, $\mathcal{M}_{B,\beta}$ is $\left(\frac{\Delta_1}{\beta}\right)$-DP.*

**Remark 1.** *Theorem 2 gives a finer characterization of the differential privacy preserved by $\mathcal{M}_{B,\beta}$ when $\beta > 0$. Unfortunately, this maximum is often hard to find, so Corollary 1 tells us that we can get rid of the order $\mathcal{O}(d)$ by controlling the $l_1$ sensitivity $\Delta_1$. One way is to let $B = \Delta_1/d$.*

**Definition 5.** *DP-flip mechanism $\mathcal{M}_{B,flip} : \{\pm 1\} \to \{\pm 1\}$ is defined as: For any $b \in \{\pm 1\}$,*

$$\mathcal{M}_{B,flip}(b) = \begin{cases} b & \text{with probability } \frac{2B+\beta}{2(B+\beta)}; \\ -b & \text{otherwise.} \end{cases} \tag{3}$$

It can be easily checked that $\mathcal{M}_{B,flip}$ is $\log\left(\frac{2B+\beta}{\beta}\right)$-DP.

**Proposition 2.** *For any $\beta > 0$, $\mathcal{M}_{B,\beta} \stackrel{d.}{=} \mathcal{M}_{B,flip} \circ \mathcal{M}_{B,0}$, where $\stackrel{d.}{=}$ denotes "equal in distribution".*

In Proposition 2, we decompose our $\mathcal{M}_{B,\beta}$ as a composition of $\mathcal{M}_{B,0}$ and $\mathcal{M}_{B,flip}$ to help the readers interpret the realization of our DP. A simple example of the equivalence in Proposition 2 is given in Fig.1. Under $\mathcal{M}_{B,0}$, an occurrence of $-1$ in any round of one of experiments suggests its input gradient is $g'$ rather than $g$. Even if we constrain the set of gradients, the ensured differential privacy is not controllable (implied by the proof of Theorem.2). In contrast, under $\mathcal{M}_{B,\beta}$, one can manipulate $\beta$ to ensure a controllable privacy quantification as per Theorem.1.

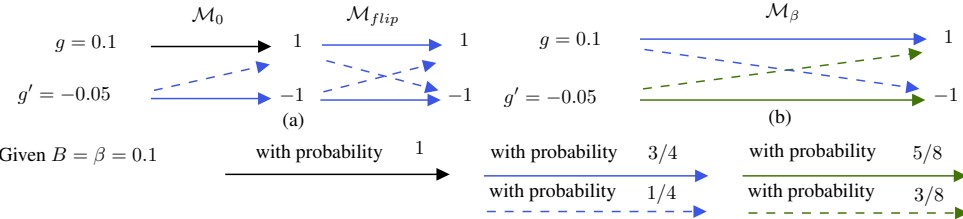

Figure 1: $d = 1$, $B = \beta = 0.1$ and two gradients $g = 0.1$ and $g' = -0.05$.

## 6 CONVERGENCE ANALYSIS

Our analysis is derived under the following technical assumptions that are standard in non-convex optimization Shalev-Shwartz & Ben-David (2014).

**Assumption 1** (Lower bound). *There exists $F^*$ such that $F(w) \geq F^*$ for all $w$.*

**Assumption 2** (Smoothness). *There exists some non-negative constant $L$ such that $F(w_1) \leq F(w_2) + \langle \nabla F(w_2), w_1 - w_2 \rangle + \frac{L}{2}\|w_1 - w_2\|_2^2$ for all $w_1, w_2$.*

**Assumption 3** (Bounded true gradient). *For any coordinate $i \in [d]$, there exists $B_i > 0$ such that $|\nabla f_{mi}(w)| \leq B_i$ for all $m \in [M]$. Let $B_0 := \max_{i \in [d]} B_i$.*

Stochastic gradients can have significantly wider range than the true gradients.

**Example 1** (Norm discrepancy between true and stochastic gradients). *Let $x \sim \mathcal{N}(\mathbf{0}, \mathbf{I}_d)$ be standard Gaussian random vector. Let $f(w, \boldsymbol{x}) = \langle w, \boldsymbol{x} \rangle + \xi$, where $\xi$ is some unknown observation noise. It can be checked easily that for any $w \in \mathbb{R}^d$, it holds that $\nabla f(w) = \nabla \mathbb{E}[f(w, \boldsymbol{x})] = \mathbb{E}[\boldsymbol{x}] = \mathbf{0}$, whereas the natural stochastic gradient $\nabla f(w, \boldsymbol{x}) = \boldsymbol{x}$ whose support is the entire $\mathbb{R}^d$.*

**Assumption 4** (Sub-Gaussianity). *For a given client $m \in [M]$, at any query $w \in \mathbb{R}^d$, the stochastic gradient $\boldsymbol{g}_m(w)$ is an independent unbiased estimate of $\nabla f_m(w)$ that is coordinate-wise related to the gradient $\nabla f_m(w)$ as $\boldsymbol{g}_{mi}(w) = \nabla f_{mi}(w) + \boldsymbol{\xi}_{mi} \, \forall i \in [d]$, where $\boldsymbol{\xi}_{mi}$ is zero-mean $\sigma_{mi}$-sub-Gaussian, i.e, $\mathbb{E}[\boldsymbol{\xi}_{mi}] = 0$, and the two deviation inequalities $\mathbb{P}\{\boldsymbol{\xi}_{mi} \geq t\} \leq \exp\left(-\frac{t^2}{2\sigma_{mi}^2}\right)$ and $\mathbb{P}\{\boldsymbol{\xi}_{mi} \leq -t\} \geq \exp\left(-\frac{t^2}{2\sigma_{mi}^2}\right)$ hold. Let $\sigma^2 := \max_{m \in [M], i \in [d]} \sigma_{mi}^2$.*

**Assumption 5** (Heavy-tailed noise). *Let $\boldsymbol{\xi}_{mi}$ defined in Assumption 4 be a zero-mean random variable, $\mathbb{E}\left[\boldsymbol{\xi}_{mi}^2\right] \leq \sigma^2$, and $\mathbb{E}\left[|\boldsymbol{\xi}_{mi}|^{p'}\right] \leq M_{p'} < \infty$ for $p' \geq 4$.*

Notably, the class of sub-Gaussian random variables contains bounded and unbounded random variable as special cases. Tighter convergence bounds can be obtained under boundedness or Gaussianity assumptions on the noise; see Appendix C.1 for details.

Following the road-map used in Bernstein et al. (2019); Jin et al. (2020); Safaryan & Richtárik (2021), we first establish an upper bound for the probability of gradient sign errors $\mathbb{P}\{\widetilde{\boldsymbol{g}}_i(t) \neq \text{sign}(\nabla_i F(w(t)))\}$, and then bound $\frac{1}{T}\sum_{t=0}^{T-1}\mathbb{E}[\|\nabla F(w(t))\|_1]$ to conclude convergence. Recall that $\boldsymbol{g}_m(t) := \nabla f_m(w(t))$ denotes the true local gradient at client $m$.

**Theorem 3.** *Choose $c_0(n, p) = \max\left\{\sqrt{\frac{8\sigma^2}{n}\log\frac{6}{c}}, \sqrt{\frac{8(B+\beta)^2}{p^2}\log\frac{6}{3-5c}}\right\}$ and $B = (1 + \epsilon_0)B_0$. Fix $t \geq 1$ and $i \in [d]$. Let $c > 0$ be any given constant such that $c < \frac{3}{5}$. Define $\delta_1(t) = \frac{2(B+\beta)}{p}\frac{\tau(t)}{M} + \frac{c_0(n,p)}{\sqrt{M}}$ and $\delta_2(t) = 3(B + \beta)\frac{\tau(t)}{M} + \frac{c_0(n,p)}{\sqrt{M}}$.*

$$\mathbb{P}\{\widetilde{\boldsymbol{g}}_i(t) \neq \text{sign}(\nabla_i F(w(t))) \mid w(t)\} \leq \frac{1 - c}{2}. \tag{4}$$

- *(Sub-Gaussian noise): Suppose Assumption 3 and 4 hold, and $\epsilon_0 > \frac{\sigma}{B_0}$. Eq. (4) holds if $|\nabla_i F(w(t))| \geq \delta_1(t) + 2(B + \beta)\exp\left(-\frac{n}{2}\right)$, when the system adversary is adaptive, or when is static but with $\tau(t) \leq \frac{2}{p^2}\log\frac{6}{c}$, and if $|\nabla_i F(w(t))| \geq \delta_2(t) + 2(B + \beta)\exp\left(-\frac{n}{2}\right)$ when is static with $\tau(t) > \frac{2}{p^2}\log\frac{6}{c}$.*

- *(Heavy-tailed noise): Suppose Assumption 3 and 5 hold, and $\epsilon_0 > \frac{M^{\frac{1}{p'}}}{B_0}$. Eq. (4) holds for $p' \geq 4$ if $|\nabla_i F(w(t))| \geq \delta_1(t) + \frac{4(B+\beta)}{n^{\frac{p'}{2}}}$ when the system adversary is adaptive, or when is static but with $\tau(t) \leq \frac{2}{p^2} \log \frac{6}{c}$, and if $|\nabla_i F(w(t))| \geq \delta_2(t) + \frac{4(B+\beta)}{n^{\frac{p'}{2}}}$ when is static with $\tau(t) > \frac{2}{p^2} \log \frac{6}{c}$.*

Theorem 3 says that when $|\nabla_i F(w(t))|$ is large enough, the sign estimation at the PS in each iteration is more likely to be correct. This is crucial in ensuring the convergence because Theorem 3 implies that when $|\nabla_i F(w(t))|$ is large enough, in expectation, Algorithm 1 pushes $w(t)$ towards a stationary point of the global objective $F$. Additionally, small $|\nabla_i F(w(t))|$ implies that $w(t)$ is already near the neighborhood of a stationary point. Different from Safaryan & Richtárik (2021) and Jin et al. (2020), we neither assume the sign error distributions across clients be identical, nor require the average probability of sign error to be less than $1/2$. Instead, we show that it is enough to let the probability of population sign errors be small when the magnitude of the gradients is large.

**Theorem 4.** *Suppose Assumptions 1, 2, 3 hold. Define $\delta_1(t) = \frac{2(B+\beta)}{p} \frac{\tau(t)}{M} + \frac{c_0(n,p)}{\sqrt{M}}$, $\delta_2(t) = 3(B+\beta)\frac{\tau(t)}{M} + \frac{c_0(n,p)}{\sqrt{M}}$, $\Xi_1(n) = 2(B+\beta)\exp\left(-\frac{n}{2}\right)$, and $\Xi_2(n) = \frac{4(B+\beta)}{n^{\frac{p'}{2}}}$. For any given $t$, $B = (1+\epsilon_0)B_0$, and $c$ such that $0 < c < \frac{3}{5}$, set the learning rate as $\eta = \frac{1}{\sqrt{dT}}$ and $c_0(n,p) := \max\left\{\sqrt{\frac{8\sigma^2}{n}\log\frac{6}{c}}, \sqrt{\frac{8(B+\beta)^2}{p^2}\log\frac{6}{3-5c}}\right\}$.*

*With Assumption 4 for $\epsilon_0 > \frac{\sigma}{B_0}$ or 5 for $\epsilon_0 > \frac{M^{\frac{1}{p'}}}{B_0}$ $(p' \geq 4)$,*

*when the system adversary is adaptive or when is static but with $\tau(t) \leq \frac{2}{p^2} \log \frac{6}{c}$,*

$$\frac{1}{T}\sum_{t=0}^{T-1} \mathbb{E}\left[\|\nabla F(w(t))\|_1\right] \leq \frac{1}{c}\left[\frac{(F(w(0)) - F^*)\sqrt{d}}{\sqrt{T}} + \frac{L\sqrt{d}}{2\sqrt{T}} + 2d\Xi(n) + \frac{2d}{T}\sum_{t=0}^{T-1}\delta_1(t)\right]; \quad (5)$$

*or when the system adversary is static with $\tau(t) > \frac{2}{p^2} \log \frac{6}{c}$,*

$$\frac{1}{T}\sum_{t=0}^{T-1} \mathbb{E}\left[\|\nabla F(w(t))\|_1\right] \leq \frac{1}{c}\left[\frac{(F(w(0)) - F^*)\sqrt{d}}{\sqrt{T}} + \frac{L\sqrt{d}}{2\sqrt{T}} + 2d\Xi(n) + \frac{2d}{T}\sum_{t=0}^{T-1}\delta_2(t)\right]. \quad (6)$$

*We present the convergence results under a unified framework, where $\Xi(n) = \Xi_1(n)$ in the case of sub-Gaussian noise, and $\Xi(n) = \Xi_2(n)$ in the case of heavy-tailed noise.*

**Remark 2.** *(1) The convergence rates in Eq. (5) and Eq. (6) only differ in their Byzantine terms by a multiplicative factor of $\frac{3p}{2}$. As long as $\tau(t)$ is sufficiently large, the impacts of $p$, the degree of partial client participation, on the convergence rate upper bound is limited. The lower bound requirement on $\tau(t)$ might be an artifact of our analysis in simplifying the boundary case derivation.*

*(2) If $\tau(t) = \tau$ for each $t$, then the Byzantine terms in Eq. (5) and Eq. (6) become $\frac{4(B+\beta)\tau d}{pM}$ and $\frac{6(B+\beta)\tau d}{M}$. Now consider the asymptotics in terms of $T$ and the client number $M$ only. If $\tau = \mathcal{O}\left(\sqrt{M}\right)$, then both $\frac{4(B+\beta)\tau d}{pM}$ and $\frac{6(B+\beta)\tau d}{M}$ scale in $M$ with order $\mathcal{O}(\frac{1}{\sqrt{M}})$, which is of the same order as the term $\frac{c_0(n,p)}{\sqrt{M}}$ in Eq. (5) and Eq. (6), the consequences of weak signal strength of the compressed gradients near a stationary point of the global objective $F$. We note that the impacts are limited as the contribution of one client is masked as a one-bit sign instead of an arbitrary value, which is also verified in the following experiments. On the other hand, if $\sum_{t=0}^{T-1}\tau(t) = \mathcal{O}\left(\sqrt{T}\right)$, the Byzantine terms in Eq. (5) and Eq. (6) scale as $O(\frac{1}{\sqrt{T}})$, of the same order as the first two terms. In either case, due to the mobility of the Byzantine faults, it is possible that $\cup_{t=0}^{T}\mathcal{B}(t) = [M]$, i.e., every client is corrupted at least once.*

*(3) The residual term $\Xi(n)$ is an immediate consequence of using mini-batch stochastic gradients rather than true gradient as in Jin et al. (2020). It turns out that this term have minimal impact on*

*the final convergence. In fact, as long as $n = \Omega\left(\log M\right)$ (sub-Gaussian noise) or $n = \Omega\left(M^{\frac{1}{p'}}\right)$ (heavy-tailed noise), this term becomes non-dominating.*

*(4) When $\tau(t) = \tau = \mathcal{O}\left(\sqrt{M}\right)$ and n of the same order as in (3), the convergence rates become $\mathcal{O}(\frac{1}{\sqrt{T}} + \frac{1}{\sqrt{M}})$, approaching the convergence rate of the standard (centralized, non-private, and adversary-free) SGD $\mathcal{O}\left(\frac{1}{\sqrt{T}}\right)$ as $M \to \infty$. Luckily, M in FL is often large McMahan et al. (2017).*

The bounds in Theorem 4 can be tightened with more structured gradient noises. We defer our results on Gaussian-tailed and bounded stochastic gradients to Appendix C.2. All of the results have a similar form and differ only in the noisy residual terms. In detail, the residual term $\mathcal{O}\left(\exp(-n/2)\right)$ in the convergence results of the Gaussian-tailed stochastic gradients is scaled by a constant $1/4\sqrt{2\pi}$, while no noisy tail term appears in the case of bounded stochastic gradients.

## 7 EXPERIMENTAL EVALUATION

In this section, we evaluate the accuracy and convergence speed of our Algorithm 1 in terms of the impacts of client sampling $p$, the differential privacy protection $\beta$, and Byzantine attack resilience. More experiment setups and comparisons with Byzantine baselines are deferred to Appendix. E.We list key elements of our experimental setup for comparisons with benchmark algorithm below.

- **Datasets:** MNIST LeCun et al. (2009), and CIFAR-10 Krizhevsky et al. (2009).
- **Models:** Simple Multi-Layer Perceptron (MLP) McMahan et al. (2017).
- **Clients Data:** 100 balanced workers with $\alpha = 1$ Dirichlet distribution Hsu et al. (2019).
- **Baseline algorithms:** Sign SGD Bernstein et al. (2018) and FedAvg McMahan et al. (2017).
- **Byzantine baselines:** Krum Blanchard et al. (2017), Geometric Median Chen et al. (2017), Centered Clipping Karimireddy et al. (2021). Adversary models: Label flipping, Inner product manipulation Xie et al. (2020), and the "A little is enough" Baruch et al. (2019).

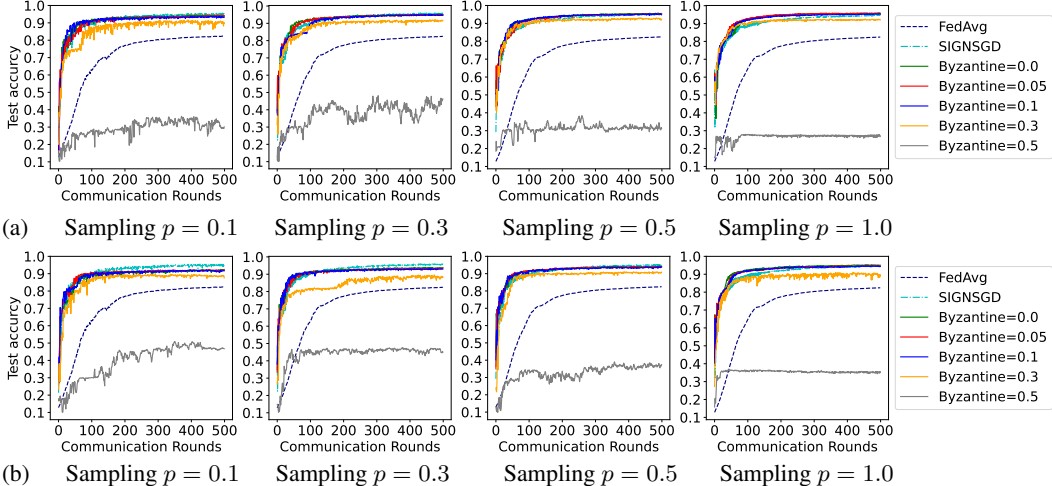

Figure 2: MNIST testing results under non-i.i.d distribution with different Byzantine fractions and (a) no differential privacy protection: $\beta = 0$, (b) differential privacy protection $\beta = B$.

As $\alpha$ decreases, the local datasets become more and more non-IID across different clients. For a fair competition, all the experiments are under the same learning rate and mini-batch size. Notably, we treat $B$ as a hyperparameter and fine-tune from extensive experiments because we allow the stochastic gradients to be unbounded and the true gradient is upper bounded by $B_0$.

| DS / BS | MNIST | CIFAR-10 |
|---|---|---|
| 32 | 89.2% | 46.03% |
| 64 | 88.6% | 46.68% |
| 128 | 89.8% | 46.78% |
| 256 | 91.8% | 46.54% |

Table 1: Testing results on two datasets (DS) with different mini-batch sizes (BS).

We train MLP under full client participation with $80$ and $300$ communication rounds in the first two comparisons for MNIST and CIFAR-10, respectively. For the client sampling and Byzantine resilience, we extend the time horizon to $500$ communication rounds for both datasets.

**Mini-batch size.** We compare the peak performances of our variant under different mini-batch sizes. It is observed in Table 1 that the Algorithm 1 is not sensitive to mini-batch size $n$. This meets Remark 2 (3) as $n = \Omega \left( \log M \right)$ or $n = \Omega \left( M^{\frac{1}{p'}} \right)$ is enough for the term not to become dominant.

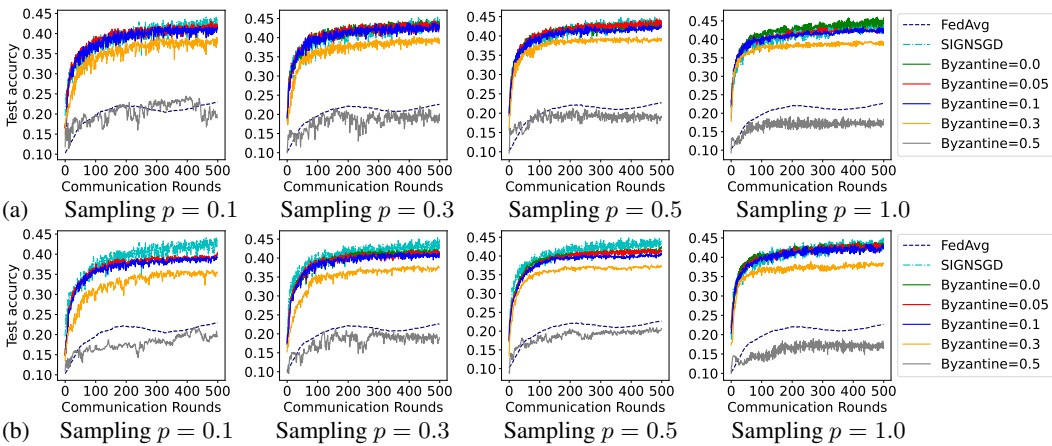

(a)  Sampling $p = 0.1$     Sampling $p = 0.3$     Sampling $p = 0.5$     Sampling $p = 1.0$

(b)  Sampling $p = 0.1$     Sampling $p = 0.3$     Sampling $p = 0.5$     Sampling $p = 1.0$

Figure 3: CIFAR-10 testing results under non-i.i.d distribution with different Byzantine fractions and (a) no differential privacy protection: $\beta = 0$, (b) differential privacy protection $\beta = B$.

**Parameter $B$ and $\beta$.** The peak performances are collected and listed in Table 2. We can observe that the testing results drop as $\beta$ increases. This matches a trivial observation from Theorem 4 such that the increase of $\beta$ pushes the bound farther away from stationary point, implying our variant suffers from the data-utility-privacy trade-off. For any given $B$, the testing accuracy is comparable when the differential privacy quantification $\beta$s are relatively small such as $\beta \in \{0.1B, B\}$, which is fortunate as we might not sacrifice too much data utility while ensuring a privacy quantification.

| $\beta/B$ | MNIST | | | |
|---|---|---|---|---|
| $B$ | $0.1$ ($\epsilon = 3.04d$) | $1.0$ ($\epsilon = 1.1d$) | $5.0$ ($\epsilon = 0.34d$) | $10.0$ ($\epsilon = 0.18d$) |
| $1.0$ | 64.0% | 61.7% | 41.0% | 16.2% |
| $0.1$ | 83.1% | 82.9% | 81.4% | 41.2% |
| $0.01$ | 89.9% | 90.0% | 90.3% | 76.7% |
| $\beta/B$ | CIFAR-10 | | | |
| $B$ | $0.1$ ($\epsilon = 3.04d$) | $1.0$ ($\epsilon = 1.1d$) | $5.0$ ($\epsilon = 0.34d$) | $10.0$ ($\epsilon = 0.18d$) |
| $1.0$ | 23.6% | 20.7% | 17.0% | 16.7% |
| $0.1$ | 35.1% | 32.4% | 25.3% | 23.2% |
| $0.01$ | 45.7% | 43.7% | 40.7% | 38.0% |

Table 2: Testing results on two datasets with different combinations of $B$ and $\beta/B$ ($\epsilon$).

**Client sampling and Byzantine resilience.** Besides normal clients, we assume a constant fraction of the reporting clients are Byzantine and only send the negation of the reporting signs to PS, the adversary which we call adaptive mobile attackers. Though this is a constant fraction Byzantine experiment, recall that our algorithm can tolerate up to $\mathcal{O}\left(\sqrt{M}\right)$ constant or up to $\sum_{t=0}^{T-1} \tau(t) = \mathcal{O}\left(\sqrt{T}\right)$ time-varying Byzantine adversaries. It is observed in Fig. 2 and Fig. 3 that the Algorithm 1 is not sensitive to client sampling. In particular, the Byzantine-free peak accuracy reaches around 96% on MNIST and around 44% on CIFAR-10, which is a direct consequence of highly heterogeneous data distributions. The accuracy drops are almost negligible when the Byzantine fraction does not exceed 0.1. We can see a sharper drop as the fraction increases. However, the testing accuracy remains stable during the final training stage when Byzantine clients account for 50% of the reporting clients except when the sampling rate $p$ is small. Two baseline algorithms with no Byzantine clients are also evaluated for comparisons. It is observed that SignSGD slightly outperforms our variant in some cases with $\beta > 0$; however, our variant is provable $d \cdot \log 3$-differentially private. FedAvg is inferior to the other algorithms in all cases.

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

# A COMPARISONS WITH JIN ET AL. (2020).

| | | Jin et al. (2020) | Our work |
|---|---|---|---|
| Gradient | Batch-size | Full-batch (True Gradient) [Eq. (33)] | Mini-batch Stochastic Gradient |
| | Distribution | Bounded [Theorem 3] | Sub-Gaussian (bounded as a subset) and heavy-tailed |
| Residual | Weak Signal Strength | $\Delta(M)$ as a function of client number $M$ without explicit form [Theorem 2 and Remark 3]. | $\frac{c_0(n,p)}{\sqrt{M}}$ |
| | Byzantine | Implicit forms without quantitative results [Theorem 7 and Remark 6]. | $\frac{(B+\beta)\sum_{t=0}^{T-1}\tau(t)}{TM}$ |
| | Gradient noise | No, since only bounded gradients are considered. | $\mathcal{O}\left(\exp\left(-\frac{n}{2}\right)\right)$ for Sub-Gaussian noise; $\mathcal{O}\left(1/n^{\frac{p'}{2}}\right)$ $(p' \geq 4)$ for heavy-tailed noise. |
| Differential Privacy | | No for sto-sign compressor; flawed arguments for DP-sign compressor. | $d\log(1 + \frac{2B}{\beta})$ for arbitrary gradients; $\left(\frac{\Delta_1}{\beta}\right)$ for gradient pairs with bounded $l_1$ sensitivity $\Delta_1$. |
| Partial Client Participation | | No. | Theoretically and empirically verified, and build adaptive Byzantine adversaries on it. |

Table 3: Point-by-point comparisons with Jin et al. (2020).

# B ALGORITHMS

**Definition 6.** *Let $\mathcal{S} \subseteq [M]$ and let $\widehat{u}_m \in \{\pm 1\}^d$ for $m \in \mathcal{S}$.*
*(A.1) The mean aggregation rule is defined as $\mathrm{agg}_{avg}\left(\{\widehat{u}_m, \ m \in \mathcal{S}\}\right) = \frac{1}{|\mathcal{S}|}\sum_{m \in \mathcal{S}}\widehat{u}_m$.*
*(A.2) The coordinate-wise $k$-trimmed-mean aggregation rule, denoted by $\mathrm{agg}_{trimmed,k}$, takes $k \in \mathbb{N}$ and $\{\widehat{u}_m, \ m \in \mathcal{S}\}$ as inputs and aggregates each coordinate $i \in [d]$ as follows: (1) sort $\{\widehat{u}_{mi}, \ m \in \mathcal{S}\}$, where $\widehat{u}_{mi}$ is the $i$-th coordinate of $\widehat{u}_m$, in an increasing order; (2) remove the top and bottom $k$ values, and denote the remained clients w. r. t. coordinate $i$ as $\mathcal{R}_i$; (3) if $\mathcal{R}_i = \emptyset$, then $\mathrm{agg}_{trimmed,k,i}\left(\{\widehat{u}_m, \ m \in \mathcal{S}\}\right) = 1$ (or $-1$) uniformly at random; otherwise, $\mathrm{agg}_{trimmed,k,i}\left(\{\widehat{u}_m, \ m \in \mathcal{S}\}\right) = \frac{1}{|\mathcal{R}_i|}\sum_{m \in \mathcal{R}_i}\widehat{u}_m$.*
*(A.3) The coordinate-wise median aggregation rule, denoted by $\mathrm{agg}_{median}$, aggregates each coordinate $i \in [d]$ as follows: it first sorts the $\{\widehat{u}_{mi}, \ m \in \mathcal{S}\}$ in an increasing order. If $|\mathcal{S}|$ is even, $\mathrm{agg}_{median,i}\left(\{\widehat{u}_m, \ m \in \mathcal{S}\}\right)$ outputs the average of the elements whose ranks are $\frac{|\mathcal{S}|}{2}$ and $\frac{|\mathcal{S}|}{2} + 1$. Otherwise, $\mathrm{agg}_{median,i}\left(\{\widehat{u}_m, \ m \in \mathcal{S}\}\right)$ outputs the element whose rank is $\lceil\frac{|\mathcal{S}|}{2}\rceil$.*
*(A.4) The coordinate-wise majority vote aggregation rule, denoted by $\mathrm{agg}_{maj}$, aggregates each coordinate $i \in [d]$ as follows: If there are more $1$ than $-1$ in $\{\widehat{u}_{mi}, \ m \in \mathcal{S}\}$, then $\mathrm{agg}_{maj,i}\left(\{\widehat{u}_m, \ m \in \mathcal{S}\}\right)$ outputs $1$. If there are more $-1$ than $1$ in $\{\widehat{u}_{mi}, \ m \in \mathcal{S}\}$, then $\mathrm{agg}_{maj,i}\left(\{\widehat{u}_m, \ m \in \mathcal{S}\}\right)$ outputs $-1$. Otherwise, $\mathrm{agg}_{median,i}\left(\{\widehat{u}_m, \ m \in \mathcal{S}\}\right)$ outputs $0$.*

# C ALTERNATIVE RESULTS

## C.1 ALTERNATIVE ASSUMPTIONS

The following two alternative assumptions on the randomness of stochastic gradients are of decreasing levels of stringency.

**Assumption 6** (Boundedness). *The $\ell_\infty$ norm of all possible stochastic gradients is upper bounded. Formally, let $m \in [M]$ be an arbitrary client and $\boldsymbol{g}$ be an arbitrary stochastic gradient that client $m$ obtains. For any coordinate $i \in [d]$, there exists $\widetilde{B}_i > 0$ such that $|g_i| \leq \widetilde{B}_i$. Let $\widetilde{B} = \max_{i \in [d]}\widetilde{B}_i$.*

The following alternative assumption relaxes the boundedness requirement, and allows the stochastic gradients to be supported over the entire $\mathbb{R}^d$.

**Assumption 7** (Gaussianity). *For a given client $m \in [M]$, at any query $w \in \mathbb{R}^d$, the stochastic gradient $\boldsymbol{g}_m(w)$ is an independent unbiased estimate of $\nabla f_m(w)$ that is coordinate-wise related to the gradient $\nabla f_{mi}(w)$ as $\boldsymbol{g}_{mi}(w) = \nabla f_{mi}(w) + \boldsymbol{\xi}_{mi} \; \forall i \in [d]$,*

*where $\boldsymbol{\xi}_{mi} \sim \mathcal{N}\left(0, \sigma_{mi}^2\right)$. Let $\sigma^2 := \max_{m \in [M], i \in [d]} \sigma_{mi}^2$.*

## C.2 Alternative Convergence Rates

**Corollary 2.** *Suppose that Assumptions 3 and 7 hold. Choose $B = (1 + \epsilon_0)B_0$ for $\epsilon_0 > \sigma/B_0$ and $c_0(n, p) = \max\left\{\sqrt{\frac{8\sigma^2}{n}\log\frac{6}{c}}, \sqrt{\frac{8(B+\beta)^2}{p^2}\log\frac{6}{3-5c}}\right\}$. Fix $t \geq 1$ and $i \in [d]$. Let $c > 0$ be any given constant such that $c < \frac{3}{5}$.*

*When the system adversary is adaptive or when the system adversary is static but with $\tau(t) \leq \frac{2}{p^2}\log\frac{6}{c}$, if $|\nabla_i F(w(t))| \geq \frac{2(B+\beta)}{pM}\tau(t) + \frac{B+\beta}{2\sqrt{2\pi}}\exp\left(-\frac{n}{2}\right) + \frac{c_0(n,p)}{\sqrt{M}}$, then Eq. (4) holds.*

*When the system adversary is static with $\tau(t) > \frac{2}{p^2}\log\frac{6}{c}$, if $|\nabla_i F(w(t))| \geq \frac{3(B+\beta)\tau(t)}{M} + \frac{B+\beta}{2\sqrt{2\pi}}\exp\left(-\frac{n}{2}\right) + \frac{c_0(n,p)}{\sqrt{M}}$, then Eq. (4) holds.*

**Corollary 3.** *Suppose Assumptions 1, 2, 3, and 7 hold. For any given $T$, $B = (1 + \epsilon_0)B_0$ for $\epsilon_0 > \frac{\sigma}{B_0}$, and $c$ such that $0 < c < \frac{3}{5}$, set the learning rate as $\eta = \frac{1}{\sqrt{dT}}$ and $c_0(n, p) = \max\left\{\sqrt{\frac{8\sigma^2}{n}\log\frac{6}{c}}, \sqrt{\frac{8(B+\beta)^2}{p^2}\log\frac{6}{3-5c}}\right\}$.*

*When the system adversary is adaptive or when the system adversary is static but with $\tau(t) \leq \frac{2}{p^2}\log\frac{6}{c}$, we have*

$$\frac{1}{T}\sum_{t=0}^{T-1}\mathbb{E}\left[\|\nabla F(w(t))\|_1\right] \leq \frac{1}{c}\left[\frac{(F(w(0)) - F^*)\sqrt{d}}{\sqrt{T}} + \frac{L\sqrt{d}}{2\sqrt{T}} + \frac{d}{\sqrt{2\pi}}(B+\beta)\exp\left(-\frac{n}{2}\right)\right.$$
$$\left. + 2d\frac{c_0(n,p)}{\sqrt{M}} + 4d\frac{(B+\beta)\sum_{t=0}^{T-1}\tau(t)}{pTM}\right].$$

*On the other hand, when the system adversary is static with $\tau(t) > \frac{2}{p^2}\log\frac{6}{c}$, we have*

$$\frac{1}{T}\sum_{t=0}^{T-1}\mathbb{E}\left[\|\nabla F(w(t))\|_1\right] \leq \frac{1}{c}\left[\frac{(F(w(0)) - F^*)\sqrt{d}}{\sqrt{T}} + \frac{L\sqrt{d}}{2\sqrt{T}} + \frac{d}{\sqrt{2\pi}}(B+\beta)\exp\left(-\frac{n}{2}\right)\right.$$
$$\left. + 2d\frac{c_0(n,p)}{\sqrt{M}} + 6d\frac{(B+\beta)\sum_{t=0}^{T-1}\tau(t)}{TM}\right].$$

**Corollary 4.** *Suppose that Assumption 6 holds. Choose $B = \widetilde{B}$ and $c_0(n, p) = \max\left\{\sqrt{\frac{8\sigma^2}{n}\log\frac{6}{c}}, \sqrt{\frac{8(B+\beta)^2}{p^2}\log\frac{6}{3-5c}}\right\}$. Fix $t \geq 1$ and $i \in [d]$. Let $c > 0$ be any given constant such that $c < \frac{3}{5}$.*

*When the system adversary is adaptive or when the system adversary is static but with $\tau(t) \leq \frac{2}{p^2}\log\frac{6}{c}$, if $|\nabla_i F(w(t))| \geq \frac{2(B+\beta)}{pM}\tau(t) + \frac{c_0(n,p)}{\sqrt{M}}$, then Eq. (4) holds.*

*When the system adversary is static with $\tau(t) > \frac{2}{p^2}\log\frac{6}{c}$, if $|\nabla_i F(w(t))| \geq \frac{3(B+\beta)}{M}\tau(t) + \frac{B+\beta}{2\sqrt{2\pi}}\exp\left(-\frac{n}{2}\right) + \frac{c_0(n,p)}{\sqrt{M}}$, then Eq. (4) holds.*

**Corollary 5.** *Suppose Assumptions 1, 2, and 6 hold. For any given $T$ and $c$ such that $0 < c < \frac{3}{5}$, set the learning rate as $\eta = \frac{1}{\sqrt{dT}}$ and $c_0(n, p) := \max\left\{\sqrt{\frac{8\sigma^2}{n}\log\frac{6}{c}}, \sqrt{\frac{8(B+\beta)^2}{p^2}\log\frac{6}{3-5c}}\right\}$.*

*When the system adversary is adaptive or when the system adversary is static but with $\tau(t) \leq \frac{2}{p^2} \log \frac{6}{c}$, we have*

$$\frac{1}{T} \sum_{t=0}^{T-1} \mathbb{E}\left[\|\nabla F(w(t))\|_1\right] \leq \frac{1}{c} \left[ \frac{(F(w(0)) - F^*)\sqrt{d}}{\sqrt{T}} + \frac{L\sqrt{d}}{2\sqrt{T}} + 2d\frac{c_0(n,p)}{\sqrt{M}} + 4d\frac{(B+\beta)\sum_{t=0}^{T-1} \tau(t)}{pTM} \right].$$

*On the other hand, when the system adversary is static with $\tau(t) > \frac{2}{p^2} \log \frac{6}{c}$, we have*

$$\frac{1}{T} \sum_{t=0}^{T-1} \mathbb{E}\left[\|\nabla F(w(t))\|_1\right] \leq \frac{1}{c} \left[ \frac{(F(w(0)) - F^*)\sqrt{d}}{\sqrt{T}} + \frac{L\sqrt{d}}{2\sqrt{T}} + 2d\frac{c_0(n,p)}{\sqrt{M}} + 6d\frac{(B+\beta)\sum_{t=0}^{T-1} \tau(t)}{TM} \right].$$

# D  PROOFS

## D.1  AGGREGATION FUNCTIONS

*Proof.* **[Proof of Proposition 1 (Equivalent to Majority Vote)]** The intuition behind this proof is to show that the signs of all the aggregation rules mentioned in the theorem statement, given $\widehat{u}_m \in \{\pm1\}^d$ for $m \in \mathcal{S}$, are equivalent to the sign of the $k$-trimmed-mean aggregation rule.

We first show that for any $k < |\mathcal{S}|/2$, the signs of the outputs of the signs of the aggregation rule $\mathsf{agg}_{\text{trimmed},k}$ are the same. When $k < |\mathcal{S}|/2$, it holds that $\mathcal{R}_i \neq \emptyset$ for each $i \in [d]$. Thus, the aggregation rules $\mathsf{agg}_{\text{trimmed},k}$ with $k < |\mathcal{S}|/2$ is deterministic.

For any given coordinate $i \in [d]$, if the sign of $\mathsf{agg}_{\text{trimmed},k}$ is 0, by definition, we know that there are equal numbers of 1 and $-1$ in $\{\widehat{u}_{mi} : m \in \mathcal{R}_i\}$, and that the top (resp. bottom) $k$ elements removed from $\{\widehat{u}_{mi} : m \in \mathcal{S}\}$ are 1 (resp. $-1$). That is, there are equal numbers of 1 and $-1$ in $\{\widehat{u}_{mi} : m \in \mathcal{S}\}$. Hence, for any $k' \neq k$, as long as the remained set $\mathcal{R}'_i$ after trimming is nonempty (which is ensured by the condition that $k' < |\mathcal{S}|/2$), it holds that $\mathsf{agg}_{\text{trimmed},k'}(\{\widehat{u}_{mi} : m \in \mathcal{S}\}) = 0$.

If the sign of $\mathsf{agg}_{\text{trimmed},k}$ is $-1$, we know that there are more $-1$ than 1 in $\{\widehat{u}_{mi} : m \in \mathcal{R}_i\}$, and that the bottom $k$ elements in $\{\widehat{u}_{mi} : m \in \mathcal{S}\}$ are all $-1$ whereas the number of 1 in the top $k$ elements is at most $k$. That is, there are more $-1$ than 1 in $\{\widehat{u}_{mi} : m \in \mathcal{S}\}$. Hence, we know that for any $k'$, as long as the remained set $\mathcal{R}'_i$ after trimming is nonempty, the sign of $\mathsf{agg}_{\text{trimmed},k'}(\{\widehat{u}_{mi} : m \in \mathcal{S}\})$ is $-1$. Similarly, we can show the case when the sign of $\mathsf{agg}_{\text{trimmed},k}$ is 1.

The above argument, combined with the definition of $\mathsf{agg}_{\text{maj}}$, immediately implies that when $k < |\mathcal{S}|/2$, the signs of $\mathsf{agg}_{\text{trimmed},k}$ and $\mathsf{agg}_{\text{maj}}$ are the same.

Finally, since $\mathsf{agg}_{\text{avg}}$ is $\mathsf{agg}_{\text{trimmed},0}$ and $\mathsf{agg}_{\text{median}} = \mathsf{agg}_{\text{trimmed}, \lfloor \frac{|\mathcal{S}|-1}{2} \rfloor}$, the signs of $\mathsf{agg}_{\text{avg}}$, $\mathsf{agg}_{\text{median}}$, and $\mathsf{agg}_{\text{trimmed},k}$ for $k < |\mathcal{S}|/2$ are all the same, proving the theorem.

$\square$

## D.2  PRIVACY PRESERVATION

**Theorem 5.** *(Dwork et al., 2014, Corollary 3.15) Let $\mathcal{M}_i : \mathbb{R}^d \to \{\pm1\}^d$ be an $\epsilon_i$-differentially private algorithm for $i \in [k]$. Then $\mathcal{M}_{[k]}(x) := (\mathcal{M}_1(x), \cdots, \mathcal{M}_k(x))$ is $\sum_{i=1}^{k} \epsilon_i$-differentially private.*

**Proof of Theorem 1 (Necessity of $\beta$).** We first consider the setting when $\beta = 0$. Let

$$\mathcal{G} = \{g \in \mathbb{R}^d : \exists i \ s.t. \min\{|g_i - B|, |g_i + B|\} \leq 1\}.$$

Let $g \in \mathcal{G}$. Without loss of generality, let us assume that $|g_1 - B| \leq 1$, where $g_1$ is the first entry of $g$. If $g_1 \geq B$, then there exists $g' \in \mathbb{R}^d$ such that $g' \neq g$, $g'_1 \in (-B, B)$, and $\|g - g'\|_1 \leq 1$. Let $\widehat{g}_1$ and $\widehat{g'}_1$ be the compressed values of $g_1$ and $g'_1$ under our compressor in Eq. (2). It holds that

$$\frac{\mathbb{P}\left\{\widehat{g'}_1 = -1\right\}}{\mathbb{P}\left\{\widehat{g}_1 = -1\right\}} = \frac{\frac{B - \mathsf{clip}\{g'_1, B\}}{2B}}{\frac{B - \mathsf{clip}\{g_1, B\}}{2B}} = \frac{B - \mathsf{clip}\{g'_1, B\}}{B - \mathsf{clip}\{g_1, B\}} = \frac{B - \mathsf{clip}\{g'_1, B\}}{B - B} = \infty.$$

If $\boldsymbol{g}_1 \in (-B, B)$, then there exists $\boldsymbol{g}' \in \mathbb{R}^d$ such that $\boldsymbol{g}' \neq \boldsymbol{g}$, $\boldsymbol{g}'_1 \geq B$, and $\|\boldsymbol{g} - \boldsymbol{g}'\|_1 \leq 1$. We have

$$\frac{\mathbb{P}\{\widehat{\boldsymbol{g}_1} = -1\}}{\mathbb{P}\{\widehat{\boldsymbol{g}'_1} = -1\}} = \frac{\frac{B - \mathsf{clip}\{\boldsymbol{g}_1, B\}}{2B}}{\frac{B - \mathsf{clip}\{\boldsymbol{g}'_1, B\}}{2B}} = \frac{B - \mathsf{clip}\{\boldsymbol{g}_1, B\}}{B - \mathsf{clip}\{\boldsymbol{g}'_1, B\}} = \frac{B - \mathsf{clip}\{\boldsymbol{g}_1, B\}}{B - B} = \infty.$$

Since a finite differential privacy quantification does not hold for any pair of gradients $\boldsymbol{g}$ and $\boldsymbol{g}'$, no differential privacy implies as per Definition 1, proving the first part of the theorem.

When $\beta > 0$, for any $\boldsymbol{g}, \boldsymbol{g}' \in \mathbb{R}^d$ such that $\boldsymbol{g}' \neq \boldsymbol{g}$ and $\|\boldsymbol{g} - \boldsymbol{g}'\|_1 \leq 1$, and for each coordinate $i \in [d]$, it holds that

$$\frac{\mathbb{P}\{\widehat{\boldsymbol{g}'_i} = -1\}}{\mathbb{P}\{\widehat{\boldsymbol{g}_i} = -1\}} = \frac{\frac{B + \beta - \mathsf{clip}\{\boldsymbol{g}'_1, B\}}{2B + 2\beta}}{\frac{B + \beta - \mathsf{clip}\{\boldsymbol{g}_1, B\}}{2B + 2\beta}} = \frac{B + \beta - \mathsf{clip}\{\boldsymbol{g}'_1, B\}}{B + \beta - \mathsf{clip}\{\boldsymbol{g}_1, B\}} \leq \frac{2B + \beta}{\beta}.$$

Similarly, we can show the same upper bound for $\mathbb{P}\{\widehat{\boldsymbol{g}'_i} = 1\} / \mathbb{P}\{\widehat{\boldsymbol{g}_i} = 1\}$. That is, for the $i$-th coordinate, the compressor $\mathcal{M}_{B,\beta}$ is coordinate-wise $\log\left(\frac{2B+\beta}{\beta}\right)$- differentially private. By Theorem 5, we conclude that the compressor $\mathcal{M}_{B,\beta}$ is $d \cdot \log\left(\frac{2B+\beta}{\beta}\right)$- differentially private for the entire gradient. $\qquad\square$

**Proof of Theorem 2 (Smaller Collection of Gradients).** For each coordinate $i \in [d]$, it holds that

$$\begin{aligned}
\frac{\mathbb{P}\{\widehat{\boldsymbol{g}'_i} = -1\}}{\mathbb{P}\{\widehat{\boldsymbol{g}_i} = -1\}} &= \frac{\frac{B + \beta - \mathsf{clip}\{\boldsymbol{g}'_i, B\}}{2B + 2\beta}}{\frac{B + \beta - \mathsf{clip}\{\boldsymbol{g}_i, B\}}{2B + 2\beta}} = \frac{B + \beta - \mathsf{clip}\{\boldsymbol{g}'_i, B\}}{B + \beta - \mathsf{clip}\{\boldsymbol{g}_i, B\}} \\
&= \frac{B + \beta - \mathsf{clip}\{\boldsymbol{g}_i, B\} + \mathsf{clip}\{\boldsymbol{g}_i, B\} - \mathsf{clip}\{\boldsymbol{g}'_i, B\}}{B + \beta - \mathsf{clip}\{\boldsymbol{g}_i, B\}} \\
&\leq 1 + \frac{|\boldsymbol{g}_i - \boldsymbol{g}'_i|}{B + \beta - \mathsf{clip}\{\boldsymbol{g}_i, B\}} \qquad\qquad (7) \\
&\leq 1 + \frac{\Delta_1}{B + \beta - \mathsf{clip}\{\boldsymbol{g}_i, B\}} \\
&\leq 1 + \frac{\Delta_1}{\beta + \mathsf{dist}(\boldsymbol{g}_i, \mathcal{C}_B)}.
\end{aligned}$$

By Theorem 5, we conclude that the compressor $\mathcal{M}_{B,\beta}$ is $\max_{\boldsymbol{g}\in\mathcal{G}} \sum_{i=1}^{d} \log\left(1 + \frac{\Delta_1}{\beta + \mathsf{dist}(\boldsymbol{g}_i, \mathcal{C}_B)}\right)$- differentially private for all gradients $\boldsymbol{g} \in \mathcal{G}$. $\qquad\square$

**Proof of Corollary 1 (Bounded DP with Bounded Sensitivity).** By Theorem 2, we conclude that the compressor $\mathcal{M}_{B,\beta}$ is $\max_{\boldsymbol{g}\in\mathcal{G}} \sum_{i=1}^{d} \log\left(1 + \frac{\Delta_1}{\beta + \mathsf{dist}(\boldsymbol{g}_i, \mathcal{C}_B)}\right)$- differentially private for all gradients $\boldsymbol{g} \in \mathcal{G}$. It turns out that this bound can be relaxed, and we start the derivation from Eq. (7):

$$(7) \leq 1 + \frac{|\boldsymbol{g}_i - \boldsymbol{g}'_i|}{\beta}.$$

Now consider the coordinate collection of the gradient pair, by Theorem 5, it remains to bound

$$\begin{aligned}
\sum_{i=1}^{d} \log\left(1 + \frac{|\boldsymbol{g}_i - \boldsymbol{g}'_i|}{\beta}\right) &\leq d \log\left[\frac{1}{d}\sum_{i=1}^{d}\left(1 + \frac{|\boldsymbol{g}_i - \boldsymbol{g}'_i|}{\beta}\right)\right] \quad \text{[Jensen's inequality]} \\
&\leq d \log\left(1 + \frac{\Delta_1}{d\beta}\right) \\
&\leq \frac{\Delta_1}{\beta} \quad \text{[follows from } \log(1 + x) < x \text{ when } x > 0.]
\end{aligned}$$

$\qquad\square$

**Proof of Proposition 2 (Equivalent as a Composition).** Let $\boldsymbol{g} \in \mathbb{R}^d$ be an arbitrary gradient. To show this proposition, it is enough to show $\mathbb{P}\left\{[\mathcal{M}_{B,\beta}]_i(\boldsymbol{g}) = 1\right\} = \mathbb{P}\left\{[\mathcal{M}_{B,\text{flip}} \circ \mathcal{M}_{B,0}]_i(\boldsymbol{g}) = 1\right\}$ holds for any $i \in [d]$.

To see this,

$$
\begin{aligned}
\mathbb{P}\left\{[\mathcal{M}_{B,\text{flip}} \circ \mathcal{M}_{B,0}]_i(\boldsymbol{g}) = 1\right\} &= \mathbb{P}\left\{[\mathcal{M}_{B,0}]_i(\boldsymbol{g}) = 1 \,\&\, \mathcal{M}_{B,\text{flip}}(1) = 1\right\} \\
&\quad + \mathbb{P}\left\{[\mathcal{M}_{B,0}]_i(\boldsymbol{g}) = -1 \,\&\, \mathcal{M}_{B,\text{flip}}(-1) = -1\right\} \\
&= \frac{B + \text{clip}\{\boldsymbol{g}_i, B\}}{2B} \frac{2B + \beta}{2(B + \beta)} + \frac{B - \text{clip}\{\boldsymbol{g}_i, B\}}{2B} \frac{\beta}{2(B + \beta)} \\
&= \frac{B + \beta + \text{clip}\{\boldsymbol{g}_i, B\}}{2(B + \beta)} \\
&= \mathbb{P}\left\{[\mathcal{M}_{B,\beta}]_i(\boldsymbol{g}) = 1\right\}.
\end{aligned}
$$

$\square$

## D.3 CONVERGENCE RESULTS

**Proposition 3** (Bounded Random Variable Variance Bound). *Given a random variable $X$ and a clipping threshold $B > 0$, if $\mu = \mathbb{E}[X] \in [-B, B]$, then $\text{var}(\text{clip}(X, B)) \leq \text{var}(X) = \sigma^2$.*

**Proof of Proposition 3.**

$$
\begin{aligned}
\text{var}(\text{clip}(X, B)) :=& \mathbb{E}\left[(\text{clip}(X, B) - \mathbb{E}[\text{clip}(X, B)])^2\right] \\
=& \mathbb{E}\left[(\text{clip}(X, B) - \mathbb{E}[X])^2\right] - (\mathbb{E}[\text{clip}(X, B) - X])^2 \\
\leq& \mathbb{E}\left[(\text{clip}(X, B) - \mathbb{E}[X])^2\right].
\end{aligned} \tag{8}
$$

For ease of exposition, we assume $X$ admits a probability density function $f(x)$. General distributions of $X$ can be shown analogously. It follows that

$$
\begin{aligned}
&\mathbb{E}\left[(\text{clip}(X, B) - \mathbb{E}[X])^2\right] \\
&= \int_B^\infty (B - \mu)^2 f(x)\mathrm{d}x + \int_{-B}^B (x - \mu)^2 f(x)\mathrm{d}x + \int_{-\infty}^{-B} (-B - \mu)^2 f(x)\mathrm{d}x \\
&\leq \int_B^\infty (x - \mu)^2 f(x)\mathrm{d}x + \int_{-B}^B (x - \mu)^2 f(x)\mathrm{d}x + \int_{-\infty}^{-B} (x - \mu)^2 f(x)\mathrm{d}x \\
&= \text{var}(X) = \sigma^2.
\end{aligned} \tag{9}
$$

Combining (8) and (9), we conclude $\text{var}(\text{clip}(X, B)) \leq \text{var}(X) = \sigma^2$. $\square$

### D.3.1 SUB-GAUSSIAN AND HEAVY-TAILED DISTRIBUTIONS

**Proof of Theorem 3 (Light and Heavy-tailed Sign Error).** Recall that

$$
\widehat{\boldsymbol{g}}_{mi}(t) = \begin{cases} [\mathcal{M}_{B,\beta}]_i\left(\frac{1}{n}\sum_{j=1}^n \boldsymbol{g}_{mi}^j(t)\right) & \text{if } m \in \mathcal{N}(t); \\ * & \text{if } m \in \mathcal{B}(t), \end{cases}
$$

where $*$ is an arbitrary value in $\{-1, 1\}$. For any client $m \in [M]$ and any coordinate $i \in [d]$, let

$$
X_{mi} = \mathbf{1}_{\{m \in \mathcal{S}(t)\}}\mathbf{1}_{\left\{\widehat{\boldsymbol{g}}_{mi} \neq \text{sign}\left(\frac{1}{M}\sum_{m=1}^M \boldsymbol{g}_{mi}\right)\right\}},
$$

and $\quad \widetilde{X}_{mi} = \mathbf{1}_{\{m \in \mathcal{S}(t)\}}\mathbf{1}_{\left\{[\mathcal{M}_\beta]_i\left(\frac{1}{n}\sum_{j=1}^n \boldsymbol{g}_{mi}^j(t)\right) \neq \text{sign}\left(\frac{1}{M}\sum_{m=1}^M \boldsymbol{g}_{mi}\right)\right\}}.$

Notably, if $m \in \mathcal{B}(t)$, then it is possible that $X_{mi} \neq \widetilde{X}_{mi}$; otherwise, $X_{mi} = \widetilde{X}_{mi}$.

Without loss of generality, we assume the true aggregation is negative, i.e., $\text{sign}(\nabla_i F(w(t))) = -1$. The case when $\text{sign}(\nabla_i F(w(t))) = 1$ can be shown analogously.

For ease of exposition, we drop a condition of $w(t)$ in the conditional probability expressions unless otherwise noted. It holds that

$$
\mathbb{P}\left\{\text{sign}\left(\frac{1}{M}\sum_{m=1}^{M}\widehat{\boldsymbol{g}}_{mi}\right)\neq-1\right\}\leq\mathbb{P}\left\{\sum_{m=1}^{M}X_{mi}\geq\frac{|\mathcal{S}(t)|}{2}\right\}
$$

$$
=\mathbb{P}\left\{\sum_{m\in\mathcal{N}(t)}\widetilde{X}_{mi}+\sum_{m\in\mathcal{B}(t)}X_{mi}\geq\frac{|\mathcal{S}(t)|}{2}\right\}
$$

$$
=\mathbb{P}\left\{\sum_{m\in\mathcal{N}(t)}\widetilde{X}_{mi}\geq\frac{|\mathcal{S}(t)|}{2}-\sum_{m\in\mathcal{B}(t)}X_{mi}\right\}
$$

$$
\leq\mathbb{P}\left\{\sum_{m=1}^{M}\widetilde{X}_{mi}\geq\frac{|\mathcal{S}(t)|}{2}-\sum_{m\in\mathcal{B}(t)}X_{mi}\right\}. \tag{10}
$$

Next, we bound $\sum_{m=1}^{M}\widetilde{X}_{mi}$ and $\sum_{m\in\mathcal{B}(t)}X_{mi}$ separately.

When the system adversary is static, i.e., the system adversary does not know $\mathcal{S}(t)$, it corrupts clients independently of $\mathcal{S}(t)$. Hence,

$$
\sum_{m\in\mathcal{B}(t)}X_{mi}\leq\sum_{m\in\mathcal{B}(t)}\mathbf{1}_{\{m\in\mathcal{S}(t)\}}. \tag{11}
$$

We know that if $\tau(t)\leq\frac{2}{p^2}\log\frac{6}{c}$, then $\sum_{m\in\mathcal{B}(t)}\mathbf{1}_{\{m\in\mathcal{S}(t)\}}\leq\frac{2}{p^2}\log\frac{6}{c}$. Otherwise, with probability at least $1-\frac{c}{6}$, it is true that $\sum_{m\in\mathcal{B}(t)}\mathbf{1}_{\{m\in\mathcal{S}(t)\}}\leq\frac{3}{2}p\tau(t)$.

On the other hand, when the system adversary is adaptive, it chooses $\mathcal{B}(t)$ based on $\mathcal{S}(t)$. In particular, if $|\mathcal{S}(t)|\leq\tau(t)$, then the adversary chooses $\mathcal{B}(t)=\mathcal{S}(t)$. Otherwise, i.e., $|\mathcal{S}(t)|>\tau(t)$, the adversary chooses an arbitrary subset of $\mathcal{S}(t)$. In both cases, it holds that

$$
\sum_{m\in\mathcal{B}(t)}X_{mi}\leq\sum_{m\in\mathcal{B}(t)}\mathbf{1}_{\{m\in\mathcal{S}(t)\}}\leq\min\{\tau(t),|\mathcal{S}(t)|\}\leq\tau(t). \tag{12}
$$

For ease of exposition, we first focus on adaptive adversary and will visit the static adversary towards the end of this proof. Observe that $|\mathcal{S}(t)|=\sum_{m=1}^{M}\mathbf{1}_{\{m\in\mathcal{S}(t)\}}$. Let $\widetilde{Y}_{mi}=\widetilde{X}_{mi}-\frac{\mathbf{1}_{\{m\in\mathcal{S}(t)\}}}{2}$. Conditioning on the mini-batch stochastic gradients $\boldsymbol{g}_{mi}^{1},\cdots,\boldsymbol{g}_{mi}^{n}$, we have

$$
\mathbb{E}\left[\widetilde{Y}_{mi}\mid\boldsymbol{g}_{mi}^{1},\cdots,\boldsymbol{g}_{mi}^{n}\right]=\mathbb{E}\left[\widetilde{X}_{mi}\mid\boldsymbol{g}_{mi}^{1},\cdots,\boldsymbol{g}_{mi}^{n}\right]-\frac{p}{2}=\frac{p}{2B+2\beta}\text{clip}\left(\frac{1}{n}\sum_{j=1}^{n}\boldsymbol{g}_{mi}^{j},B\right).
$$

Taking expectation over $\boldsymbol{g}_{mi}^{1},\cdots,\boldsymbol{g}_{mi}^{n}$, we get

$$
\mathbb{E}\left[\mathbb{E}\left[\widetilde{Y}_{mi}\mid\boldsymbol{g}_{mi}^{1},\cdots,\boldsymbol{g}_{mi}^{n}\right]\right]=\mathbb{E}\left[\mathbb{E}\left[\widetilde{Y}_{mi}\mid\boldsymbol{g}_{mi}^{1},\cdots,\boldsymbol{g}_{mi}^{n}\right]-p\frac{\frac{1}{n}\sum_{j=1}^{n}\boldsymbol{g}_{mi}^{j}}{2B+2\beta}\right]+\frac{p\boldsymbol{g}_{mi}}{2B+2\beta}
$$

$$
=\mathbb{E}\left[\mathbb{E}\left[\widetilde{Y}_{mi}\mid\boldsymbol{g}_{mi}^{1},\cdots,\boldsymbol{g}_{mi}^{n}\right]-p\frac{\frac{1}{n}\sum_{j=1}^{n}\boldsymbol{g}_{mi}^{j}}{2B+2\beta}\right]+\frac{p\boldsymbol{g}_{mi}}{2B+2\beta}. \tag{13}
$$

It turns out that $\mathbb{E}\left[\mathbb{E}\left[\widetilde{Y}_{mi}\mid\boldsymbol{g}_{mi}^{1},\cdots,\boldsymbol{g}_{mi}^{n}\right]-p\frac{\frac{1}{n}\sum_{j=1}^{n}\boldsymbol{g}_{mi}^{j}}{2B+2\beta}\right]$ is small:

$$
\frac{1}{p}\mathbb{E}\left[\mathbb{E}\left[\widetilde{Y}_{mi}\mid\boldsymbol{g}_{mi}^{1},\cdots,\boldsymbol{g}_{mi}^{n}\right]-p\frac{\frac{1}{n}\sum_{j=1}^{n}\boldsymbol{g}_{mi}^{j}}{2B+2\beta}\right]
$$

$$
=\underbrace{\frac{B\mathbb{P}\left\{\frac{1}{n}\sum_{j=1}^{n}\boldsymbol{g}_{mi}^{j}\geq B\right\}-B\mathbb{P}\left\{\frac{1}{n}\sum_{j=1}^{n}\boldsymbol{g}_{mi}^{j}\leq-B\right\}}{2B+2\beta}}_{(A)}+\underbrace{\frac{\mathbb{E}\left[-\frac{1}{n}\sum_{j=1}^{n}\boldsymbol{g}_{mi}^{j}\mathbf{1}_{\left\{\left|\frac{1}{n}\sum_{j=1}^{n}\boldsymbol{g}_{mi}^{j}\right|\geq B\right\}}\right]}{2B+2\beta}}_{(B)}.
$$

We bound (A) and (B) for sub-Gaussian and heavy-tailed noise separately.

First, for sub-Gaussian distributions with Assumption 4, we have

$$
\begin{aligned}
(A) \leq & \frac{B}{2B+2\beta} \mathbb{P}\left\{\frac{1}{n}\sum_{j=1}^{n}\boldsymbol{g}_{mi}^{j} - \mathbb{E}\left[\frac{1}{n}\sum_{j=1}^{n}\boldsymbol{g}_{mi}^{j}\right] \geq B - \mathbb{E}\left[\frac{1}{n}\sum_{j=1}^{n}\boldsymbol{g}_{mi}^{j}\right]\right\} \\
\leq & \frac{B}{2B+2\beta} \exp\left(-\frac{n\left(B-\boldsymbol{g}_{mi}\right)^{2}}{2\sigma_{mi}^{2}}\right) \\
\leq & \frac{B}{2B+2\beta} \exp\left(-\frac{n\epsilon_{0}^{2}B_{0}^{2}}{2\sigma_{mi}^{2}}\right) \\
\leq & \frac{1}{2} \exp\left(-\frac{n}{2}\right) \quad [\text{since } \epsilon_{0} > \frac{\sigma}{B_{0}}],
\end{aligned}
$$

and

$$
\begin{aligned}
(B) = & \frac{\mathbb{E}\left[-\frac{1}{n}\sum_{j=1}^{n}\boldsymbol{g}_{mi}^{j}\mathbf{1}_{\left\{\left|\frac{1}{n}\sum_{j=1}^{n}\boldsymbol{g}_{mi}^{j}\right|\geq B\right\}}\right]}{2B+2\beta} \\
= & \frac{\int_{-\infty}^{-B}\mathbb{P}\left\{\frac{1}{n}\sum_{j=1}^{n}\boldsymbol{g}_{mi}^{j}<t\right\}\mathrm{d}t - \int_{B}^{+\infty}\mathbb{P}\left\{\frac{1}{n}\sum_{j=1}^{n}\boldsymbol{g}_{mi}^{j}>t\right\}\mathrm{d}t}{2B+2\beta} \\
\leq & \frac{\int_{-\infty}^{-B}\mathbb{P}\left\{\frac{1}{n}\sum_{j=1}^{n}\boldsymbol{g}_{mi}^{j} - \mathbb{E}\left[\frac{1}{n}\sum_{j=1}^{n}\boldsymbol{g}_{mi}^{j}\right]<t-\mathbb{E}\left[\frac{1}{n}\sum_{j=1}^{n}\boldsymbol{g}_{mi}^{j}\right]\right\}\mathrm{d}t}{2B+2\beta} \\
\leq & \frac{\int_{-\infty}^{-B}\exp\left(-\frac{(t-\boldsymbol{g}_{mi})^{2}}{2\sigma_{mi}^{2}/n}\right)\mathrm{d}t}{2B+2\beta} \quad [\text{Mill's ratio Gordon (1941)}] \\
= & \frac{1}{2B+2\beta}\int_{-\infty}^{-B}\left[-\frac{2\sigma_{mi}^{2}/n}{2\left(t-\boldsymbol{g}_{mi}\right)}\right]\left[-\frac{2\left(t-\boldsymbol{g}_{mi}\right)}{2\sigma_{mi}^{2}/n}\right]\exp\left[-\frac{(t-\boldsymbol{g}_{mi})^{2}}{2\sigma_{mi}^{2}/n}\right]\mathrm{d}t \\
\leq & \frac{\sigma_{mi}^{2}/n}{(2B+2\beta)\left(B+\boldsymbol{g}_{mi}\right)}\int_{-\infty}^{-B}\left[-\frac{2\left(t-\boldsymbol{g}_{mi}\right)}{2\sigma_{mi}^{2}/n}\right]\exp\left(-\frac{(t-\boldsymbol{g}_{mi})^{2}}{2\sigma_{mi}^{2}/n}\right)\mathrm{d}t \\
\leq & \frac{\sigma_{mi}^{2}}{n\epsilon_{0}B_{0}(2B+2\beta)}\exp\left(-\frac{n\epsilon_{0}^{2}B_{0}^{2}}{2\sigma_{mi}^{2}}\right) \\
\leq & \frac{\sigma_{mi}^{2}}{2n\epsilon_{0}^{2}B_{0}^{2}}\exp\left(-\frac{n\epsilon_{0}^{2}B_{0}^{2}}{2\sigma_{mi}^{2}}\right) \quad [\beta>0 \text{ and } B:=(1+\epsilon_{0})B_{0}>\epsilon_{0}B_{0}] \\
\leq & \frac{1}{2n}\exp\left(-\frac{n}{2}\right),
\end{aligned}
$$

where the last inequality follows from the choice of $\epsilon_{0} > \frac{\sigma}{B_{0}}$. Combining the bounds of (A) and (B), we get $\mathbb{E}\left[\mathbb{E}\left[\widetilde{Y}_{mi} \mid \boldsymbol{g}_{mi}^{1}, \cdots, \boldsymbol{g}_{mi}^{n}\right] - p\frac{\frac{1}{n}\sum_{j=1}^{n}\boldsymbol{g}_{mi}^{j}}{2B+2\beta}\right] \leq p\exp\left(-\frac{n}{2}\right)$. Hence,

$$
\mathbb{E}\left[\widetilde{Y}_{mi}\right] \leq p\exp\left(-\frac{n}{2}\right) + \frac{p\boldsymbol{g}_{mi}}{2B+2\beta}. \tag{14}
$$

Second, for heavy-tailed distributions with Assumption 5, we have

$$
\begin{aligned}
\text{(A)} \leq & \frac{B}{2B+2\beta}\mathbb{P}\left\{\frac{1}{n}\sum_{j=1}^{n}\boldsymbol{g}_{mi}^{j} - \mathbb{E}\left[\frac{1}{n}\sum_{j=1}^{n}\boldsymbol{g}_{mi}^{j}\right] \geq B - \mathbb{E}\left[\frac{1}{n}\sum_{j=1}^{n}\boldsymbol{g}_{mi}^{j}\right]\right\} \\
\leq & \frac{B}{2B+2\beta}\mathbb{P}\left\{\left|\sum_{j=1}^{n}\boldsymbol{g}_{mi}^{j} - \mathbb{E}\left[\sum_{j=1}^{n}\boldsymbol{g}_{mi}^{j}\right]\right|^{p'} \geq n^{p'}\left|B - \boldsymbol{g}_{mi}\right|^{p'}\right\} \\
\leq & \frac{B}{2B+2\beta}\frac{\mathbb{E}\left[\left|\sum_{j=1}^{n}\boldsymbol{g}_{mi}^{j} - \mathbb{E}\left[\sum_{j=1}^{n}\boldsymbol{g}_{mi}^{j}\right]\right|^{p'}\right]}{n^{p'}\left|B - \boldsymbol{g}_{mi}\right|^{p'}} \quad \text{[Markov's inequality]} \\
\leq & \underbrace{\frac{B\sum_{j=1}^{n}\mathbb{E}\left[\left|\boldsymbol{g}_{mi}^{j} - \mathbb{E}\left[\boldsymbol{g}_{mi}^{j}\right]\right|^{p'}\right] + B\left(\sum_{j=1}^{n}\mathbb{E}\left[\left|\boldsymbol{g}_{mi}^{j} - \mathbb{E}\left[\boldsymbol{g}_{mi}^{j}\right]\right|^{2}\right]\right)^{\frac{p'}{2}}}{(2B+2\beta)n^{p'}\left|B - \boldsymbol{g}_{mi}\right|^{p'}}}_{\text{Rosenthal-type inequality Merlevède \& Peligrad (2013)}} \\
\leq & \frac{1}{2}\frac{nM_{p'} + n^{\frac{p'}{2}}M_{p'}}{n^{p'}\left|B - \boldsymbol{g}_{mi}\right|^{p'}} \quad \left[M_{2}^{\frac{1}{2}} \leq M_{p'}^{\frac{1}{p'}} \text{ for } p' \geq 4\right] \\
\leq & \frac{M_{p'}}{n^{\frac{p'}{2}}\epsilon_{0}^{p'}B_{0}^{p'}} \leq \frac{1}{n^{\frac{p'}{2}}}
\end{aligned}
$$

and

$$
\begin{aligned}
\text{(B)} = & \frac{\mathbb{E}\left[-\frac{1}{n}\sum_{j=1}^{n}\boldsymbol{g}_{mi}^{j}\mathbf{1}_{\left\{\left|\frac{1}{n}\sum_{j=1}^{n}\boldsymbol{g}_{mi}^{j}\right|\geq B\right\}}\right]}{2B+2\beta} \\
= & \frac{\int_{-\infty}^{-B}\mathbb{P}\left\{\frac{1}{n}\sum_{j=1}^{n}\boldsymbol{g}_{mi}^{j} < t\right\}\mathrm{d}t - \int_{B}^{+\infty}\mathbb{P}\left\{\frac{1}{n}\sum_{j=1}^{n}\boldsymbol{g}_{mi}^{j} > t\right\}\mathrm{d}t}{2B+2\beta} \\
\leq & \frac{\int_{-\infty}^{-B}\mathbb{P}\left\{\frac{1}{n}\sum_{j=1}^{n}\boldsymbol{g}_{mi}^{j} - \mathbb{E}\left[\frac{1}{n}\sum_{j=1}^{n}\boldsymbol{g}_{mi}^{j}\right] < t - \mathbb{E}\left[\frac{1}{n}\sum_{j=1}^{n}\boldsymbol{g}_{mi}^{j}\right]\right\}\mathrm{d}t}{2B+2\beta} \\
\leq & \frac{1}{2B+2\beta}\int_{-\infty}^{-B}\frac{2M_{p'}}{n^{\frac{p'}{2}}\left|t - \boldsymbol{g}_{mi}\right|^{p'}}\mathrm{d}t \quad \text{[similar argument as in (A)]} \\
\leq & \frac{1}{2B+2\beta}\frac{1}{\epsilon_{0}^{p'-1}B_{0}^{p'-1}(p'-1)n^{\frac{p'}{2}}} \leq \frac{1}{(p'-1)n^{\frac{p'}{2}}} \leq \frac{1}{n^{\frac{p'}{2}}},
\end{aligned}
$$

where the last inequality follows from the choice of $\epsilon_{0} > \frac{M_{p'}^{\frac{1}{p'}}}{B_{0}}$. Combining the bounds of (A) and (B), we get $\mathbb{E}\left[\mathbb{E}\left[\widetilde{Y}_{mi} \mid \boldsymbol{g}_{mi}^{1}, \cdots, \boldsymbol{g}_{mi}^{n}\right] - p\frac{\frac{1}{n}\sum_{j=1}^{n}\boldsymbol{g}_{mi}^{j}}{2B+2\beta}\right] \leq \frac{2p}{n^{\frac{p'}{2}}}$. Hence,

$$
\mathbb{E}\left[\widetilde{Y}_{mi}\right] \leq \frac{2p}{n^{\frac{p'}{2}}} + \frac{p\boldsymbol{g}_{mi}}{2B+2\beta}. \tag{15}
$$

Let us consider two mutually complement events $\mathcal{E}_{1}$ and $\mathcal{E}_{2}$:

$$
\mathcal{E}_{1} := \left\{\frac{1}{2(B+\beta)}\sum_{m=1}^{M}\mathsf{clip}\left(\frac{1}{n}\sum_{j=1}^{n}\boldsymbol{g}_{mi}^{j}, B\right) - \mathbb{E}\left[\frac{1}{2(B+\beta)}\sum_{m=1}^{M}\mathsf{clip}\left(\frac{1}{n}\sum_{j=1}^{n}\boldsymbol{g}_{mi}^{j}, B\right)\right] \leq \frac{c_{0}(n,p)}{4(B+\beta)}\sqrt{M}\right\},
$$

$$
\mathcal{E}_{2} := \left\{\frac{1}{2(B+\beta)}\sum_{m=1}^{M}\mathsf{clip}\left(\frac{1}{n}\sum_{j=1}^{n}\boldsymbol{g}_{mi}^{j}, B\right) - \mathbb{E}\left[\frac{1}{2(B+\beta)}\sum_{m=1}^{M}\mathsf{clip}\left(\frac{1}{n}\sum_{j=1}^{n}\boldsymbol{g}_{mi}^{j}, B\right)\right] > \frac{c_{0}(n,p)}{4(B+\beta)}\sqrt{M}\right\}.
$$

We have

$$\mathbb{P}\left\{\sum_{m=1}^{M}\widetilde{X}_{mi} \geq \frac{|\mathcal{S}(t)|}{2} - \tau(t)\right\} \leq \mathbb{P}\left\{\sum_{m=1}^{M}\widetilde{Y}_{mi} \geq -\tau(t) \mid \mathcal{E}_1\right\} + \mathbb{P}\left\{\mathcal{E}_2\right\}. \qquad (16)$$

By Proposition 3, we know that

$$\mathsf{var}\left(\mathsf{clip}\left(\frac{1}{n}\sum_{j=1}^{n}\boldsymbol{g}_{mi}^{j}, B\right)\right) \leq \mathsf{var}\left(\frac{1}{n}\sum_{j=1}^{n}\boldsymbol{g}_{mi}^{j}\right) \leq \frac{1}{n}\mathsf{var}\left(\boldsymbol{g}_{mi}^{1}\right) = \frac{1}{n}\sigma_{mi}^{2} \leq \frac{1}{n}\sigma^{2}.$$

In addition, $\mathsf{clip}\left(\frac{1}{n}\sum_{j=1}^{n}\boldsymbol{g}_{mi}^{j}, B\right)$ is bounded and thus sub-Gaussian. Hence, we have

$$\mathbb{P}\left\{\mathcal{E}_2\right\} \leq \exp\left(-\frac{\frac{c_0^2(n,p)M}{4}}{\frac{2M\sigma^2}{n}}\right).$$

Since $c_0(n,p) \geq \sqrt{\frac{8\sigma^2}{n}\log\frac{6}{c}}$, we have $\mathbb{P}\left\{\mathcal{E}_2\right\} \leq \frac{c}{6}$.

For the first term in the right-hand side of Eq. (16), we have

$$\mathbb{P}\left\{\sum_{m=1}^{M}\widetilde{Y}_{mi} \geq -\tau(t) \mid \mathcal{E}_1\right\}$$

$$=\mathbb{P}\left\{\sum_{m=1}^{M}\widetilde{Y}_{mi} - \mathbb{E}\left[\sum_{m=1}^{M}\widetilde{Y}_{mi} \mid \boldsymbol{g}_{mi}^{1}, \cdots, \boldsymbol{g}_{mi}^{n}\right] \geq \underbrace{-\tau(t) - \mathbb{E}\left[\sum_{m=1}^{M}\widetilde{Y}_{mi} \mid \boldsymbol{g}_{mi}^{1}, \cdots, \boldsymbol{g}_{mi}^{n}\right]}_{(C)} \mid \mathcal{E}_1\right\}$$

Recall that $\mathbb{E}\left[\widetilde{Y}_{mi} \mid \boldsymbol{g}_{mi}^{1}, \cdots, \boldsymbol{g}_{mi}^{n}\right] = \frac{p}{2B+2\beta}\mathsf{clip}\left(\frac{1}{n}\sum_{j=1}^{n}\boldsymbol{g}_{mi}^{j}, B\right)$. We have

$$(C) \mid \mathcal{E}_1 = -\tau(t) - \frac{p}{2B+2\beta}\sum_{m=1}^{M}\mathsf{clip}\left(\frac{1}{n}\sum_{j=1}^{n}\boldsymbol{g}_{mi}^{j}, B\right) \mid \mathcal{E}_1$$

$$\geq -\tau(t) - \mathbb{E}\left[\frac{p}{2B+2\beta}\sum_{m=1}^{M}\mathsf{clip}\left(\frac{1}{n}\sum_{j=1}^{n}\boldsymbol{g}_{mi}^{j}, B\right)\right] - \frac{pc_0(n,p)}{4(B+\beta)}\sqrt{M}$$

$$= -\tau(t) - \sum_{m=1}^{M}\mathbb{E}\left[\widetilde{Y}_{mi}\right] - \frac{pc_0(n,p)}{4(B+\beta)}\sqrt{M}$$

$$\begin{cases} \geq -\tau(t) - Mp\exp\left(-\frac{n}{2}\right) - \frac{pM}{2(B+\beta)}\nabla_i F(w(t)) - \frac{pc_0(n,p)}{4(B+\beta)}\sqrt{M} & \text{[Sub-Gaussian Noise]} \\ \geq -\tau(t) - \frac{2Mp}{n^{\frac{p'}{2}}} - \frac{pM}{2(B+\beta)}\nabla_i F(w(t)) - \frac{pc_0(n,p)}{4(B+\beta)}\sqrt{M} & \text{[Heavy-tailed Noise]} \end{cases}$$

Recall that $\nabla_i F(w(t)) < 0$. When $\frac{pM}{2(B+\beta)}|\nabla_i F(w(t))| \geq \tau(t) + Mp\exp\left(-\frac{n}{2}\right) + \frac{pc_0(n,p)}{2(B+\beta)}\sqrt{M}$ (sub-Gaussian noise) or when $\frac{pM}{2(B+\beta)}|\nabla_i F(w(t))| \geq \tau(t) + \frac{2Mp}{n^{\frac{p'}{2}}} + \frac{pc_0(n,p)}{2(B+\beta)}\sqrt{M}$ (heavy-tailed noise), we get

$$\mathbb{P}\left\{\sum_{m=1}^{M}\widetilde{Y}_{mi} \geq -\tau(t) \mid \mathcal{E}_1\right\} \leq \mathbb{P}\left\{\sum_{m=1}^{M}\widetilde{Y}_{mi} - \mathbb{E}\left[\sum_{m=1}^{M}\widetilde{Y}_{mi} \mid \boldsymbol{g}_{mi}^{1}, \cdots, \boldsymbol{g}_{mi}^{n}\right] \geq \frac{pc_0(n,p)}{4(B+\beta)}\sqrt{M} \mid \mathcal{E}_1\right\}$$

$$\leq \exp\left(-\frac{p^2 c_0^2(n,p)}{8(B+\beta)^2}\right)$$

$$\leq \frac{3-5c}{6},$$

where the last inequality holds because $c_0(n, p) \geq \sqrt{\frac{8(B+\beta)^2}{p^2} \log \frac{6}{3-5c}}$.

Therefore, for adaptive system adversary, choosing $c_0(n, p) = \max \left\{ \sqrt{\frac{8\sigma^2}{n} \log \frac{6}{c}}, \sqrt{\frac{8(B+\beta)^2}{p^2} \log \frac{6}{3-5c}} \right\}$, we conclude that if $\frac{pM}{2(B+\beta)} |\nabla_i F(w(t))| \geq \tau(t) + Mp \exp\left(-\frac{n}{2}\right) + \frac{pc_0(n,p)}{2(B+\beta)} \sqrt{M}$ (sub-Gaussian Noise) or if $\frac{pM}{2(B+\beta)} |\nabla_i F(w(t))| \geq \tau(t) + \frac{2Mp}{n^{\frac{p'}{2}}} + \frac{pc_0(n,p)}{2(B+\beta)} \sqrt{M}$ (heavy-tailed noise), then

$$\mathbb{P}\left\{ \text{sign}\left( \frac{1}{M} \sum_{m=1}^{M} \widehat{\boldsymbol{g}}_{mi} \right) \neq \text{sign}\left( \nabla_i F(w(t)) \right) \mid w(t) \right\} \leq \frac{1-c}{2}.$$

Otherwise, $\mathbb{P}\left\{ \text{sign}\left( \frac{1}{M} \sum_{m=1}^{M} \widehat{\boldsymbol{g}}_{mi} \right) \neq \text{sign}\left( \nabla_i F(w(t)) \right) \mid w(t) \right\} \leq 1$.

It remains to show the case for static adversary. When $\tau(t) \leq \frac{2}{p^2} \log \frac{6}{c}$, we bound Eq. (10) as

$$\mathbb{P}\left\{ \sum_{m=1}^{M} \widetilde{X}_{mi} \geq \frac{|\mathcal{S}(t)|}{2} - \sum_{m \in \mathcal{B}(t)} X_{mi} \right\} \leq \mathbb{P}\left\{ \sum_{m=1}^{M} \widetilde{X}_{mi} \geq \frac{|\mathcal{S}(t)|}{2} - \tau(t) \right\}.$$

When $\tau(t) > \frac{2}{p^2} \log \frac{6}{c}$, we bound Eq. (10) as

$$\mathbb{P}\left\{ \sum_{m=1}^{M} \widetilde{X}_{mi} \geq \frac{|\mathcal{S}(t)|}{2} - \sum_{m \in \mathcal{B}(t)} X_{mi} \right\} \leq \mathbb{P}\left\{ \sum_{m=1}^{M} \widetilde{X}_{mi} \geq \frac{|\mathcal{S}(t)|}{2} - \frac{3p}{2}\tau(t) \right\} + \frac{c}{6}.$$

The remaining proof follows the above argument for adaptive adversary. $\square$

**Proof of Theorem 4 (Sub-Gaussian and Heavy-tailed Convergence Rate).** By Assumption 2, we have

$$F(w(t+1)) - F(w(t)) \leq \langle \nabla F(w(t)), w(t+1) - w(t) \rangle + \frac{L}{2} \|w(t+1) - w(t)\|^2$$

$$= -\eta \sum_{i=1}^{d} |\nabla F(w(t))_i| \mathbf{1}_{\{\widetilde{\boldsymbol{g}}_i = \text{sign}(\nabla_i F(w(t)))\}}$$

$$+ \eta \sum_{i=1}^{d} |\nabla F(w(t))_i| \mathbf{1}_{\{\widetilde{\boldsymbol{g}}_i \neq \text{sign}(\nabla_i F(w(t)))\}} + \frac{Ld}{2}\eta^2$$

$$= -\eta \|\nabla F(w(t))\|_1 + 2\eta \sum_{i=1}^{d} |\nabla F(w(t))_i| \mathbf{1}_{\{\widetilde{\boldsymbol{g}}_i \neq \text{sign}(\nabla_i F(w(t)))\}} + \frac{Ld}{2}\eta^2,$$

where $\nabla F(w(t))_i$ is the $i$-th coordinate of $\nabla F(w(t))$. Then, by conditioning on parameter $w(t)$, we get

$$\mathbb{E}\left[ F(w(t+1)) - F(w(t)) \,\big|\, w(t) \right]$$

$$\leq \mathbb{E}\left[ -\eta \|\nabla F(w(t))\|_1 + 2\eta \sum_{i=1}^{d} |\nabla F(w(t))_i| \mathbf{1}_{\{\widetilde{\boldsymbol{g}}_i \neq \text{sign}(\nabla F(w(t))_i)\}} + \frac{Ld}{2}\eta^2 \right]$$

$$= -\eta \|\nabla F(w(t))\|_1 + \frac{Ld}{2}\eta^2 + 2\eta \sum_{i=1}^{d} |\nabla F(w(t))_i| \mathbb{P}\left\{ \widetilde{\boldsymbol{g}}_i \neq \text{sign}(\nabla F(w(t))_i) \right\}.$$

Recall that $\Xi_1(n) = 2(B+\beta) \exp\left(-\frac{n}{2}\right)$, and $\Xi_2(n) = \frac{4(B+\beta)}{n^{\frac{p'}{2}}}$. Define

$$\begin{cases} A_1 = \left\{ |\nabla_i F(w(t))| \geq \frac{2(B+\beta)}{pM}\tau(t) + \frac{c_0(n,p)}{\sqrt{M}} + 2(B+\beta) \exp\left(-\frac{n}{2}\right) \right\}; \\ A_2 = \left\{ |\nabla_i F(w(t))| \geq \frac{2(B+\beta)}{pM}\tau(t) + \frac{c_0(n,p)}{\sqrt{M}} + \frac{4(B+\beta)}{n^{\frac{p'}{2}}} \right\}. \end{cases}$$

In the following proof, we denote $A = A_1$, $\Xi(n) = \Xi_1(n)$ for sub-Gaussian noise and $A = A_2$, $\Xi(n) = \Xi_2(n)$ for heavy-tailed noise.

We now have two cases:

First, when the system adversary is adaptive or the system adversary is static but with $\tau(t) \leq \frac{2}{p^2} \log \frac{6}{c}$, then

$$
\mathbb{E}\left[F\left(w(t+1)\right) - F\left(w(t)\right) \big| w(t)\right]
$$

$$
= -\eta \|\nabla F(w(t))\|_1 + \frac{Ld}{2}\eta^2 + 2\eta \sum_{i=1}^{d} |\nabla F(w(t))_i| \, \mathbb{P}\left\{\widetilde{g}_i \neq \mathrm{sign}\left(\nabla_i F(w(t))\right)\right\} \mathbf{1}_{\{A\}}
$$

$$
+ 2\eta \sum_{i=1}^{d} |\nabla F(w(t))_i| \, \mathbb{P}\left\{\widetilde{g}_i \neq \mathrm{sign}\left(\nabla_i F(w(t))\right)\right\} \mathbf{1}_{\{A^{\complement}\}}
$$

$$
\leq -\eta \|\nabla F(w(t))\|_1 + \frac{Ld}{2}\eta^2
$$

$$
+ 2\eta \sum_{i=1}^{d} |\nabla F(w(t))_i| \frac{1-c}{2} \mathbf{1}_{\{A\}}
$$

$$
+ 2\eta \sum_{i=1}^{d} \left[\frac{2(B+\beta)\tau(t)}{pM} + \frac{c_0(n,p)}{\sqrt{M}} + \Xi(n)\right] \mathbf{1}_{\{A^{\complement}\}}
$$

$$
\leq -\eta c \|\nabla F(w(t))\|_1 + \frac{Ld}{2}\eta^2 + 2\eta d \frac{c_0(n,p)}{\sqrt{M}} + 4\eta d \frac{(B+\beta)\tau(t)}{pM} + 2\eta d \Xi(n).
$$

Therefore, by Assumption 1, we have

$$
F^* - F(w(0)) \leq \mathbb{E}\left[F\left(w(T)\right) - F\left(w(0)\right)\right]
$$

$$
\leq -\eta c \sum_{t=0}^{T-1} \mathbb{E}\left[\|\nabla F(w(t))\|_1\right] + \frac{\eta^2 LdT}{2} + 2\eta dT \frac{c_0(n,p)}{\sqrt{M}} + 2\eta dT \Xi(n) + 4\eta d \frac{(B+\beta)\sum_{t=0}^{T-1}\tau(t)}{pM}.
$$

Rearrange the inequality and plug in $\eta = \frac{1}{\sqrt{dT}}$, we get

$$
\eta c \sum_{t=0}^{T-1} \mathbb{E}\left[\|\nabla F(w(t))\|_1\right] \leq F(w(0)) - F^* + \frac{\eta^2 LdT}{2} + 2\eta dT \frac{c_0(n,p)}{\sqrt{M}} + 2\eta dT \Xi(n) + 4\eta d \frac{(B+\beta)\sum_{t=0}^{T-1}\tau(t)}{pM}
$$

$$
\frac{1}{T}\sum_{t=0}^{T-1} \mathbb{E}\left[\|\nabla F(w(t))\|_1\right] \leq \frac{1}{c}\left[\frac{(F(w(0)) - F^*)\sqrt{d}}{\sqrt{T}} + \frac{L\sqrt{d}}{2\sqrt{T}} + 2d\frac{c_0(n,p)}{\sqrt{M}} + 4d\frac{(B+\beta)\sum_{t=0}^{T-1}\tau(t)}{pTM} + 2d\Xi(n)\right].
$$

Second, when the system adversary is static with $\tau(t) > \frac{2}{p^2} \log \frac{6}{c}$, follow a similar proof as above, we get

$$
\frac{1}{T}\sum_{t=0}^{T-1} \mathbb{E}\left[\|\nabla F(w(t))\|_1\right] \leq \frac{1}{c}\left[\frac{(F(w(0)) - F^*)\sqrt{d}}{\sqrt{T}} + \frac{L\sqrt{d}}{2\sqrt{T}} + 2d\frac{c_0(n,p)}{\sqrt{M}} + 6d\frac{(B+\beta)\sum_{t=0}^{T-1}\tau(t)}{TM} + 2d\Xi(n)\right].
$$

$\square$

### D.3.2 GAUSSIAN DISTRIBUTION

**Proof of Corollary 2 (Gaussian Tail Sign Errors).** Most of the proofs are the same with Theorem 3. We start from Eq. 13.

It turns out that $\mathbb{E}\left[\mathbb{E}\left[\widetilde{Y}_{mi} \mid \boldsymbol{g}_{mi}^1, \cdots, \boldsymbol{g}_{mi}^n\right] - p \frac{\frac{1}{n}\sum_{j=1}^n \boldsymbol{g}_{mi}^j}{2B+2\beta}\right]$ is small:

$$\frac{1}{p}\mathbb{E}\left[\mathbb{E}\left[\widetilde{Y}_{mi} \mid \boldsymbol{g}_{mi}^1, \cdots, \boldsymbol{g}_{mi}^n\right] - p \frac{\frac{1}{n}\sum_{j=1}^n \boldsymbol{g}_{mi}^j}{2B+2\beta}\right] = \underbrace{\frac{(B - \boldsymbol{g}_{mi})\,\mathbb{P}\left\{\frac{1}{n}\sum_{j=1}^n \boldsymbol{g}_{mi}^j \geq B\right\}}{2B+2\beta}}_{(A)}$$

$$-\underbrace{\frac{(B + \boldsymbol{g}_{mi})\,\mathbb{P}\left\{\frac{1}{n}\sum_{j=1}^n \boldsymbol{g}_{mi}^j \leq -B\right\}}{2B+2\beta}}_{(B)}$$

$$+\underbrace{\frac{\mathbb{E}\left[\left(-\frac{1}{n}\sum_{j=1}^n \boldsymbol{g}_{mi}^j + \boldsymbol{g}_{mi}\right)\mathbf{1}_{\left\{|\frac{1}{n}\sum_{j=1}^n \boldsymbol{g}_{mi}^j| \geq B\right\}}\right]}{2B+2\beta}}_{(C)}. \tag{17}$$

We have,

$(2B+2\beta)\,(A)$

$$\leq (B - \boldsymbol{g}_{mi}) \cdot \frac{\sigma_{mi}/\sqrt{n}}{B - \boldsymbol{g}_{mi}} \cdot \frac{1}{\sqrt{2\pi}} \cdot \exp\left(-\frac{(B - \boldsymbol{g}_{mi})^2}{2\left(\sigma_{mi}/\sqrt{n}\right)^2}\right) = \frac{\sigma_{mi}/\sqrt{n}}{\sqrt{2\pi}}\exp\left(-\frac{(B - \boldsymbol{g}_{mi})^2}{2\left(\sigma_{mi}/\sqrt{n}\right)^2}\right);$$

$(2B+2\beta)\,(B)$

$$\geq (B + \boldsymbol{g}_{mi}) \cdot \frac{\frac{B+\boldsymbol{g}_{mi}}{\sigma_{mi}/\sqrt{n}}}{\left(\frac{B+\boldsymbol{g}_{mi}}{\sigma_{mi}/\sqrt{n}}\right)^2 + 1} \cdot \frac{1}{\sqrt{2\pi}} \cdot \exp\left(-\frac{(B + \boldsymbol{g}_{mi})^2}{2\left(\sigma_{mi}/\sqrt{n}\right)^2}\right)$$

$$= \left[1 - \frac{(\sigma_{mi}/\sqrt{n})^2}{(B + \boldsymbol{g}_{mi})^2 + (\sigma_{mi}/\sqrt{n})^2}\right]\frac{\sigma_{mi}/\sqrt{n}}{\sqrt{2\pi}}\exp\left(-\frac{(B + \boldsymbol{g}_{mi})^2}{2\left(\sigma_{mi}/\sqrt{n}\right)^2}\right);$$

$(2B+2\beta)\,(C)$

$$= -\int_B^\infty \frac{x - \boldsymbol{g}_{mi}}{\sqrt{2\pi}\sigma_{mi}/\sqrt{n}}\exp\left(-\frac{(x - \boldsymbol{g}_{mi})^2}{2\left(\sigma_{mi}/\sqrt{n}\right)^2}\right)\mathrm{d}x - \int_{-\infty}^{-B} \frac{x - \boldsymbol{g}_{mi}}{\sqrt{2\pi}\sigma_{mi}/\sqrt{n}}\exp\left(-\frac{(x - \boldsymbol{g}_{mi})^2}{2\left(\sigma_{mi}/\sqrt{n}\right)^2}\right)\mathrm{d}x;$$

$$= \frac{\sigma_{mi}/\sqrt{n}}{\sqrt{2\pi}}\left[\exp\left(-\frac{(B + \boldsymbol{g}_{mi})^2}{2\left(\sigma_{mi}/\sqrt{n}\right)^2}\right) - \exp\left(-\frac{(B - \boldsymbol{g}_{mi})^2}{2\left(\sigma_{mi}/\sqrt{n}\right)^2}\right)\right],$$

where (A) and (B) follow because of Mill's ratio Gordon (1941).

Combining (A), (B), and (C), we get

$$(17) \leq \frac{p\left(\sigma_{mi}/\sqrt{n}\right)^3}{\sqrt{2\pi}\,(2B+2\beta)\left[(B + \boldsymbol{g}_{mi})^2 + (\sigma_{mi}/\sqrt{n})^2\right]}\exp\left(-\frac{(B + \boldsymbol{g}_{mi})^2}{2\left(\sigma_{mi}/\sqrt{n}\right)^2}\right) + \frac{p\boldsymbol{g}_{mi}}{2B+2\beta}$$

$$\leq \frac{p\left(\sigma_{mi}/\sqrt{n}\right)^3}{\sqrt{2\pi}\,(2B+2\beta)\left[\epsilon_0^2 B_0^2 + (\sigma_{mi}/\sqrt{n})^2\right]}\exp\left(-\frac{\epsilon_0^2 B_0^2}{2\left(\sigma_{mi}/\sqrt{n}\right)^2}\right) + \frac{p\boldsymbol{g}_{mi}}{2B+2\beta}$$

$$\leq \frac{p}{4\sqrt{2\pi}}\exp\left(-\frac{n}{2}\right) + \frac{p\boldsymbol{g}_{mi}}{2B+2\beta},$$

where the last inequality follows because $\epsilon_0 > \frac{\sigma}{B_0}$ and $B := B_0 + \epsilon_0 B_0 > \epsilon_0 B_0$.

For the first term in the right hand side of Eq. (16), we have

$$
\mathbb{P}\left\{\sum_{m=1}^{M}\widetilde{Y}_{mi} \geq -\tau(t) \mid \mathcal{E}_1\right\}
$$

$$
=\mathbb{P}\left\{\sum_{m=1}^{M}\widetilde{Y}_{mi} - \mathbb{E}\left[\sum_{m=1}^{M}\widetilde{Y}_{mi} \mid \boldsymbol{g}_{mi}^1,\cdots,\boldsymbol{g}_{mi}^n\right] \geq \underbrace{-\tau(t) - \mathbb{E}\left[\sum_{m=1}^{M}\widetilde{Y}_{mi} \mid \boldsymbol{g}_{mi}^1,\cdots,\boldsymbol{g}_{mi}^n\right]}_{(D)} \mid \mathcal{E}_1\right\}
$$

Recall that $\mathbb{E}\left[\widetilde{Y}_{mi} \mid \boldsymbol{g}_{mi}^1,\cdots,\boldsymbol{g}_{mi}^n\right] = \frac{p}{2B+2\beta}\mathsf{clip}\left(\frac{1}{n}\sum_{j=1}^{n}\boldsymbol{g}_{mi}^j, B\right)$. We have

$$
(D) \mid \mathcal{E}_1 = -\tau(t) - \frac{p}{2B+2\beta}\sum_{m=1}^{M}\mathsf{clip}\left(\frac{1}{n}\sum_{j=1}^{n}\boldsymbol{g}_{mi}^j, B\right) \mid \mathcal{E}_1
$$

$$
\geq -\tau(t) - \mathbb{E}\left[\frac{p}{2B+2\beta}\sum_{m=1}^{M}\mathsf{clip}\left(\frac{1}{n}\sum_{j=1}^{n}\boldsymbol{g}_{mi}^j, B\right)\right] - \frac{pc_0(n,p)}{4(B+\beta)}\sqrt{M}
$$

$$
= -\tau(t) - \sum_{m=1}^{M}\mathbb{E}\left[\widetilde{Y}_{mi}\right] - \frac{pc_0(n,p)}{4(B+\beta)}\sqrt{M}
$$

$$
\geq -\tau(t) - \frac{Mp}{4\sqrt{2\pi}}\exp\left(-\frac{n}{2}\right) - \frac{p}{2(B+\beta)}\sum_{m=1}^{M}\boldsymbol{g}_{mi} - \frac{pc_0(n,p)}{4(B+\beta)}\sqrt{M}
$$

Recall that $\nabla_i F(w(t)) < 0$. When $\frac{Mp}{2(B+\beta)}|\nabla_i F(w(t))| \geq \tau(t) + \frac{Mp}{4\sqrt{2\pi}}\exp\left(-\frac{n}{2}\right) + \frac{pc_0(n,p)}{2(B+\beta)}\sqrt{M}$, we get

$$
\mathbb{P}\left\{\sum_{m=1}^{M}\widetilde{Y}_{mi} \geq -\tau(t) \mid \mathcal{E}_1\right\} \leq \mathbb{P}\left\{\sum_{m=1}^{M}\widetilde{Y}_{mi} - \mathbb{E}\left[\sum_{m=1}^{M}\widetilde{Y}_{mi} \mid \boldsymbol{g}_{mi}^1,\cdots,\boldsymbol{g}_{mi}^n\right] \geq \frac{pc_0(n,p)}{4(B+\beta)}\sqrt{M} \mid \mathcal{E}_1\right\}
$$

$$
\leq \exp\left(-\frac{p^2 c_0^2(n,p)}{8(B+\beta)^2}\right)
$$

$$
\leq \frac{3-5c}{6},
$$

where the last inequality holds because $c_0(n,p) \geq \sqrt{\frac{8(B+\beta)^2}{p^2}\log\frac{6}{3-5c}}$.

The remaining proof follows the arguments in Theorem 3. $\qquad\square$

**Proof of Corollary 3 (Gaussian Tail Convergence Rate).** This proof follows from Theorem 4. We also consider two cases here.

First, when the system adversary is adaptive or the system adversary is static but with $\tau(t) \leq \frac{2}{p^2}\log\frac{6}{c}$, plug in $|\nabla_i F(w(t))| \geq \frac{2(B+\beta)}{Mp}\tau(t) + \frac{(B+\beta)}{2\sqrt{2\pi}}\exp\left(-\frac{n}{2}\right) + \frac{c_0(n,p)}{\sqrt{M}}$, we get

$$
\frac{1}{T}\sum_{t=0}^{T-1}\mathbb{E}\left[\|\nabla F(w(t))\|_1\right] \leq \frac{1}{c}\left[\frac{(F(w(0))-F^*)\sqrt{d}}{\sqrt{T}} + \frac{L\sqrt{d}}{2\sqrt{T}} + 2d\frac{c_0(n,p)}{\sqrt{M}} + \frac{d}{\sqrt{2\pi}}(B+\beta)\exp\left(-\frac{n}{2}\right)\right.
$$

$$
\left. + 4d\frac{(B+\beta)\sum_{t=0}^{T-1}\tau(t)}{pTM}\right].
$$

**Second**, when the system adversary is static with $\tau(t) > \frac{2}{p^2} \log \frac{6}{c}$, plug in $|\nabla_i F(w(t))| \geq \frac{3(B+\beta)\tau(t)}{M} + \frac{(B+\beta)}{2\sqrt{2\pi}} \exp(-n/2) + \frac{c_0(n,p)}{\sqrt{M}}$, we get

$$\frac{1}{T} \sum_{t=0}^{T-1} \mathbb{E}\left[\|\nabla F(w(t))\|_1\right] \leq \frac{1}{c} \left[ \frac{(F(w(0)) - F^*)\sqrt{d}}{\sqrt{T}} + \frac{L\sqrt{d}}{2\sqrt{T}} + 2d\frac{c_0(n,p)}{\sqrt{M}} + \frac{d}{\sqrt{2\pi}}(B+\beta)\exp\left(-\frac{n}{2}\right) \right.$$
$$\left. + 6d\frac{(B+\beta)\sum_{t=0}^{T-1}\tau(t)}{TM} \right].$$

$\square$

### D.4 BOUNDED STOCHASTIC GRADIENTS

**Proof of Corollary 4 (Bounded Gradient Sign Errors).** This proof follows from Theorem 3. Notably, if we choose $B = \widetilde{B}$, clip $\left(\frac{1}{n}\sum_{j=1}^n g_{mi}^j, B\right) = \frac{1}{n}\sum_{j=1}^n g_{mi}^j$ by Assumption 6. Thus, the bias introduced by the tail bound will be gone.

For the first term in the right-hand side of Eq. (16), we have

$$\mathbb{P}\left\{\sum_{m=1}^M \widetilde{Y}_{mi} \geq -\tau(t) \mid \mathcal{E}_1\right\}$$

$$= \mathbb{P}\left\{\sum_{m=1}^M \widetilde{Y}_{mi} - \mathbb{E}\left[\sum_{m=1}^M \widetilde{Y}_{mi} \mid g_{mi}^1, \cdots, g_{mi}^n\right] \geq \underbrace{-\tau(t) - \mathbb{E}\left[\sum_{m=1}^M \widetilde{Y}_{mi} \mid g_{mi}^1, \cdots, g_{mi}^n\right]}_{(A)} \mid \mathcal{E}_1\right\}$$

Recall that $\mathbb{E}\left[\widetilde{Y}_{mi} \mid g_{mi}^1, \cdots, g_{mi}^n\right] = \frac{p}{2B+2\beta} \frac{1}{n}\sum_{j=1}^n g_{mi}^j$. We have

$$(A) \mid \mathcal{E}_1 = -\tau(t) - \frac{p}{2B+2\beta}\sum_{m=1}^M \frac{1}{n}\sum_{j=1}^n g_{mi}^j \mid \mathcal{E}_1$$

$$\geq -\tau(t) - \mathbb{E}\left[\frac{p}{2B+2\beta}\sum_{m=1}^M \frac{1}{n}\sum_{j=1}^n g_{mi}^j\right] - \frac{pc_0(n,p)}{4(B+\beta)}\sqrt{M}$$

$$= -\tau(t) - \sum_{m=1}^M \mathbb{E}\left[\widetilde{Y}_{mi}\right] - \frac{pc_0(n,p)}{4(B+\beta)}\sqrt{M}$$

$$\geq -\tau(t) - \frac{p}{2(B+\beta)}\sum_{m=1}^M g_{mi} - \frac{pc_0(n,p)}{4(B+\beta)}\sqrt{M}$$

Recall that $\nabla_i F(w(t)) < 0$. When $|\nabla_i F(w(t))| \geq \frac{2(B+\beta)\tau(t)}{Mp} + \frac{c_0(n,p)}{\sqrt{M}}$, we get

$$\mathbb{P}\left\{\sum_{m=1}^M \widetilde{Y}_{mi} \geq -\tau(t) \mid \mathcal{E}_1\right\} \leq \mathbb{P}\left\{\sum_{m=1}^M \widetilde{Y}_{mi} - \mathbb{E}\left[\sum_{m=1}^M \widetilde{Y}_{mi} \mid g_{mi}^1, \cdots, g_{mi}^n\right] \geq \frac{pc_0(n,p)}{4(B+\beta)}\sqrt{M} \mid \mathcal{E}_1\right\}$$

$$\leq \exp\left(-\frac{p^2 c_0^2(n,p)}{8(B+\beta)^2}\right)$$

$$\leq \frac{3-5c}{6},$$

The remaining proof also follows the arguments in Theorem 3.

$\square$

**Proof of Corollary 5 (Bounded Gradient Convergence Rate).** This proof follows from Theorem 4. We also consider two cases here.

First, when the system adversary is adaptive or the system adversary is static but with $\tau(t) \leq \frac{2}{p^2} \log \frac{6}{c}$, plug in $|F_i(w(t))| \geq \frac{2(B+\beta)\tau(t)}{Mp} + \frac{c_0(n,p)}{\sqrt{M}}$, we get

$$\frac{1}{T} \sum_{t=0}^{T-1} \mathbb{E}\left[\|\nabla F(w(t))\|_1\right] \leq \frac{1}{c} \left[ \frac{(F(w(0)) - F^*)\sqrt{d}}{\sqrt{T}} + \frac{L\sqrt{d}}{2\sqrt{T}} + 2d\frac{c_0(n,p)}{\sqrt{M}} + 4d\frac{(B+\beta)\sum_{t=0}^{T-1}\tau(t)}{pTM} \right].$$

Second, when the system adversary is static with $\tau(t) > \frac{2}{p^2} \log \frac{6}{c}$, plug in $|\nabla_i F(w(t))| \geq \frac{3(B+\beta)\tau(t)}{M} + \frac{c_0(n,p)}{\sqrt{M}}$, we get

$$\frac{1}{T} \sum_{t=0}^{T-1} \mathbb{E}\left[\|\nabla F(w(t))\|_1\right] \leq \frac{1}{c} \left[ \frac{(F(w(0)) - F^*)\sqrt{d}}{\sqrt{T}} + \frac{L\sqrt{d}}{2\sqrt{T}} + 2d\frac{c_0(n,p)}{\sqrt{M}} + 6d\frac{(B+\beta)\sum_{t=0}^{T-1}\tau(t)}{TM} \right].$$

$\square$

# E  EXPERIMENT DETAILS

## E.1  DATASETS AND PREPROCESSING

- **MNIST LeCun et al. (2009).** MNIST contains $60,000$ training images and $10,000$ testing images of 10 classes.
- **CIFAR-10 Krizhevsky et al. (2009).** CIFAR-10 contains $50,000$ training images and $10,000$ testing images of 10 classes.

**Implementation.** We build our codes upon PyTorch Paszke et al. (2019). We run all the experiments with 4 GPUs of type Tesla P100 and 1 GPU of type RTX 3060.

## E.2  PARAMETERS

**Communication rounds**: $500$ for both datasets in the section of client sampling. For the other sections, $80$ and $300$ communication rounds for MNIST and CIFAR-10, respectively.

**Dataset partition:** Clients' local datasets are evenly partitioned into balanced subsets. However, the distributions are non-IID since they follow Dirichlet distribution with a concentration $\alpha$.

We consider a constant learning rate in all cases, and the choices are tuned through grid search. Specifically, $\eta \in \{0.0001, 0.001, 0.01, 0.1\}$, $B \in \{0.001, 0.01, 0.1, 1\}$. Although our theory indicates the algorithm is not sensitive to mini-batch size, we set a large batch size $n = 256$ for both datasets.

| | Universal | | | $\beta$-Stochastic Sign SGD | FedAvg |
|---|---|---|---|---|---|
| | Learning Rate $\eta$ | Mini-batch Size $n$ | Hidden Units | $B$ | Local Epochs |
| MNIST | 0.01 | 256 | 64 | 0.01 | 1 |
| CIFAR-10 | 0.01 | 256 | 200 | 0.01 | 1 |

Table 4: Hyperparameters

## E.3  BYZANTINE BASELINE COMPARISONS

In this section, we reuse the network model and parameter settings in Table 4, set local epoch to be 1 for non-signed aggregation rules. We evaluate the algorithms on MNIST dataset and partition in a same manner as Appendix.E.1. We consider a total of 100 clients under full-participation with 20 Byzantine clients. The aggregation-rule-specific parameters are illustrated in the following part. All the experiment results are collected with 5 repetitions.

We compare our $\beta$ stochastic sign compressor with Krum Blanchard et al. (2017), geometric median Chen et al. (2017), centered clipping Karimireddy et al. (2021) under three adversary models, including label flipping, inner product manipulation Xie et al. (2020), the "A little is enough" Baruch et al. (2019). Following Karimireddy et al. (2021), $\tau$ is set to be 10 in centered clipping since momentum is switched off. We first illustrate the adversary models below:

- **Label flipping**: Suppose original label is $x$, the adversary will replace it with $9 - x$;
- **Inner Product Manipulation**: The adversaries send $-\frac{\gamma}{|\mathcal{N}|} \sum_{i \in \mathcal{N}} \nabla f(\boldsymbol{w}_i)$, instead of honest messages, to mislead the parameter server, where $\epsilon$ is the strength of the adversary. Let $\gamma = 0.1$.
- **A Little is Enough**: The adversaries estimate the benign clients' mean $\mu_{\mathcal{N}}$ and standard deviation $\sigma_{\mathcal{N}}$. Then, they will construct new messages as $\mu_{\mathcal{N}} + z\sigma_{\mathcal{N}}$ and upload to the parameter server, where $z$ is the strength of the adversary. We choose $z$ according to Baruch et al. (2019):

$$z = \max_z \left( \Phi(z) < \frac{M - s}{M} \right),$$

where $z = \lfloor \frac{M}{2} + 1 \rfloor - |\mathcal{B}(t)|$, and $\Phi$ is the cumulative distribution function of standard normal distribution. For us, $z \approx 0.5$.

In Section 7, we present the performance of our compressor under flipping sign attacks. For sign-bit messages, this is the worst-case scenario as adversaries' messages cannot escape a binary value. Otherwise, it will be detected by PS and filtered out.

We consider a milder condition than the sign flipping attacks for a fair of competition. We allow adversaries to manipulate the mini-batch stochastic gradient but assume an honest compressor that will send out the correctly compressed corrupted messages to PS.

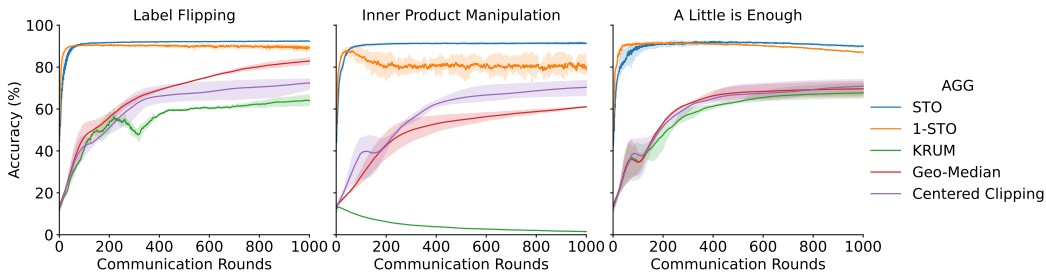

Figure 4: Comparisons with baselines: Krum, Geometric Median, Centered Clipping under Label Flipping, Inner Product Manipulation, and the "A Little is Enough" Adversaries, where 1-STO refers to our $\beta$ stochastic sign compressor with $\beta = B = 0.01$.

Throughout the experiments, it is observed that our $\beta$ stochastic sign compressor outperforms all other baseline algorithms when $\beta = 0$ or $\beta = B$. Notably, our compressor saves up to 31x communications and is differentially private when $\beta > 0$.

