# OpenReview forum: "$\beta$-Stochastic Sign SGD: A Byzantine Resilient and Differentially Private Gradient Compressor for Federated Learning"
_ICLR.cc/2023/Conference — Submitted to ICLR 2023_

### Official Review · Reviewer_nUrq · 2022-10-24

**Confidence:** 4
**Correctness:** 2
**Technical Novelty And Significance:** 2
**Empirical Novelty And Significance:** 2
**Recommendation:** 3

**Clarity, Quality, Novelty And Reproducibility:**

- This paper is not well-written.
- Several claims are not well-supported.
- The novelty of this paper is limited.
- There are almost no concerns about reproducibility.

**Strength And Weaknesses:**

**Strength**: The main idea of randomly flipping the sign is easy to understand and makes sense. The proposed method is not hard to implement.

---
**Weaknesses**:
1. The novelty of this work is limited since the proposed method seems to be a combination of stochastic sign SGD and random flipping.
2. The $L_1$ norm of gradients can be directly bounded by $\sum_{i\in[d]}B_i$ according to Assumption 3. It seems that the constant terms in the right-hand side of (5) and (6) are very likely to be not smaller than the trivial upper bound $\sum_{i\in[d]}B_i$. Could the authors compare the results in Theorem 5 with this?
3. The empirical results are not solid enough. The proposed method is only compared with SignSGD and FedAvg. Comparing $\beta$-stochastic sign SGD with more Byzantine-resilient methods is required.
4. This paper is not well-written.
  - $\beta$ is frequently used before being formally defined in the abstract and the introduction. Besides, $d$ appears in the statement of contributions (page 2) without any definition.
  - It is ambiguous what the notation $\nabla F_i(\cdot)$ means. Does it mean the partial derivative w.r.t. the $i$-th variable (i.e., the $i$-coordinate in the gradient)? If so, it is highly suggested to use $\nabla_i F(\cdot)$ or $(\nabla F(\cdot))_i$ since the original notation is very likely to be confused with $\nabla f_m(\cdot)$, which denotes the gradient of function $f_m(\cdot)$.
  - It is highly suggested to explicitly clarify which norm the notation $||\cdot||$ in Assumption 2 denotes.


**Summary Of The Paper:**

In this paper, the authors propose a new communication compression method called $\beta$-stochastic sign SGD, which is Byzantine-resilient and differentially private. The authors provide theoretical results about differential privacy and convergence. Besides, the proposed method is empirically compared with SignSGD and FedAvg.

**Summary Of The Review:**

Although the proposed method is easy to implement, given the concerns about novelty, quality, and readability, this work is currently below the bar of ICLR.

---

> ### Author Response · Authors · 2022-11-18
> **Novelty, trivial bound, Byzantine empirical evidence, presentation, and beyond.**
>
>  We thank Reviewer nUrq upfront for his time and efforts in our work.
>
>  On your question *"The novelty of this work is limited since the proposed method seems to be a combination of stochastic sign SGD and random flipping."*
>
>  **Our response:** Thank you for sharing your concerns on novelty. However, we need to clarify that Proposition 2 is only in hindsight such that we decompose our $\beta$ compressor into a non-DP compressor and an extra random flipping mechanism for the readers to parse our DP quantification. We do not want the readers to overinterpret this decomposition and think our work is merely achieved through random flipping.
>
> So we want to note that we propose a $\beta$ stochastic sign compressor, which is a one-step enablor of DP, communication efficiency, and Byzantine resilience. On the other hand, more discussions on the technical contributions and the relations between our work and [Jin et al. 2020] can be found in the general response and a point-by-point comparison (Table 3 in Appendix.(A)).
>
>  On your question *"The $L_1$ norm of gradients can be directly bounded by $\sum_{i\in[d]}B_i$ according to Assumption 3. It seems that the constant terms in the right-hand side of (5) and (6) are very likely to be not smaller than the trivial upper bound $\sum_{i\in [d]}B_i$. Could the authors compare the results in Theorem 5 with this?"*
>
> **Our response:** Thanks for sharing your concern with us. While we proceed, we want to first iterate that all of our residual terms can scale with $M$ in some order given some conditions. FL usually involves a large number of clients, so these terms will not dominate given a sufficient large number of $M$.
>
> Now, let us begin our derivation. We take the sub-Gaussian noise as an example, it can be verified a similar result holds for the heavy-tailed distribution.
> Let $n=\log M\in\Omega(\log M)$ (Remark 1 (3)). As Eq. (5) and (6) differ only a constant multiplier in the last term, without loss of generality, let us look at Eq. (6) only. The residual terms become
>
> $$\begin{align*}
>     &2d\frac{c_0}{\sqrt{M}}+4d(B+\beta)\exp(-\frac{n}{2})+6d\frac{(B+\beta)\sum_{t=0}^{T-1}\tau (t)}{TM}\\
>     &\le 2d\frac{c_0}{\sqrt{M}}+4d\frac{(B+\beta)}{\sqrt{M}}+6d\frac{(B+\beta)\sum_{t=0}^{T-1}\tau (t)}{TM}.
> \end{align*}
> $$
>
> As stated in the theorem statement, we assume a gap between the true gradient upper bound and clipping threshold, i.e., $\sum_{i\in[d]}B_i\le dB$. A trivial observation from the above equation is that it will decrease w.r.t. client number $M$. In other words, the bound becomes ${\cal{O}}(1/\sqrt{M})$ and will disappear as $M\to\infty$ (Remark 1 (4)). By contrast, the trivial bound $dB$, scaling with the model size $d$, will always stay as a constant.
>
> On your question *The empirical results are not solid enough. The proposed method is only compared with SignSGD and FedAvg. Comparing $\beta$-stochastic sign SGD with more Byzantine-resilient methods is required.""*
>
> **Our response:** Thank you for your valuable suggestions. We have added more Byzantine-resilient experiments in Appendix.(E3) for a comparison with some baseline algorithms, including Krum [1], Geometric Median [2], Centered Clipping [3], and some adversarial models, including label-flipping attack, inner product manipulation [4], and the "a little is enough" [5]. The overall performance is promising, our $\beta$ Compressor outperforms all the other benchmark algorithms in all of the adversarial models.
>
> [1] Blanchard, P., El Mhamdi, E. M., Guerraoui, R., \& Stainer, J. (2017). Machine learning with adversaries: Byzantine tolerant gradient descent. Advances in Neural Information Processing Systems, 30.
>
> [2] Chen, Y., Su, L., & Xu, J. (2017). Distributed statistical machine learning in adversarial settings: Byzantine gradient descent. Proceedings of the ACM on Measurement and Analysis of Computing Systems, 1(2), 1-25.
>
> [3] Karimireddy, S. P., He, L., & Jaggi, M. (2021, July). Learning from history for byzantine robust optimization. In International Conference on Machine Learning (pp. 5311-5319). PMLR.
>
> [4] Xie, C., Koyejo, O., & Gupta, I. (2020, August). Fall of empires: Breaking byzantine-tolerant sgd by inner product manipulation. In Uncertainty in Artificial Intelligence (pp. 261-270). PMLR.
>
> [5] Baruch, G., Baruch, M., & Goldberg, Y. (2019). A little is enough: Circumventing defenses for distributed learning. Advances in Neural Information Processing Systems, 32.

---

> > ### Author Response · Authors · 2022-11-18
> > **Presentation**
> >
> >
> > "This paper is not well-written.
> >
> > * $\beta$ is frequently used before being formally defined in the abstract and the introduction. Besides, $d$ appears in the statement of contributions (page 2) without any definition.
> > * It is ambiguous what the notation $\nabla F_i(\cdot)$ means...
> > * It is highly suggested to explicitly clarify which norm the notation $\\|\cdot\\|$ in Assumption 2 denotes."
> >
> > **Our response:** Thank you for sharing your concerns on presentations with us. We are sorry to confuse you on the nomenclatures. We hope the revision will meet your expectation. First, we respond to your concerns point-by-point
> > * While we agree with the reviewer that all the parameters should be stated first and used then, $\beta$ is the defining feature of our stochastic sign compressor, and it is not possible to present all the details, for example, the probability distribution in a limited space abstract. However, for $d$, we are more than glad to do it and thank the reviewer's reminder. The statement has been added in the revision and highlighted in blue.
> > * Yes, $i$ is the gradient coordinate. We take the reviewer's suggestion and revise all the related terms as $\nabla_i F(\cdot)$.
> > * It is $l_2$ norm, and we added a subscript 2. Please refer to it in the revision.

---

> ### Comment · Reviewer_nUrq · 2022-11-18
> **My concern about the trivial bound remains**
>
> I appreciate the authors' detailed response and their effort to improve this work. However, my major concern about the trivial bound remains.
>
> In the response, the authors seem to have omitted the effect of $p$. Specifically, since $c_0 \geq \sqrt{\frac{8(B+\beta)^2}{p^2}\log\frac{6}{3-5c}}$, the term $$2d\frac{c_0}{\sqrt{M}}\geq \frac{4\sqrt{2}d(B+\beta)}{\sqrt{M}\cdot p}\sqrt{\log\frac{6}{3-5c}} > \frac{4\sqrt{2}(dB+d\beta)}{\sqrt{Mp}\cdot\sqrt{p}}.$$
> Besides, Eq(6) holds only when $\tau(t)>\frac{2}{p^2}\log\frac{6}{c}$. Therefore, the term $$6d\frac{(B+\beta)\sum_{t=0}^{T-1}\tau(t)}{TM} > 6d\frac{(B+\beta)\cdot(2\log\frac{6}{c})}{(Mp)\cdot p}.$$
> Note that $Mp$ is the expected number of clients participating in each round. Given the overhead of the server, $Mp$ is usually kept not too large. Furthermore, $p$ will be close to $0$ when the total client number $M$ is large. Therefore, the theorem does not present a better result than the trivial bound $\sum_{i\in[d]}B_i \leq dB$.
>
> In summary, the bound presented in Theorem 4 is not better than the trivial bound $\sum_{i\in[d]}B_i$ directly from Assumption 3 (bounded gradient assumption). Although I appreciate the authors' patience in improving this work, the response is not satisfactory given the reasons above. Therefore, my rating keeps unchanged.

---

> > ### Author Response · Authors · 2022-11-18
> > **Author's Response to Reviewer nUrq on the trivial bound continued.**
> >
> > *"Note that $Mp$ is the expected number of clients participating in each round. Given the overhead of the server, $Mp$ is usually kept not too large. Furthermore, $p$ will be close to 0 when the total client number is large. Therefore, the theorem does not present a better result than the trivial bound ."*
> >
> > **Our response:** We thank Reviewer nUrq for the prompt response and for sharing further concern with us.
> >
> > While we agree with the reviewer that one intention behind partial client sampling is to save communication overhead, we want to note that it is not practical to deem the term $Mp$ as $\Theta(1)$ for a scaling FL system and this is a misunderstanding.
> >
> > Although the argument is lacking in FL literature, we use the coupon collector's problem to counteract that assumption.
> >
> > First, let us recall the coupon collector's problem:
> >
> > *There are $n$ different coupons. Each purchase comes with a random coupon, and each coupon is equal likely to appear. How many coupons are needed to collect all the coupons?*
> >
> > Let us denote the number of coupon purchases by $N$, then by a standard argument, we will arrive at the conclusion that
> >
> > $$\mathbb{E}[N]=n\sum_{i=1}^n\frac{1}{i}\approx n\int_{i=1}^n\frac{1}{x}\mathrm{d}x=n\log n.$$
> >
> > To see this, we can also invoke the fact that the Harmonic number
> >
> > $$H_n=\sum_{i=1}^n\frac{1}{i}=\log(n)+\gamma+\cal O (\frac{1}{n}),$$ where $\gamma$ is the Euler-Mascheroni constant [1].
> >
> > [1] http://www.cs.cmu.edu/~odonnell/toolkit13/lecture01-anonymous.pdf
> >
> >
> >
> > Now, back to your concerns. Let us assume $Mp=N=\Theta(1)$. The equivalent number of coupons will be $\\#=\frac{M}{N}$.
> >
> > By the scaling law of coupon's collector theorem, the time to encounter all the coupons (clients) in expectation will be at the order of $\Theta(\frac{M}{N}\log(\frac{M}{N}))=\Theta(M\log (M))$, this follows because $N=\Theta(1)$. In our case, this can be interpreted as the time to sample all the clients and thus the samples in the training set, i.e., one epoch training time. This is extremely large as $M\to\infty$ for only one training epoch.
> >
> > To numerically appreciate the results, let us consider two cases both with an average number of sampled clients $Mp=10$, which is quite small, but with client number $M_1=1000$ and $M_2=10000$.
> >
> > Through the above argument, we can see the equivalent coupons number is $\\#_1=100$ and $\\#_2=1000$. We denote the corresponding encountering time as $T_1$ and $T_2$, it follows that
> >
> > $$\mathbb{E}[T_1]=100\sum_{i=1}^{100}\frac{1}{i}\approx 100\log 100=460.517;$$
> >
> > $$\mathbb{E}[T_2]=1000\sum_{i=1}^{1000}\frac{1}{i}\approx 1000\log 1000=6907.8.$$
> >
> > More importantly, the expected training iterations are for just one training epoch. It is known that learning takes time and involves multiple epochs of training. In light of the argument above, we cannot simply let $Mp=\Theta(1)$ for a scaling FL system.
> >
> > That said, the usual practice is to keep $p=\Theta(1)$ instead of $Mp$. This is verified numerically in the existing literature [2,3], which enables us to go back to our argument and talk about the asymptotic in terms of client number $M$.
> >
> >
> > We hope this response addresses your concerns about this bound.
> >
> >
> > [2] McMahan, B., Moore, E., Ramage, D., Hampson, S., \& y Arcas, B. A. (2017, April). Communication-efficient learning of deep networks from decentralized data. In Artificial intelligence and statistics (pp. 1273-1282). PMLR.
> >
> > [3] Philippenko, C., & Dieuleveut, A. (2020). Bidirectional compression in heterogeneous settings for distributed or federated learning with partial participation: tight convergence guarantees. arXiv preprint arXiv:2006.14591.

---

> > > ### Comment · Reviewer_nUrq · 2022-11-18
> > > **Follow-up reply on the trivial bound**
> > >
> > > I thank the authors' response. I would like to clarify that $(Mp)$ was considered as a whole due to the limited computation power and bandwidth of the server. Actually, the total client number $M$ is determined by the FL task itself. We can properly set the total iteration number $T$ to achieve the desired error. However, $M$ cannot be manually set in a specific FL task. Analyzing the theoretical bounds as $M\rightarrow\infty$ can help to explore the speed-up property of an algorithm. However, it is not an excuse to omit the case of small client number $M$. Actually, the client number is not always that large in FL. For example, in cross-silo federated learning, $M$ is typically ranging from $2$ to $100$ [1].
> > >
> > > To improve this work, I recommend the authors provide a specific condition and compare their theoretical results to the trivial bound under the condition with rigorous proof. It would make readers more clear about the theoretical results and help to evaluate this work.
> > >
> > > [1] Peter Kairouz, H Brendan McMahan, Brendan Avent, Aurelien Bellet, Mehdi Bennis, Arjun Nitin Bhagoji, Kallista Bonawitz, Zachary Charles, Graham Cormode, Rachel Cummings, et al. Advances and open problems in federated learning. Foundations and Trends in Machine Learning, 14(1–2):1–210, 2021.

---

> > > > ### Author Response · Authors · 2022-11-19
> > > > **Response to the follow-up question**
> > > >
> > > > *To improve this work, I recommend the authors provide a specific condition and compare their theoretical results to the trivial bound under the condition with rigorous proof. It would make readers more clear about the theoretical results and help to evaluate this work.*
> > > >
> > > > We thank Reviewer nUrq for the valuable suggestions and giving us a chance to improve our work.
> > > >
> > > > Before diving into the discussion of the trivial bound, we want to first talk about some presentational revisions:
> > > >
> > > > * We want to thank Reviewer nUrq again for letting us clarify the FL system that our work is fitting into, and we apologize for not doing so in the first place. This has been highlighted in blue in the revision. We consider a cross-device FL system, where a large number of workers are involved [1].
> > > > * We replaced all the $c_0$ with $c_0(n,p)$ to avoid any confusion in assuming $c_0=\Theta(1)$. Rather, it is a function of mini-batch size $n$, and sampling probability $p$.
> > > >
> > > > Now, we formally discuss the scenario, where $Mp=N=\Theta(1)$ and $N\in\mathbb{N}^+$.
> > > >
> > > > It turns out that we need to impose some restrictions on the sample size $N$ and mini-batch size $n$ so that the residual terms are within the radius of the trivial bound.
> > > >
> > > > Let us define $\beta=\gamma B\~(\gamma \ge 0)$, we assume non-negative weights $w_1,w_2,w_3~(w_1+w_2+w_3=1)$ on three residual terms (sub-Gaussian noise case), and assume $M$ is quite large, whereas $p$ is quite small.
> > > >
> > > > Since $c_0(n,p):=\max\left\\{\sqrt{\frac{8 \sigma^2}{n}\log(\frac{6}{c})},\sqrt{\frac{8(B+\beta)^2}{p^2}\log \frac{6}{3-5c}}\right\\}$, the second element $\sqrt{\frac{8(B+\beta)^2}{p^2}\log \frac{6}{3-5c}}$ dominates under the above assumption.
> > > >
> > > > We now abbreviate $c_0(n,p)$ as  $c_0$ for ease of presentation.
> > > >
> > > > The $c_0$ residual term is
> > > >
> > > > $$\frac{2dc_0}{c\sqrt{M}}=\frac{4d(B+\beta)}{cp}\sqrt{\frac{2}{M}\log \frac{6}{3-5c}}=\frac{4dB(1+\gamma)}{cp}\sqrt{\frac{2}{M}\log \frac{6}{3-5c}}.$$
> > > >
> > > >
> > > >
> > > > Now, let the above term fall into the radius of the trivial bound,
> > > >
> > > > $$\frac{4dB(1+\gamma)}{cp}\sqrt{\frac{2}{M}\log \frac{6}{3-5c}}\le w_1 dB\Rightarrow p\ge\frac{4(1+\gamma)}{cw_1}\sqrt{\frac{2}{M}\log \frac{6}{3-5c}}.$$
> > > >
> > > > Replace $p$ with $\frac{N}{M}$, we get
> > > >
> > > > $$N\ge4(1+\gamma)\sqrt{2M}\frac{\sqrt{\log \frac{6}{3-5c}}}{cw_1}.$$
> > > >
> > > > For Eq. (6), we restate here to be more precise, this is for **static adversaries** with $\tau(t)>\frac{2}{p^2}\log(\frac{6}{c})$. In Remark 2 (1), we note that *the lower bound requirement on $\tau(t)$ might be an artifact of our analysis in simplifying the boundary case deviation.*
> > > >
> > > > With a small $p$ as in the assumption, it is quite likely that $N<\frac{2}{p^2}\log(\frac{6}{c})$ **Eq. (1)**, as $N$ is quite small when compared with $M$, and if so, we go back to Eq. (5) for static adversaries.
> > > >
> > > > To be more precise, for **Eq. (1)** to hold,
> > > >
> > > > $$N<\sqrt[3]{2M^2\log(\frac{6}{c})}.$$
> > > >
> > > > That said, we go back to Eq. (5) for both **static** and **adaptive** adversaries.
> > > >
> > > > $\frac{4d(B+\beta)\sum_{t=0}^{T-1}\tau(t)}{cTN}\le w_2dB\Rightarrow \sum_{t=0}^{T-1}\tau(t)\le \frac{cw_2TN}{4(1+\gamma)},$ i.e, $N\ge4(1+\gamma)\frac{\sum_{t=0}^{T-1}\tau(t)}{cw_2T}$.
> > > >
> > > > For the sub-Gaussian tail noise term,
> > > >
> > > > $$\frac{4d(B+\beta)\exp(-\frac{n}{2})}{c}\le w_3dB\Rightarrow n\ge2\log(\frac{cw_3}{4(1+\gamma)}).$$
> > > >
> > > > Combining the results above, we conclude that
> > > >
> > > > $N\ge\frac{4(1+\gamma)}{c}\max\left\\{\frac{1}{w_1}\sqrt{2M\log\frac{6}{3-5c}},\frac{\sum_{t=0}^{T-1}\tau(t)}{w_2T}\right\\}$ and $n\ge2\log(\frac{cw_3}{4(1+\gamma)}).$
> > > >
> > > > The same kind of argument works for heavy-tailed noise and bounded gradient cases, and we skip them for now.
> > > >
> > > > [1] Peter Kairouz, H Brendan McMahan, Brendan Avent, Aurelien Bellet, Mehdi Bennis, Arjun Nitin Bhagoji, Kallista Bonawitz, Zachary Charles, Graham Cormode, Rachel Cummings, et al. Advances and open problems in federated learning. Foundations and Trends in Machine Learning, 14(1–2):1–210, 2021.

---

> > > > > ### Comment · Reviewer_nUrq · 2022-11-22
> > > > > **Follow-up reply and questions about the changed parts**
> > > > >
> > > > > I appreciate the authors' effort to improve this work. My comments about the latest convergence results are listed below.
> > > > >
> > > > > 1. I have roughly checked the added proof process in the response and it looks correct.
> > > > > 2. Theorem 4 combined with the added discussion says that when a bounded true gradient (Assumption 3) is assumed, $\mathbb{E}||\nabla F(w(t))||_1$ is guaranteed to be smaller than the assumed bound. However, it is still uncertain how small could $\mathbb{E}||\nabla F(w(t))||_1$ be guaranteed and how fast the upper bound decreases w.r.t. $N$, $M$ and $n$. Please note that even if the conditions of $N$ and $n$ are satisfied, the RHS of (5) and (6) could just be a little bit smaller than the trivial bound, which limits the contribution of Theorem 4.
> > > > > 3. In spite of the weakness discussed above, the quality of the convergence theory is generally better than the initial version.
> > > > > ---
> > > > >
> > > > > Besides, there are also some questions about the changed parts.
> > > > > 1. I fail to find the definition of $\mathcal{G}$ in Theorem 2. Could the authors help me find it?
> > > > > 2. It is assumed that $B=(1+\epsilon_0)B_0$ in Theorem 3 and it is suggested to let $B=\Delta_1/d$ in Remark 1. Do the two settings conflict with each other?

---

> > > > > > ### Author Response · Authors · 2022-11-24
> > > > > > **Response to the concerns on the changed parts**
> > > > > >
> > > > > > We thank Reviewer nUrq for appreciating our response and revision.
> > > > > >
> > > > > > For your concerns on the changed parts, we respond as follows:
> > > > > >
> > > > > > *1. I fail to find the definition of $\mathcal{G}$ in Theorem 2. Could the authors help me find it?*
> > > > > >
> > > > > > **Our response:** Sure, we are more than glad to help. We first restate the initial part of Theorem 2 here,
> > > > > > **Theorem 2. Let $g,g^\prime\in\cal G \subseteq \mathbb{R}^d,\ldots$** Therefore, $\cal G$, the set to which $g,g^\prime$ belong, is a subset of $R^d$, where $d$ is the gradient dimension.
> > > > > >
> > > > > > *2. It is assumed that $B=(1+\epsilon_0)B_0$ in Theorem 3 and it is suggested to let $B=\Delta_1/d$ in Remark 1. Do the two settings conflict with each other?*
> > > > > >
> > > > > > **Our response:** In the revision, the conditions for $\epsilon_0$ are different. In detail, we choose $\epsilon_0>\frac{\sigma}{B_0}$ in the case of sub-Gaussian noise, whereas $\epsilon_0>\frac{{M_{p^\prime}}^{\frac{1}{{p^\prime}}}}{B_0}~(p^\prime\ge 4)$ in the case of heavy-tailed noise.
> > > > > >
> > > > > > The expression $B=(1+\epsilon_0)B_0$ quantifies the gap between the clipping threshold $B$ and true gradient upper bound $B_0$. Since we do not specify the exact value of $\epsilon$, it is an inequality in some sense. For a better presentation, the conditions can be illustrated as $B>B_0+\sigma$ (sub-Gaussian noise) and $B>B_0+{M_{p^\prime}}^{\frac{1}{{p^\prime}}}~(p^\prime\ge 4)$ (heavy-tailed noise), we will change them in a future revision. In contrast,  $B=\Delta_1/d$ is a practical choice that depends on $\Delta_1$.
> > > > > >
> > > > > > In accordance with our response to Reviewer W1pz, how to choose a sound $B$ is challenging, and we leave a future direction. We refer interested readers to our response in the following link.
> > > > > >
> > > > > > https://openreview.net/forum?id=oVPqFCI1g7q&noteId=sxR_FMpS3Jl

---

> > > > > > > ### Comment · Reviewer_nUrq · 2022-11-24
> > > > > > > **Reply**
> > > > > > >
> > > > > > > I thank the authors for their reply. I think that the authors might have misunderstood my questions. I apologize for not making it clear before and would like to clarify it.
> > > > > > >
> > > > > > > 1. Actually, what I meant was that the notation $\mathcal{G}$ seems to appear in Theorem 2 without a specific definition. Does it denote an arbitrary subset of $\mathbb{R}^d$?
> > > > > > >
> > > > > > > 2. What I meant was that $\Delta_1/d$ seems smaller than $B_0$. If so, the assumption that $B=(1+\epsilon_0)B_0$ will not hold if we set $B=\Delta_1/d$. However, I failed to estimate the value of $\Delta_1$ since it depends on the set $\mathcal{G}$.

---

> > > > > > > > ### Author Response · Authors · 2022-11-24
> > > > > > > > **Corrections**
> > > > > > > >
> > > > > > > > We want to thank Reviewer nUrq for the comments. As we are not allowed to update the manuscript at this point, we want to state some corrections and will update them in a future version.
> > > > > > > >
> > > > > > > > *Actually, what I meant was that the notation $\mathcal{G}$ seems to appear in Theorem 2 without a specific definition. Does it denote an arbitrary subset of $\mathbb{R}^d$?*
> > > > > > > >
> > > > > > > > **Our response:** Yes, we do not impose any restrictions on $\cal G$, it can be an arbitrary subset.
> > > > > > > >
> > > > > > > > *What I meant was that $\Delta_1/d$ seems smaller than $B_0$. If so, the assumption that $B=(1+\epsilon_0)B_0$ will not hold if we set $B=\Delta_1/d$. However, I failed to estimate the value of $\Delta_1$ since it depends on the set $\mathcal{G}$.*
> > > > > > > >
> > > > > > > > **Our response:** This is a good catch. We thank the reviewer for pointing this out and apologize for the rush revision.
> > > > > > > >
> > > > > > > > In fact, $l_1$ sensitivity $\Delta_1$ here is an abuse of notations, and we want to correct the theorem, corollary, and remark here. The corrected ones do not depend on $\Delta_1$ and should address the reviewer's concern automatically.
> > > > > > > >
> > > > > > > > Our mechanism $\cal M_{\it B,\beta}$ can be viewed as a mapping $\mathbf{g}\to \\{-1,1\\}^d$. Meanwhile, our objective is to take care of the DP of stochastic gradients.
> > > > > > > >
> > > > > > > > In this case, $l_1$ sensitivity is (Definition 3.1 [1])
> > > > > > > >
> > > > > > > > $\Delta \cal M=\max_{\\|x-y\\|_1=1}|\cal M(x)-\cal M(y)|$.
> > > > > > > >
> > > > > > > > Since we are talking about the DP of stochastic gradients, the gradient pairs we are interested in are only the so-called **"adjacent inputs"** (i.e., $\\|g-g^\prime\\|\le 1$) rather than any arbitrary gradient pairs.
> > > > > > > >
> > > > > > > > That said, theorem 2 in the revision should stay the same as theorem 3 in the original draft, and we restate it as follows:
> > > > > > > >
> > > > > > > > **Theorem 2**
> > > > > > > >
> > > > > > > > Let $\beta>0$ and $\cal G\subseteq \mathbb{R}^d$. $\cal M_{\it B,\beta}$ is - $\max_{\mathbf{g}\in \cal G}\sum_{i=1}^d \log\left({1 +  \frac{1}{\beta + \mathsf{dist}\left({g_i, \cal C_{\it B}}\right)}}\right)$-differentially private on $\cal G$.
> > > > > > > >
> > > > > > > > The reason why $\cal G$ is presented because we care about $\mathsf{dist}\left({g_i, \cal C_{\it B}}\right)$.
> > > > > > > >
> > > > > > > > The corrections of Corollary 1 and Remark 1 follow,
> > > > > > > >
> > > > > > > > **Corollary 1**
> > > > > > > >
> > > > > > > >  Given the same definitions as in Theorem 2, $\cal M_{\it B,\beta}$ is $\left({\frac{1}{\beta}}\right)$-DP.
> > > > > > > >
> > > > > > > > **Remark 1 should be corrected as follows,**
> > > > > > > >
> > > > > > > > **Remark 1:**   Theorem 2 gives a finer characterization of the differential privacy preserved by $\cal M_{B,\beta}$ when $\beta>0$. Unfortunately, this maximum is often hard to find, so Corollary 1 tells us that we can get rid of the order $\cal O({d})$ *by looking at the upper bound of Theorem 2*.
> > > > > > > >
> > > > > > > >
> > > > > > > >
> > > > > > > > The corrections in the proofs are easy to follow by replacing $\Delta_1$ with 1. We apologize again for the confusion it invites.
> > > > > > > >
> > > > > > > > [1] Dwork, C., & Roth, A. (2014). The algorithmic foundations of differential privacy. *Foundations and Trends® in Theoretical Computer Science*, *9*(3–4), 211-407.

---

> > > > > > > > > ### Comment · Reviewer_nUrq · 2022-11-26
> > > > > > > > > **Comment on the corrections**
> > > > > > > > >
> > > > > > > > > I appreciate the authors' prompt correction. Before making my final evaluation, I would like to ask more questions about the definition of "adjacent inputs".
> > > > > > > > >
> > > > > > > > > I understand that the initial definition of differential privacy (DP) leaves the definition of "adjacent inputs" open. It is natural to consider two vectors $\mathbf{g}$ and $\mathbf{g}'$ that are close to each other. However, I am curious why the RHS of the inequality is the constant $1$ in the definition ($||\mathbf{g}-\mathbf{g}'||\leq 1$). Specifically, when the loss function is multiplied by a constant $k$, both sides on equation (5) and (6) (in Theorem 4) will also be multiplied by $k$. That is to say, the convergence result remains the same. However, the adjacency ($||\mathbf{g}-\mathbf{g}'||\leq 1$) would significantly change.
> > > > > > > > >
> > > > > > > > > Could the authors provide some references or discuss about this? Besides, it is important to make the definition of adjacent inputs explicit in the main text.

---

> > > > > > > > > > ### Author Response · Authors · 2022-12-13
> > > > > > > > > > **Follow-up response**
> > > > > > > > > >
> > > > > > > > > > We thank reviewer nUrq for sharing the concerns.
> > > > > > > > > >
> > > > > > > > > > The definition of adjacent inputs is in accordance with the definition of differential privacy. One useful reference would be [1].
> > > > > > > > > >
> > > > > > > > > > Dwork, C., & Roth, A. (2014). The algorithmic foundations of differential privacy. Foundations and Trends® in Theoretical Computer Science, 9(3–4), 211-407.

---

### Official Review · Reviewer_W1pz · 2022-10-25

**Confidence:** 4
**Clarity, Quality, Novelty And Reproducibility:** 1. The reviewer does not quite follow…
**Correctness:** 4
**Technical Novelty And Significance:** 2
**Empirical Novelty And Significance:** 2
**Recommendation:** 6

**Strength And Weaknesses:**

Strength:
1. The paper shows that $\beta$-stochastic sign SGD achieves communication efficiency, differential privacy, and Byzantine resilience simultaneously.
2. The proposed algorithm is very practical, easy to implement, and have good performance in MNIST and CIFAR-10 datasets in experiments.

Weakness:
1. The technical contribution in proving differential privacy of $\beta$-stochastic sign SGD is not that significant.
2. The assumptions for bounded gradients and sub-Gaussian element-wise gradient noise are kind of strong for federated learning or requires more justification, and these two assumptions again make the technical contributions less significant.
3. More discussions with existing works such as [1] are needed for comparing the final error bounds for Byzantine resilience.
4. Basically, the reviewer gets the idea that $\beta$-stochastic sign SGD achieves two-fold benefits, but wants to see some comparisons with existing works in each dimension, so that the paper is more persuasive in that $\beta$-stochastic sign SGD achieves performances matching the state of the art of at least comparable, and the technical challenges therein is non-trivial given that the algorithm itself is not that novel.
5. In the experiments, it seems that the authors tune the algorithm to find good parameters $B$, which makes the algorithm less practical since it means that this algorithm needs to be run several times to achieve good performance (this will contradict to the target benefits of the algorithm), so a good estimate of a favorable $B$ would be helpful.
6. Can the authors add some experiments to compare the performance of $\beta$-stochastic sign SGD with other existing algorithms either designed for DP or Byzantine resilience?


[1] Karimireddy, S. P., He, L., & Jaggi, M. (2021, September). Byzantine-Robust Learning on Heterogeneous Datasets via Bucketing. In International Conference on Learning Representations.

**Summary Of The Paper:**

This paper analyzes an optimization algorithm $\beta$-stochastic sign SGD. The authors proves that $\beta$-stochastic sign SGD is differentially private in the case of $\beta > 0$. The authors also show that $\beta$-stochastic sign SGD achieves Byzantine resilience under some problem assumptions.

**Summary Of The Review:**

This paper has a very interesting finding that $\beta$-stochastic sign SGD achieves Byzantine resilience and differential privacy simultaneously. It will help the reviewer to understand the contributions of this paper better if there were more performance comparisons of $\beta$-stochastic sign SGD with existing methods designed for DP or Byzantine resilience only, whether they are comparable theoretically or in experiments in each dimension, and maybe some more explanations on the significance of the presented results.

---

> ### Author Response · Authors · 2022-11-18
> **Novelty, gradient distribution, concurrent work comparisons, and more**
>
> We thank Reviewer W1pz upfront for appreciating our $\beta$ compressor and his valuable suggestions and comments on our work.
> #### Weaknesses
> On your question *"The technical contribution in proving differential privacy of $\beta$-stochastic sign SGD is not that significant."*
>
> **Our response:** We thank the reviewer for sharing his concerns on novelty with us. As we have pointed out in the general response, our $\beta$ stochastic sign compressor and sto-sign compressor in [Jin et al. 2020] are admittedly structural alike however depart in significant ways. We have taken out our statement *"Notably, for the special case when each of the stochastic gradients involved is bounded with known bounds, our gradient compressor with $\beta=0$ coincides with the compressor proposed in [Jin et al. 2020]"* throughout the draft. This is not correct as (1) [Jin et al. 2020] did not consider mini-batch gradient (2) [Jin et al. 2020] did not push out an explicit form of residual term both w.r.t. Byzantine resilience and weak signal strength. To further see this, we recommend the reviewer to refer to Table 3 in Appendix.(A).
>
> On your question *"The assumptions for bounded gradients and sub-Gaussian element-wise gradient noise are kind of strong for federated learning or requires more justification, and these two assumptions again make the technical contributions less significant."*
>
> **Our response:** We thank the reviewer for sharing concerns on the gradient distribution. To complete the case, we add the analysis for heavy-tailed gradient noise using Rosenthal-type inequality. The step-by-step proof can be found in the appendix of the revision. We here only present the assumption and the noisy term results. The heavy-tailed noisy term scales with $n$ in the order of $\cal O(\frac{1}{n^{\frac{p^\prime}{2}}})$, and it can become non-dominating as long as $n=\Omega(M^{\frac{1}{p^\prime}})$ for $p^\prime\ge4$.
>
> **Assumption 5 [Heavy-tailed noise]**
> Assume the stochastic gradient $\mathbf{g_m} (w)$ is an independent unbiased estimate of $\nabla f_{m}(w)$ that is coordinate-wise related to the gradient $\nabla f_m(w)$ as $\mathbf{g_{mi}}(w)=\nabla f_{mi}(w)+\mathbf{\xi_{mi}} ~ \forall\, i\in [d]$. Let $\mathbf{\xi_{mi}}$ be a zero-mean random variable, $\mathbb{E}{[\mathbf{\xi_{mi}}^{2}]}\le \sigma^2$, and $\mathbb{E}{|\mathbf{\xi_{mi}}|}^{p^{\prime}}\le M_{p^\prime}<\infty$ for $p^\prime\ge 4$ .
>
> On your question *"More discussions with existing works such as [1] are needed for comparing the final error bounds for Byzantine resilience."*
>
> **Our response:** Thanks for your suggestions. We first want to note some major differences between [1] and our paper from three aspects.
>
> * Problem-formulation-wise: our work is interested in solving a population risk minimization problem Eq. (1) over all the clients. In contrast, [1] is to minimize a population risk over only the good workers.
> * Byzantine-wise:
> 	*we incorporate client sampling in our work, enabling us to explore more worse-case adversaries that we call adaptive Byzantine adversaries. This kind of adversaries is a more stringent case as all the Byzantine clients will be admitted into the sampling set.
> 	* In addition, we allow Byzantine clients to change over time and collude a mobile attack. For example, even if we let Byzantine client number $\tau(t)=1$, it is possible that ${\cal{B}}=\cup_{t=1}^T{\cal{B}}(t)=[M]$, where $M$ is the number of clients, meaning that all clients can become Byzantine at least once during the whole training trajectory.
> 	* Although we only empirically verify the bit-flipping attack in our original draft, we note that this is the worst-case attack in the realm of sign-based compressors. More empirical comparisons with different attack strategies can be found in the revision.
> 	* Moreover,  [1] considers only (1) static and (2) immobile adversaries.
> * Theoretical-wise:
> 	* we consider a bit-compression mechanism to reduce the regular communications by 31x (by regular, we refer to the usual 32-bit float number,) which [1] does not.
> 	* Our rate is about $l_1$ norm of the gradient and will reach the first-order optimum given $M\to\infty$, which matches the practice as $M$ is usually large in FL [2].
> 	* By contrast, [1] does not consider gradient compression. The residual term ${\cal O }(c\delta\zeta^2)$ will not vanish given a sufficient large number of clients $M$.

---

> > ### Author Response · Authors · 2022-11-18
> > **Continued**
> >
> >
> > On your question *"Basically, the reviewer gets the idea that $\beta$-stochastic sign SGD achieves two-fold benefits, but wants to see some comparisons with existing works in each dimension, so that the paper is more persuasive in that $\beta$-stochastic sign SGD achieves performances matching the state of the art of at least comparable, and the technical challenges therein is non-trivial given that the algorithm itself is not that novel."*
> >
> > **Our response:** We thank the reviewer for giving us a chance to compare our results with the state-of-the-art theoretical results. When the residual terms are not considered, our result matches the centered, non-private, and adversary-free SGD with an order of $\cal O(\frac{1}{\sqrt{T}})$[1]. Unfortunately, we cannot achieve the speedup in convergence with more FL clients in the presence of non-IID data. This follows because, unlike IID datasets, non-IID datasets are not replicable across different clients. For the residual term, this is a fundamental issue of non-IID data as shown in [1]. Combining the residual term and asymptotic in terms of training iterations $T$ and client number $M$, our convergence rate becomes $\cal O(\frac{1}{\sqrt{T}}+\frac{1}{\sqrt{M}})$. Thus, it approaches the centralized SGD rate as $M\to \infty$ Luckily, $M$ is often quite large in FL [2].
> >
> > On your question *"In the experiments, it seems that the authors tune the algorithm to find good parameters $B$, which makes the algorithm less practical since it means that this algorithm needs to be run several times to achieve good performance (this will contradict to the target benefits of the algorithm), so a good estimate of a favorable $B$ would be helpful."*
> >
> > **Our response:** Thank you for your suggestions. This is indeed an interesting question. A naive way to find the stochastic gradient upper bound is to find the neural network lipschitz and plug it back into our compressor. However, [3] has shown that exact Lipschitz computation for neural network is NP-hard, showing that it is a challenging issue. For now, we leave it as a future direction and thus an open question.
> >
> > On your question *"Can the authors add some experiments to compare the performance of $\beta$-stochastic sign SGD with other existing algorithms either designed for DP or Byzantine resilience?"*
> >
> > **Our response:** Thank you for your suggestions. Yes! We have added more empirical evidence on Byzantine resilience for the reviewers and the community to evaluate our compressor's performance. This can be found in Appendix.(E3). Specifically, we compare our compressor subject to $\beta=0$ and $\beta=1$ with  some baseline algorithms, including Krum [4], Geometric Median [5], Centered Clipping [6], and some adversarial models, including label-flipping attack, inner product manipulation [7], and the "a little is enough" [8]. In all the results, our $\beta$ compressor with either $\beta=0$ or $\beta=B$ outperforms the other benchmark algorithms, corroborating its superiority. For DP experiments, as we have pointed out in the response to Reviewer YSJ7, our initial DP results scale with an order of $\cal O (d)$ and thus it is not a fair competition with benchmark algorithms, for example, dp-sgd. However, we did get some theoretical results that this $d$ can be taken away, please refer to Corollary 1.
> >
> > #### Clarity and Quality
> > On your question *"The reviewer does not quite follow why Theorem 1 is a standalone result, it seems to me it is an intermediate lemma for proving convergence, can the authors provide more explanations?"*
> >
> > **Our response:** Sure, we are more than glad to elaborate it. First, we apologize for leaving this alone and have reduced it to a Proposition (Proposition 1.) The intuition behind this is to connect the majority vote aggregation rules with the other renowned Byzantine-resilient aggregation rules in the realm of sign aggregation. It turns out the relation is equivalence, and we present as it is.
> >
> > *"The definition of sub-Gaussian has a typo that one side should be $-t$."
> >
> > **Our response:** We thank the reviewer for pointing this out. This typo has been corrected and the change is highlighted in blue.

---

> > > ### Author Response · Authors · 2022-11-18
> > > **Reference**
> > >
> > > [1] Karimireddy, S. P., He, L.,Jaggi, M. (2021, September). Byzantine-Robust Learning on Heterogeneous Datasets via Bucketing. In International Conference on Learning Representations.
> > >
> > > [2] McMahan, B., Moore, E., Ramage, D., Hampson, S., \& y Arcas, B. A. (2017, April). Communication-efficient learning of deep networks from decentralized data. In Artificial intelligence and statistics (pp. 1273-1282). PMLR.
> > >
> > > [3] Virmaux, A., & Scaman, K. (2018). Lipschitz regularity of deep neural networks: analysis and efficient estimation. Advances in Neural Information Processing Systems, 31.
> > >
> > > [4] Blanchard, P., El Mhamdi, E. M., Guerraoui, R., \& Stainer, J. (2017). Machine learning with adversaries: Byzantine tolerant gradient descent. Advances in Neural Information Processing Systems, 30.
> > >
> > > [5] Chen, Y., Su, L., & Xu, J. (2017). Distributed statistical machine learning in adversarial settings: Byzantine gradient descent. Proceedings of the ACM on Measurement and Analysis of Computing Systems, 1(2), 1-25.
> > >
> > > [6] Karimireddy, S. P., He, L., & Jaggi, M. (2021, July). Learning from history for byzantine robust optimization. In International Conference on Machine Learning (pp. 5311-5319). PMLR.
> > >
> > > [7] Xie, C., Koyejo, O., & Gupta, I. (2020, August). Fall of empires: Breaking byzantine-tolerant sgd by inner product manipulation. In Uncertainty in Artificial Intelligence (pp. 261-270). PMLR.
> > >
> > > [8] Baruch, G., Baruch, M., & Goldberg, Y. (2019). A little is enough: Circumventing defenses for distributed learning. Advances in Neural Information Processing Systems, 32.

---

> > > > ### Comment · Reviewer_W1pz · 2022-11-29
> > > > **Reply to the rebuttal**
> > > >
> > > > Thank the authors for the detailed replies. More theoretical and empirical comparisons with other works make me understand the contributions of this work better. But I am still concerned with the bounded gradient assumption, which, from my experience, is an essential part in convergence analysis. For other contributions of this paper (which I thought is still not clean enough to me), I'll be willing to raise my score. I have no other questions and comments.

---

> > > > > ### Author Response · Authors · 2022-12-13
> > > > > **Thank you**
> > > > >
> > > > > We thank reviewer W1pz for the evaluation and for raising the scores.

---

### Official Review · Reviewer_9qJV · 2022-10-27

**Confidence:** 2
**Clarity, Quality, Novelty And Reproducibility:** 1)Author motivates the subject from a…
**Correctness:** 4
**Technical Novelty And Significance:** 3
**Empirical Novelty And Significance:** 3
**Recommendation:** 6

**Strength And Weaknesses:**

1)Author motivates the subject from a theoretical and practical point of view.
2)In the introduction, the author reviewed the previous works carefully.
3)the author successfully addressed the issue and recommended the algorithms.
4)The authors successfully provide simulation results for their algorithm.
5)The proof is mathematically correct.



**Summary Of The Paper:**

This paper provides the first algorithm based on sign SGD, which achieves Byzantine resilience and differential privacy.

**Summary Of The Review:**

I think this is a good paper and should get accepted. But, the writing should improve.

---

> ### Author Response · Authors · 2022-11-18
> **Thank you for your evaluations!**
>
> We thank Reviewer 9qJV for his evaluation and share our excitement on this work.
>
> The reviewer shared concerns on the presentation of our work. In this revision, we made some major changes outlined below:
> * We compare our work with [Jin et al. 2020] in a point-by-point manner. Our contributions can be appreciated in Table 3;
>
> * We improve our DP guarantee with a finer characterization of gradient distance (Theorem 2) and an upper bound (Corollary 1);
>
> * We complete the case of stochastic gradient noise distribution with a heavy-tailed noise through Rosenthal-type inequality. This gives our $\beta$ stochastic sign compressor a stronger theoretical guarantee with a broader outreach;
>
> * We add empirical results on Byzantine resilience with some baseline algorithms, including Krum [1], Geometric Median [2], Centered Clipping [3], and some adversarial models, including label-flipping attack, inner product manipulation [4], and the "a little is enough" [5];
>
> * For other nomenclatures, we doublecheck to make sure they are properly presented and in place.
>
>
>
>   We hope this revision meets the reviewer's expectation. Please let us know if you have any other concerns.
>
>
>
>   [1] Blanchard, P., El Mhamdi, E. M., Guerraoui, R., \& Stainer, J. (2017). Machine learning with adversaries: Byzantine tolerant gradient descent. Advances in Neural Information Processing Systems, 30.
>
>   [2] Chen, Y., Su, L., & Xu, J. (2017). Distributed statistical machine learning in adversarial settings: Byzantine gradient descent. Proceedings of the ACM on Measurement and Analysis of Computing Systems, 1(2), 1-25.
>
>   [3] Karimireddy, S. P., He, L., & Jaggi, M. (2021, July). Learning from history for byzantine robust optimization. In International Conference on Machine Learning (pp. 5311-5319). PMLR.
>
>   [4] Xie, C., Koyejo, O., & Gupta, I. (2020, August). Fall of empires: Breaking byzantine-tolerant sgd by inner product manipulation. In Uncertainty in Artificial Intelligence (pp. 261-270). PMLR.
>
>   [5] Baruch, G., Baruch, M., & Goldberg, Y. (2019). A little is enough: Circumventing defenses for distributed learning. Advances in Neural Information Processing Systems, 32.

---

### Official Review · Reviewer_YSJ7 · 2022-11-02

**Confidence:** 3
**Correctness:** 3
**Technical Novelty And Significance:** 2
**Empirical Novelty And Significance:** 2
**Recommendation:** 3

**Clarity, Quality, Novelty And Reproducibility:**

**Clarity and Quality:** The presentation can be significantly improved. There are several questions that need more details:

1. What is an upper bound on the number of Byzantine clients that the method can tolerate? The second part of Theorem 5 only considers a lower bound on $\tau(t)$. What is the upper bound on $\tau(t)$ up to which equation (6) holds? As mentioned under ‘Weaknesses’, when the fraction of Byzantine clients is greater than 1/2, then it is not feasible to achieve robustness. Without such an upper bound on $\tau(t)$, it is not possible to assess the correctness of the result, and it is critical to add details.

2. Is there any upper bound on $\beta$? The $\epsilon$ value of the DP guarantee is $d \log\left(\frac{2B + \beta}{\beta}\right)$, and larger values of $\beta$ may be able to lower $\epsilon$. However, the paper does not discuss details about how large can $\beta$ be.

**Novelty:** Considering [Jin et al., 2020], the novelty of the method is fairly limited. The main novelty is in terms of analysis. I did not go through the detailed proofs.

1. The result of Theorem 1 is quite straightforward.

2. For adaptive adversaries, the method can support the following upper bound on the number of Byzantine clients: $\tau(t) \leq \frac{2}{p^2} \log\frac{6}{c}$, where $p$ is a probability with which a client participates in a FL round and $c$ is a positive constant. When $p = O(1)$, then the $\tau(t)$ seems to be a constant, and it would not increase with the number of clients. In contrast, several prior works on Byzantine robust FL can tolerate a constant fraction of Byzantine clients (as opposed to a constant number of Byzantine clients).

3. Empirical results only consider a multi-layer perceptron (MLP) model. It would be helpful to understand how the proposed method performs on more sophisticated models such as CNNs and deeper networks such as ResNet.

**Strength And Weaknesses:**

**Strengths:**

1. The family of sign-based compressors is practically appealing, and the paper proposes an interesting variant of stochastic signSGD.

**Weaknesses:**

1. The paper does not cover all details and the technical presentation leaves open several questions. For instance, the paper does not discuss any upper bound on the number of Byzantine clients $\tau(t)$ in an FL iteration. It is well-known in Byzantine robustness literature that, when the fraction of Byzantine clients is greater than 1/2, then it is not feasible to achieve robustness. Remark 1 discusses the case of sufficiently large $\tau(t)$, which is quite confusing. (Other comments and suggestions are mentioned later).

2. The paper does not give details on whether DP guarantees are for client-level or item-level privacy. There are also no empirical comparisons with DP-SGD and/or local-DP algorithms with the proposed method.

3. The $\epsilon$ of DP for the proposed method is $O(d)$, where $d$ is the number of parameters of ML model (i.e., the length of gradients). This is going to be significantly large for several ML models such as deep neural networks. Would such a high epsilon be practical in terms of DP?


**Summary Of The Paper:**

The paper considers gradient compression in federated/distributed learning with two additional requirements: robustness against Byzantine clients and differential privacy (DP). The paper proposes a sign-based gradient compression method, called $\beta$-stochastic signSGD, which applies clipping and a stochastic sign operator to gradients (Definition 3) for compression. SignSGD, which simply applies a sign operator to gradients was proposed in [Bernstein et al., 2019], and its stochastic version was proposed in [Jin et al., 2020]. The method proposed in the paper reduces to that in [Jin et al., 2020] for $\beta = 0$. The paper presents convergence result and DP guarantees, along with experimental results.

**Summary Of The Review:**

The proposed method of $\beta$-stochastic signSGD and the analysis are interesting. However, there are several questions that need to be addressed before the paper becomes ready for publication.

---

> ### Author Response · Authors · 2022-11-18
> **Byzantine resilience in non-IID, DP, and more**
>
> We thank Reviewer YSJ7 for evaluating our work and appreciating our variant and the family of sign-based methods.
>
> #### Weaknesses
>
> On your question *"The paper does not cover all details and the technical presentation leaves open several questions. For instance, the paper does not discuss any upper bound on the number of Byzantine clients $\tau(t)$ in an FL iteration. It is well-known in Byzantine robustness literature that, when the fraction of Byzantine clients is greater than 1/2, then it is not feasible to achieve robustness. Remark 1 discusses the case of sufficiently large $\tau(t)$, which is quite confusing. (Other comments and suggestions are mentioned later)."*
>
> **Our response:** Thank you for your comments and sharing your concerns. Unfortunately, most of them are misleading specifically in the presence of non-IID data distribution. We improved our presentation in the related work section by investigating Byzantine-resilience in the presence of IID and non-IID data separately.
>
> With IID data, the well-known "middle-seeking" Byzantine resilient algorithms such as Krum [1], Geometric_Median [2], Coordinate-wise median [3], etc. are guaranteed to be robust up to $1/2$ fraction of Byzantine adversaries. Intuitively, this follows because every agent's data looks alike so that the far-away data points are more likely to be malicious.
>
> In sharp contrast, things are completely different in the presence of **non-IID** data. A naive example can be constructed as four clients each having their optimizer as $1,2,4,9$, and thus the global optimizer is $4$. For example, if the client holding a local minimum $9$ becomes an adversary and we actively filter it out, even all the other parties are benigh, the population mean will lean towards 2 instead of the true global mean $4$. That said, Byzantine resilience in FL with non-IID distribution is still an active ongoing research field, and it is non-trivial due to the difficulty of distinguishing the statistical heterogeneity from the Byzantine attacks [4]. This can also be interpreted as *"no single worker is representative of the whole dataset."* [5] That is why the "middle-seeking" Byzantine resilient algorithms often fall apart with non-IID data. The notion of Byzantine resilience in the presence of non-IID data is yet to be defined, and the constant fraction guarantee does not often hold. Here, we define Byzantine resilience as a convergence rate comparable to the Byzantine-free scenario for the asymptotic in terms of $M$.
>
>
>
> [1] Blanchard, P., El Mhamdi, E. M., Guerraoui, R., \& Stainer, J. (2017). Machine learning with adversaries: Byzantine tolerant gradient descent. Advances in Neural Information Processing Systems, 30.
>
> [2] Chen, Y., Su, L., & Xu, J. (2017). Distributed statistical machine learning in adversarial settings: Byzantine gradient descent. Proceedings of the ACM on Measurement and Analysis of Computing Systems, 1(2), 1-25.
>
> [3] Yin, D., Chen, Y., Kannan, R., & Bartlett, P. (2018, July). Byzantine-robust distributed learning: Towards optimal statistical rates. In International Conference on Machine Learning (pp. 5650-5659). PMLR.
>
> [4] Li, L., Xu, W., Chen, T., Giannakis, G. B., \& Ling, Q. (2019, July). RSA: Byzantine-robust stochastic aggregation methods for distributed learning from heterogeneous datasets. In Proceedings of the AAAI Conference on Artificial Intelligence (Vol. 33, No. 01, pp. 1544-1551).
>
> [5] Karimireddy, S. P., He, L.,Jaggi, M. (2021, September). Byzantine-Robust Learning on Heterogeneous Datasets via Bucketing. In International Conference on Learning Representations.
>
> On your question *"The paper does not give details on whether DP guarantees are for client-level or item-level privacy. There are also no empirical comparisons with DP-SGD and/or local-DP algorithms with the proposed method."*
>
> **Our response:** We thank the reviewer for sharing this concern. We assume the reviewer refers to *"client-level"* as *one client's gradient-level* and *"item-level"* as *coordinate-level*. That said, our DP guarantee is client-level instead of item-level. This can be seen from the presence of gradient dimension $d$ in the original draft is based on the composition theorem in [1], as stated in our revision Appendix D.2 Theorem 5 (Theorem 6 in the original draft.) For the empirical results, the original DP guarantee scales with $d$ in the order of $\cal O (d)$, which is much larger than and not comparable to a constant $\epsilon$ in DP-SGD. However, it is fortunate that we have Corollary 1.
>
> [1] Dwork, C., & Roth, A. (2014). The algorithmic foundations of differential privacy. *Foundations and Trends® in Theoretical Computer Science*, *9*(3–4), 211-407.

---

> > ### Author Response · Authors · 2022-11-18
> > **Continued on Weakness**
> >
> > On your question *"The $\epsilon$ of DP for the proposed method is $\cal O(d)$, where $d$ is the number of parameters of ML model (i.e., the length of gradients). This is going to be significantly large for several ML models such as deep neural networks. Would such a high epsilon be practical in terms of DP?"*
> >
> > **Our response**: Thanks for sharing your concerns with us. In the original version, this is indeed the case. In this revision, we added Corollary 1, together with Theorem 2, to show that it is possible to get rid of the order $d$. Then, we can show that $\cal M_{\it B,\beta}$ is $(\frac{1}{\beta})$-DP. The implication is discussed in the added Remark 1 as well. We provide two ways: if one knows the global maximum as in Theorem 2, then the constant DP guarantee is trivial; however, as it is often hard to get in practice, then one can look at Corollary 1.
> >
> > #### Clarity and Quality
> > On your question *"What is an upper bound on the number of Byzantine clients that the method can tolerate? The second part of Theorem 5 only considers a lower bound on $\tau(t)$. What is the upper bound on $\tau(t)$ up to which equation (6) holds? As mentioned under ‘Weaknesses’, when the fraction of Byzantine clients is greater than 1/2, then it is not feasible to achieve robustness. Without such an upper bound on $\tau(t)$
> > , it is not possible to assess the correctness of the result, and it is critical to add details."*
> >
> > **Our response:** We thank the reviewer's evaluations on Byzantine resilience. However, as we pointed out in the first response, this constant fraction resilience is misleading and even impossible in the presence of non-IID data. Our Byzantine resilience is discussed in an asymptotic view and, for example, we allow the Byzantine clients' number $\tau(t)=\tau=\cal O(M^{\frac{\alpha}{2}})~(0<\alpha\le1)$.
> >
> > On your question *"Is there any upper bound on $\beta$? The $\epsilon$ value of the DP guarantee is $d\log(\frac{2B+\beta}{\beta})$, and larger values of $\beta$ may be able to lower $\epsilon$. However, the paper does not discuss details about how large can $\beta$ be."*
> >
> > **Our response:** We thank the reviewer for discussing the possibility of lowering the DP quantification. The choice of $\beta$ plays an important role in both privacy guarantees and convergence bounds. From the perspective of system convergence, if $\beta={\cal{O}}(M^{\alpha})~(0\le\alpha<1/2)$, $\tau(t)=\tau={\cal{O}}(\sqrt{M})$, and let client number $M\to \infty$, the convergence bound reaches a stationary point at the sacrifice of convergence speed. For a more general case, we shall interplay with the product of $\beta$ and $1/T\sum_{t=0}^{T-1}\tau(t)$. Specifically, let $\beta/T\sum_{t=0}^{T-1}\tau(t)=\cal O (M^{\alpha})(0\le\alpha<1)$ .
> >
> > #### Novelty:
> >
> >  Considering [Jin et al., 2020], the novelty of the method is fairly limited. The main novelty is in terms of analysis. I did not go through the detailed proofs.
> >
> > **Our response:** Thanks for sharing your concerns on novelty with us. We refer the reviewer to take a look at our general response by the authors and a point-by-point comparison in Appendix.(A) of the revision. Although we admit the compressors are structurally alike, one of our upfront contributions is to explicitly push out the residual terms in the convergence bound, which are missing in the existing literature.
> >
> > On your question *"The result of Theorem 1 is quite straightforward."*
> >
> > **Our response:** We thank the reviewer for pointing this out. We revised our presentations on the aggregation function for a better presentation in the revision. The Theorem 1 is reduced to Proposition 1, the motivation is to show the connection between majority vote aggregation rule and the other many renowned Byzantine resilient aggregation rules, i.e., median and trimmed-mean, when we map the gradients to sign-bits, and we count the sign of the outcomes. In this case, the connection is equivalence.

---

> > > ### Author Response · Authors · 2022-11-18
> > > **Continued**
> > >
> > > On your question *"For adaptive adversaries, the method can support the following upper bound on the number of Byzantine clients: $\tau(t)\le\frac{2}{p^2}\log\frac{6}{c}$, where $p$ is a probability with which a client participates in a FL round and $c$ is a positive constant. When $p=\cal O(1)$, then the $\tau(t)$ seems to be a constant, and it would not increase with the number of clients. In contrast, several prior works on Byzantine robust FL can tolerate a constant fraction of Byzantine clients (as opposed to a constant number of Byzantine clients)."*
> > >
> > > **Our response:** In Remark 2, we comment that *"The lower bound requirement on $\tau(t)$ might be an artifact of our analysis in simplifying the boundary case derivation."* in the first bullet point. We do not require $\tau(t)=\cal O (1)$ since the upper bound the reviewer referred to is merely a threshold dividing the case. To see this, we discuss the asymptotic w.r.t. client number $M$ in the second bullet point. Specifically, if we consider a constant number of Byzantine clients and $\tau(t)=\tau=\cal O (\sqrt{M})$, the Byzantine residual term will scale with order $\cal O (\frac{1}{\sqrt{M}})$ and becomes non-dominating as $M\to\infty$.
> > >
> > > On your question *"Empirical results only consider a multi-layer perceptron (MLP) model. It would be helpful to understand how the proposed method performs on more sophisticated models such as CNNs and deeper networks such as ResNet."*
> > >
> > > **Our response:** We thank the reviewer for sharing this concern. The additional experiments are still on-going, and if time permits, we are more than glad to share the results with the reviewers and the community. However, we did add some empirical comparisons on Byzantine resilience in Appendix.(E3) with some baseline algorithms, including Krum [1], Geometric Median [2], Centered Clipping [3], and some adversarial models, including label-flipping attack, inner product manipulation [4], and the "a little is enough" [5].
> > >
> > > [1] Blanchard, P., El Mhamdi, E. M., Guerraoui, R., \& Stainer, J. (2017). Machine learning with adversaries: Byzantine tolerant gradient descent. Advances in Neural Information Processing Systems, 30.
> > >
> > > [2] Chen, Y., Su, L., & Xu, J. (2017). Distributed statistical machine learning in adversarial settings: Byzantine gradient descent. Proceedings of the ACM on Measurement and Analysis of Computing Systems, 1(2), 1-25.
> > >
> > > [3] Karimireddy, S. P., He, L., & Jaggi, M. (2021, July). Learning from history for byzantine robust optimization. In International Conference on Machine Learning (pp. 5311-5319). PMLR.
> > >
> > > [4] Xie, C., Koyejo, O., & Gupta, I. (2020, August). Fall of empires: Breaking byzantine-tolerant sgd by inner product manipulation. In Uncertainty in Artificial Intelligence (pp. 261-270). PMLR.
> > >
> > > [5] Baruch, G., Baruch, M., & Goldberg, Y. (2019). A little is enough: Circumventing defenses for distributed learning. Advances in Neural Information Processing Systems, 32.

---

> > > > ### Comment · Reviewer_YSJ7 · 2022-12-01
> > > > **Thank you for the response**
> > > >
> > > > I thank the authors for clarifications, and for their efforts in improving the paper. Unfortunately, however, several concerns still remain. Therefore, I retain my original score.
> > > >
> > > > * For an upper bound on the fraction of Byzantine clients, the authors mention that the key difference from the prior work is due to non-IID data. However, upper bound on the number of Byzantine workers typically comes from information theoretic arguments, depending on the threat model. As an example, for an omniscient adversary (which can observe all gradients), 1/2 fraction of Byzantine clients can stall the convergence. For instance, consider binary vectors $( +/- 1 )^d$ with four clients. Let’s take $d=1$ for simplicity. If clients 3 and 4 are Byzantine, then they form their inputs as the flipped signs of the inputs of clients 1 and 2, respectively. That is, if clients 1 and 2 have their inputs +1, -1, then clients 3 and 4 will choose their inputs as -1, +1. With this attack, the output of average (trimmed mean, median, or majority) will always be 0, and convergence is not possible.
> > > >
> > > > * The case of $\tau(t) \leq \frac{2}{p^2} \log\frac{6}{c}$ is a threshold dividing the regime, however, for adaptive adversary, only this case is considered. So, the result for adaptive adversary is quite limited.
> > > >
> > > > * It would be helpful to add details of what would be neighboring datasets for the DP definition.

---

> > > > > ### Author Response · Authors · 2022-12-13
> > > > > **Follow-up Response**
> > > > >
> > > > > We thank reviewer YSJ7 for sharing further concerns with us.
> > > > >
> > > > > * We discuss the Byzantine tolerance upper bound in asymptotic w.r.t. client number $M$. Our bound can not cover $\Theta(M)$, to which $\frac{1}{2}$ fraction belongs. That said, in the last response, we said the constant fractional guarantee is not possible in our compressor but did not disagree with the $\frac{1}{2}$ upper bound.
> > > > >
> > > > > * The convergence bound for the **adaptive** adversaries is the first inequality. The dividing threshold is only for **static** adversaries. As one can check the two convergence bounds cover all $\tau(t)\in\mathbb{R}$, we do not impose any restrictions on the number of Byzantine clients in the analysis. We refer interested readers to the proof, and it is immediately evident.
> > > > >
> > > > > * As one can see from Example 1, we are protecting the raw gradient and thus talking about the adjacent gradient pair of $\mathbf{g}$ and $\mathbf{g}^\prime$  as $\\|\mathbf{g}-\mathbf{g}^\prime\\|_1\le 1.$

---

### Author Response · Authors · 2022-11-16
**General Response by Authors**

Dear area chair and reviewers,

We would like to thank the area chair for handling the paper and all the reviewers for their time and efforts in evaluating this paper. Overall, the reviewers thought our work is interesting, practical and easy to implement (YSJ7, W1pz, nUrq), empirically sound (W1pz), and theoretically correct (9qJV). The major concerns lie in our novelty compared with [Jin et al. 2020] (YSJ7, W1pz, nUrq), Theorem 1 (YSJ7, W1pz), presentations (YSJ7,nUrq), additional empirical evidence (YSJ7, W1pz, nUrq). The concerns and suggestions are very helpful. We are working on improving the writing and running experiments as suggested.  We do agree that there is some room to improve our algorithmic novelty; however, at least part of the novelty concerns arise from misunderstandings, which we would like to clarify upfront. We highlighted all our changes in blue. We first restate our contributions and resolve some big-picture issues.

**Our contributions:**
 Our contributions to federated learning (FL) are two-fold. While we admit our compressors are structurally alike with [Jin et al. 2020], our contributions are more than adding differential privacy, and our theoretical analysis is non-trivial. The details can be found below. We reserve a point-by-point comparison in Appendix.(A).
* On the theoretical side:
	* We explicitly characterize the impacts of client number $M$, compressor hyperparameter $B$, privacy quantification $\beta$, mini-batch size $n$, and adaptive Byzantine adversaries $\tau(t)$, which are missing in existing literature. We want to emphasize that such characterization is essential. At a high level, the stochastic sign compression becomes highly noisy when the gradient magnitude is small. Hence, the encoded small gradients' magnitude cannot be passed through to the parameter server, which adds up and leads to a residual term. This is observed in [Jin et al. 2020] Remark 3 but without explicit form. For Byzantine resilience, the same problem peresists.
  * On the other hand, [Jin et al. 2020] adopts only the *full-batch* gradient. We refer interested readers to look at Eq.(33) and Theorem 1. In sharp contrast, we adopt mini-batch stochastic gradients and show that our algorithm converges in the presence of both bounded and unbounded ones. We want to note that extending the analysis to stochastic gradients is imperative and technically non-trivial:
    * First, the computing clients in FL are often the edge devices with limited computing power and storage, such as smartphones, which are not always able to compute the true gradients [1].
    * Next, the stochasticity of our compressor requires a careful characterization of the distance between stochastic and true gradient using a variety of concentration bounds.
    * Finally, we show that our compressor works for both bounded and unbounded stochastic gradients. By comparison, [Jin et al. 2020] considers only bounded gradient in Remark 2, which may not hold in the case of stochastic gradient. To complete the case, we add the analysis for heavy-tailed stochastic gradients using Rosenthal-type inequalities in the revision.
* On the algorithmic side:
	* For client sampling, we consider partial client participation, which is more realistic in practice. This enables us to consider a more complex Byzantine adversary model, which we call adaptive Byzantine adversaries. The Byzantine adversaries will be unanimously admitted into the sample set and craft tailored messages to poison PS. Besides, a standard static Byzantine adversary model is investigated as well, where all the clients participate training process independently irrespective of their malicious status.
	* For privacy preservation, we introduce a parameter $\beta$ to guarantee DP. In hindsight, we decompose our compressor $\cal M_{\it B,\beta}$ as $\cal M_{\it B,0}$ and $\cal M_{\it B,flip}$ to help readers interpret our privacy quantification. However, rather than introducing random flipping, we note that our contribution is to propose $\beta$ stochastic compressor, which is a one-step compressor and an enabler of both DP and Byzantine resilience in the presence of data heterogeneity. To the best of our knowledge, this is the first work to ensure provable DP with signed compression in FL;
	* For Byzantine resilience, in addition to adaptive and static Byzantine adversaries,  mobile adversaries are considered, where Byzantine clients can change at any given time. A direct consequence of this is that each client may be corrupted at least once during the whole trajectory. We give an explicit Byzantine residual term as ${\cal O}(\frac{\sum_{i=0}^{T-1} (B+\beta)\tau(t)} {TM}).$ Some possible interpretations are discussed in Remark 2 of the revision.

[1] McMahan, B., Moore, E., Ramage, D., Hampson, S., \& y Arcas, B. A. (2017, April). Communication-efficient learning of deep networks from decentralized data. In Artificial intelligence and statistics (pp. 1273-1282). PMLR.

---

> ### Author Response · Authors · 2022-11-24
> **Corrections**
>
> We thank Reviewer nUrq and want to make the following corrections in the part of Privacy Preservation, which mainly relate to a finer characterization of our DP in the revision.
>
> **Theorem 2**
>
> Let $\beta>0$ and $\cal G\subseteq \mathbb{R}^d$. $\cal M_{\it B,\beta}$ is - $\max_{\mathbf{g}\in \cal G}\sum_{i=1}^d \log\left({1 + \frac{1}{\beta + \mathsf{dist}\left({g_i, \cal C_{\it B}}\right)}}\right)$-differentially private on $\cal G$.
>
>
> **Corollary 1**
>
> Given the same definitions as in Theorem 2, $\cal M_{\it B,\beta}$ is $\left({\frac{1}{\beta}}\right)$-DP.
>
> **Remark 1:** Theorem 2 gives a finer characterization of the differential privacy preserved by $\cal M_{\it B,\beta}$ when $\beta>0$. Unfortunately, this maximum is often hard to find, so Corollary 1 tells us that we can get rid of the order $\cal O({d})$ by looking at an upper bound of Theorem 2.

---

### Decision · Program_Chairs · 2023-01-20

**Decision:**

Reject

**Justification For Why Not Higher Score:**

Due to the weaknesses above.

**Justification For Why Not Lower Score:**

N/A

**Metareview: Summary, Strengths And Weaknesses:**

The paper proposed a new optimizer for federated learning, generalizing upon a stochastic version of signSGD due to (Jin et al. 2020), that enjoys byzantine robustness and differential privacy.  Reviewers liked the simplicity of the algorithm and the promising set of experiments (though mostly on toy-scale datasets), but find the theoretical results incremental or not quite informative (e.g., with large epsilon parameters for DP than typically considered a meaningful DP guarantee).   Several reviewers also found the paper hard to read.

Some of these issues were clarified in the rebuttal, but it does not change the overall level of excitement among reviewers.